# SWIFT-FEDGNN: FEDERATED GRAPH LEARNING WITH LOW COMMUNICATION AND SAMPLE COMPLEXITIES

## ABSTRACT

Graph neural networks (GNNs) have achieved great success in a wide variety of graph-based learning applications. While distributed GNN training with sampling-based mini-batches expedites learning on large graphs, it is not applicable to geo-distributed data that must remain on-site to preserve privacy. On the other hand, federated learning (FL) has been widely used to enable privacy-preserving training under data parallelism. However, applying FL directly to GNNs either results in cross-client neighbor information loss or incurs expensive cross-client neighbor sampling and communication costs due to the large graph size and the dependencies between nodes among different clients. To overcome these challenges, we propose a new federated graph learning (FGL) algorithmic framework called Swift-FedGNN that primarily performs efficient parallel local training and periodically conducts cross-client training. Specifically, in Swift-FedGNN, each client *primarily* trains a local GNN model using only its local graph data, and some randomly sampled clients *periodically* learn the local GNN models based on their local graph data and the dependent nodes across clients. We theoretically establish the convergence performance of Swift-FedGNN and show that it enjoys a convergence rate of $\mathcal{O}\left(T^{-1/2}\right)$, matching the state-of-the-art (SOTA) rate of sampling-based GNN methods, despite operating in the challenging FL setting. Extensive experiments on real-world datasets show that Swift-FedGNN significantly outperforms the SOTA FGL approaches in terms of efficiency, while achieving comparable accuracy.

## 1 INTRODUCTION

**1) Background and Motivation:** Graph neural networks (GNNs) have received increasing attention in recent years and have been widely used across various applications, such as social networks Deng et al. (2019); Qiu et al. (2018), recommendation systems Ying et al. (2018); Wang et al. (2019a), and drug discovery Wang et al. (2022b); Do et al. (2019). GNN learns high-level graph representations by iteratively aggregating neighboring features of each node, which is then used for downstream tasks, such as node classification Kipf & Welling (2017); Hamilton et al. (2017), link prediction Yao et al. (2023b); Zhang & Chen (2018), and graph classification Zhang et al. (2018); Bacciu et al. (2018).

Real-world graph datasets can be extensive in scale (*e.g.*, Microsoft Academic Graph Wang et al. (2020) with over 100 million nodes) and often reside across geo-distributed sites where data protection laws prohibit direct data sharing Yao et al. (2023a). Single devices (*e.g.*, GPUs) often lack the capacity for training such large-scale datasets, which leads to *a compelling need* for distributed graph learning (DGL) Fey & Lenssen (2019); Zheng et al. (2020). However, the common DGL paradigm, consisting of subgraph sampling Zeng et al. (2020) and mini-batch training Luo et al. (2022), requires direct data sharing among workers, which conflicts with privacy regulations.

Meanwhile, federated learning (FL) McMahan et al. (2017); Yang et al. (2021); Karimireddy et al. (2020), which has emerged as a promising learning paradigm, enables collaborative training of a model using geo-distributed traditional datasets under the coordination of a central server. However, applying FL to geo-distributed graph data is highly non-trivial due to the dependencies between the nodes in a graph and the fact that the neighbors of the node may be located on different clients, which we refer to as *"cross-client neighbors"* (shown as the dashed links between nodes in Figure 1). Ignoring the cross-client neighbors as in Wang et al. (2022a) would degrade the performance of

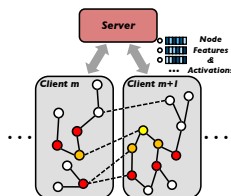
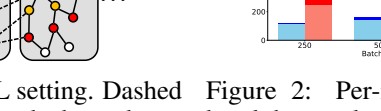
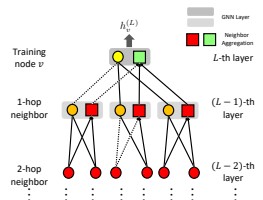

Figure 1: FGL setting. Dashed lines show graph dependency cross-clients.

Figure 2: Per-iteration time breakdown: local vs. cross-client training.

Figure 3: Federated GNN model. Dashed lines show communication between clients.

the models and prevent them from reaching the same accuracy as the models trained on a single device/machine, which is due to the information loss of the cross-client neighbors.

**2) Technical Challenges:** Despite the appeal of leveraging a trusted server for federated graph learning (FGL) Zhang et al. (2021), there remain several non-trivial challenges that hinder efficient and effective cross-client training. Specifically, we highlight the following major technical challenges:

**Large Overhead from a Naive Design.** A straightforward approach uses a trusted server to gather graph data and perform subgraph sampling and neighbor aggregation for each client Zhang et al. (2021). For instance, in a healthcare setting with multiple hospitals and one central authority, patient data stay locally due to privacy regulation. The central authority acting as the trusted server must coordinate all subgraph sampling and part of the training operation (*i.e.*, neighbor aggregation) (see Figure 1). As shown in Figure 2, training a two-layer GNN (with sampling fanout values 15 and 10 for the two layers) on the Amazon product co-purchasing dataset Leskovec et al. (2007) under 1 Gbps network bandwidth, this approach leads to significant communication overhead: the server exchanges large amounts of node and edge information with each hospital sequentially, causing cross-client sampling and communication time to dominate the *total* training time, making it *five times* slower than purely local training.

**Communication and Memory Overheads from Cross-Client Neighbors.** While some methods ignore cross-client neighbors He et al. (2021) or assume overlapping nodes Wu et al. (2021), these assumptions often fail in geo-distributed graphs (*e.g.*, patients visiting multiple hospitals). Alternatives that preserve cross-client neighbor information Zhang et al. (2021); Du & Wu (2022); Yao et al. (2023a) require significant data transfers among clients—leading to high communication costs—and compel each client to store additional graph structure and features for these neighbors. This not only creates memory-intensive requirements but could also *potentially violate data privacy constraints*. Hence, mitigating these cross-client overheads (both communication and storage) is crucial to achieve efficient, privacy-preserving FGL (see detailed discussions in Section 2).

**3) Our Contributions:** The key contribution of this paper is that, by addressing the above challenges, we develop a mini-batch-based and sampling-based FGL framework called Swift-FedGNN. The main results and technical contributions of this paper are as follows:

- We develop Swift-FedGNN, a communication- and sample-efficient mini-batch FGL algorithm for geo-distributed graphs. In Swift-FedGNN, clients *primarily* conduct local training in parallel, performing cross-client training *only occasionally* among sampled clients, thereby reducing sampling and communication overhead while preserving minimal information loss. The cross-client neighbor information is aggregated at remote clients before communicating to the server and accumulated one more time before transferring to the training client, further minimizing data transfer cost and enhancing privacy by ensuring only aggregated neighborhood features - never raw node features - are exchanged.

- We conduct rigorous theoretical convergence analysis for Swift-FedGNN, which is highly non-trivial due to *biased* stochastic gradients and *structural entanglement* (neighbor aggregation intertwined with non-linear transformations across multiple layers) in GNNs . In stark contrast to existing works in the literature that made strong assumptions on the biases of stochastic gradients (*e.g.*, unbiased Chen et al. (2018) or consistent Chen & Luss (2018) gradient), for the *first time* in the literature, we are able to *bound* stochastic gradient approximation errors rather than resorting to these unrealistic assumptions in practice, offering insights of independent theoretical interest.

- We show that the *biased* stochastic gradients in GNNs—arising from missing cross-client neighbors and neighbor sampling—are *positively correlated* with the network depth, which is *unique* to FGL.

By putting the above insights together, we show that Swift-FedGNN achieves a convergence rate of $\mathcal{O}\left(T^{-1/2}\right)$, which *matches* the state-of-the-art (SOTA) convergence rate of sampling-based GNN methods (hence low communication and sample complexities), *despite* operating in the far more challenging FL setting with much less frequent information exchanges among clients.

• We conduct extensive experiments on real-world graph datasets to evaluate the performance of Swift-FedGNN. The results show that Swift-FedGNN outperforms the SOTA FGL algorithms in terms of *efficiency*, achieving $\times 4$ speed-up and competitive accuracy.

## 2 RELATED WORK

In this section, we provide an overview on distributed graph learning and offer a comprehensive comparison with the most relevant work on federated graph learning.

**1) Distributed Graph Learning:** Distributed graph learning framework (*e.g.*, DistDGL Wang et al. (2019b); Zheng et al. (2020), Pytorch Geometric Fey & Lenssen (2019), AliGraph Zhao et al. (2019) and Dorylus Thorpe et al. (2021)) have been developed to train large-scale graph datasets via cross-device sampling and direct worker-to-worker communication, and often spend up to 80% of the total training time on data communication Gandhi & Iyer (2021). Although various optimizations (graph partitioning Zheng et al. (2020), caching Liu et al. (2023); Zhang et al. (2023), communication strategies Cai et al. (2021); Luo et al. (2022), parallel training Gandhi & Iyer (2021); Wan et al. (2022); Du et al. (2024)) have been proposed to expedite DGL, they commonly require direct data sharing between workers, violating data privacy constraints in geo-distributed settings. To our knowledge, LLCG Ramezani et al. (2022) is the only DGL framework that avoids transferring node features between workers, making it potentially applicable to geo-distributed graphs. In LLCG, each worker trains only on its local graph partition. To address missing cross-device neighbor information, LLCG employs a central server to periodically perform full-neighbor training with neighbor aggregation across all workers. However, this approach imposes significant communication overhead on the server, which needs to communicate with every worker to perform the full-neighbor training.

**2) Federated Graph Learning:** To date, the research on federated graph learning remains in its infancy and results in this area are quite limited. In He et al. (2021), it is assumed that graphs are dispersed across multiple clients and the information of the cross-client neighbors is ignored, which does not align with the real-world scenarios and would degrade the performance of the trained model. In Wu et al. (2021), it is assumed that the clients' local graphs have overlapped nodes and the edges are distributed, which may not be true in real-world situations. Zhang et al. (2021) mitigates the information loss of the cross-client neighbors by exchanging such information in each training round. However, this approach incurs considerable communication overhead and exposes private node information to other clients. While Yao et al. (2023a) employs a one-time exchange of full cross-client neighbor information prior to training, this design relies on full-graph training and causes significant per-client memory overhead, making it impractical for large-scale graphs. Adapting it to sampling-based FGL would require per-iteration cross-client exchanges (since each mini-batch has a different training node set and sampled neighbors), further exacerbating communication overhead.

Du & Wu (2022) uses sparse cross-client neighbor sampling to supplement the lost information of the cross-client neighbors and reduce the communication overhead, which is most related to ours. Each client periodically samples and exchanges these neighbors with other clients, reusing the most recent sampled neighbors in between exchanges. However, as training progresses, the frequency of information exchange increases, leading to higher communication costs. Furthermore, privacy constraints are relaxed by allowing direct client-to-client data transfers and caching, and repeatedly reusing the same neighbor data introduces bias that degrades performance. In contrast, our Swift-FedGNN method limits cross-client training to a *subset* of sampled clients and avoids direct graph data exchange between clients by offloading certain operations to the central server. Before communication with the training clients, cross-client neighbor information is aggregated twice: first at the remote clients and then on the server—helping to preserve data privacy and reduce communication costs.

## 3 FEDERATED GRAPH LEARNING: PRELIMINARIES

In this section, we provide the background of the mathematical formulation for training GNNs in a federated setting. For convenience, we provide a list of key notations used in this paper in

Appendix B. In order for this paper to be self-contained and to facilitate easy comparisons, we provide the background for training GNNs on a single machine in Appendix C.

Consider a graph $\mathcal{G}(\mathcal{V}, \mathcal{E})$, where $\mathcal{V}$ is a set of nodes with $N = |\mathcal{V}|$ and $\mathcal{E}$ is a set of edges. We consider a standard federated setting that has a central server and a set of $\mathcal{M}$ clients with $M = |\mathcal{M}|$. The graph $\mathcal{G}$ is geographically distributed over these clients, and each client $m$ contains a subgraph represented by $\mathcal{G}^m(\mathcal{V}^m, \mathcal{E}^m)$. Note that $\bigcup_{m=1}^{M} \mathcal{G}^m \neq \mathcal{G}$ due to the missing cross-client edges between clients ($\bigcup_{m=1}^{M} \mathcal{E}^m \neq \mathcal{E}$). In addition, we assume that the nodes are *disjointly* partitioned across clients, *i.e.*, $\bigcup_{m=1}^{M} \mathcal{V}^m = \mathcal{V}$ and $\bigcap_{m=1}^{M} \mathcal{V}^m = \emptyset$. Each node $v \in \mathcal{V}^m$ has a feature vector $\boldsymbol{x}_v^m \in \mathbb{R}^d$, and each node $v \in \mathcal{V}_{train}^m$ corresponds to a label $y_v^m$, where $\mathcal{V}_{train}^m \subseteq \mathcal{V}^m$.

In FGL, the clients collaboratively learn a model with distributed graph data and under the coordination of the central server. Typically, the clients receive the model from the server, compute local model updates iteratively, and then send the updated model to the server. The server periodically aggregates the models and then sends the aggregated model back to the clients. The goal in FGL is to solve the following optimization problem:

$$\min \mathcal{L}(\boldsymbol{\theta}) := \frac{1}{|\mathcal{M}|} \sum_{m \in \mathcal{M}} F^m(\boldsymbol{\theta}) = \frac{1}{|\mathcal{M}|} \sum_{m \in \mathcal{M}} \frac{1}{|\mathcal{V}_B^m|} \sum_{v \in \mathcal{V}_B^m} \ell^m\left(\boldsymbol{h}_v^{(L),m}, y_v^m\right), \tag{1}$$

where $\ell^m$ is a loss function (*e.g.*, cross-entropy loss) at client $m$, $\mathcal{V}_B^m$ denotes a mini-batch of training nodes uniformly sampled from $\mathcal{V}^m$, and $\boldsymbol{\theta} := \left\{\boldsymbol{W}^{(l)}\right\}_{l=1}^{L}$ corresponds to all model parameters.

GNNs aim to generate representations (embeddings) for each node in the graph by combining information from its neighboring nodes. Recall that in FGL, the neighbors of node $v$ may be located on its local client $m(v)$ or on remote clients $\bar{m}(v) \in \bar{\mathcal{M}}(v)$, where $\bar{\mathcal{M}}(v)$ represents a set of the remote clients that host the neighbors of node $v$, and $\bar{\mathcal{M}}(v) \subseteq \mathcal{M} \setminus \{m(v)\}$. As shown in Figure 3, to compute the embedding of node $v$ at the $l$-th layer in a GNN with $L$ layers, the client $m(v)$ first aggregates the neighbor information from both itself and the remote clients $\bar{m}(v)$, and then updates the embedding of node $v$, as follows:

$$\boldsymbol{h}_{\mathcal{N}(v)}^{(l)} = \mathrm{AGG}\left(\underbrace{\left\{\boldsymbol{h}_u^{(l-1),m(v)} \mid u \in \mathcal{N}^{m(v)}(v)\right\}}_{\text{local}} \cup \underbrace{\left\{\cup_{\bar{m}(v) \in \bar{\mathcal{M}}(v)} \left\{\boldsymbol{h}_u^{(l-1),\bar{m}(v)} \mid u \in \mathcal{N}^{\bar{m}(v)}(v)\right\}\right\}}_{\text{remote}}\right),$$

$$\boldsymbol{h}_v^{(l),m(v)} = \sigma\left(\boldsymbol{W}^{(l)} \cdot \mathrm{COMB}\left(\boldsymbol{h}_v^{(l-1),m(v)}, \boldsymbol{h}_{\mathcal{N}(v)}^{(l)}\right)\right), \tag{2}$$

where $\mathcal{N}^{m(v)}(v)$ is a set of the neighbors of node $v$ located on its local client $m(v)$, $\mathcal{N}^{\bar{m}(v)}(v)$ is a set of the neighbors of node $v$ located on remote client $\bar{m}(v)$, $\boldsymbol{h}_{\mathcal{N}(v)}^{(l)}$ is the aggregated embedding from node $v$'s neighbors, $\boldsymbol{h}_v^{(l),m(v)}$ is the embedding of node $v$ located on client $m(v)$ and is initialized as $\boldsymbol{h}_v^{(0),m(v)} = \boldsymbol{x}_v^{m(v)}$, $\boldsymbol{W}^{(l)}$ represents the weight matrix at $l$-th layer, $\sigma(\cdot)$ corresponds to an activation function (*e.g.*, ReLU), $\mathrm{AGG}(\cdot)$ is an aggregation function (*e.g.*, mean), and $\mathrm{COMB}(\cdot)$ is a combination function (*e.g.*, concatenation). Compared to DGL where clients can directly transfer node features, the *key difference* in FGL is that clients *cannot* do so due to privacy concerns, requiring additional modifications.

## 4 THE Swift-FedGNN ALGORITHM

In this section, we propose a new algorithmic framework called Swift-FedGNN , designed to efficiently solve Problem (1) by reducing both sampling and communication costs in FGL. The overall algorithmic framework of Swift-FedGNN is illustrated in Algorithms 1-3. Rather than each client performing cross-client training in every round, the clients in Swift-FedGNN primarily conduct the *efficient* local training in parallel, and a set of randomly selected clients periodically carry out the time-consuming cross-client training. By offloading part of the graph operation to the server and remote clients, Swift-FedGNN eliminates the need for sharing graph features among clients.

Algorithm 1 outlines the main framework of Swift-FedGNN. Specifically, it performs parallel local training across clients for every $I - 1$ iterations, followed by one iteration of cross-client training involving randomly selected clients. In the local training iterations ($t$), every client $m$ updates the local GNN model only using its local graph, as presented in Algorithm 3. Client $m$ samples a

**Algorithm 1:** Swift-FedGNN Algorithm.

**Input:** Initial parameters $\boldsymbol{\theta}_0$, learning rate $\alpha$, and correction frequency $I$
**for** $t = 0$ **to** $T - 1$ **do**
  **if** $t \bmod I = 0$ **then**
    Randomly sample $|\mathcal{K}|$ clients
    **for** $m \in \mathcal{M}$ **in parallel do**
      **if** $m \in \mathcal{K}$ **then**
        Client update with local graph and cross-client neighbors using Algorithm 2
      **else**
        Client update with local graph using Algorithm 3
  **else**
    **for** $m \in \mathcal{M}$ **in parallel do**
      Client update with local graph using Algorithm 3
  **Server:**
  Aggregate and update global model parameter as:
$$\boldsymbol{\theta}_{t+1} = \boldsymbol{\theta}_t - \alpha \frac{1}{|\mathcal{M}|} \sum_{m \in \mathcal{M}} \nabla \widetilde{F}^m (\boldsymbol{\theta}_t^m)$$

**Algorithm 2:** Client $m$ in the $t$-th iteration: update with local graph data and cross-client neighbors.

Receive global parameter $\boldsymbol{\theta}_t^m = \boldsymbol{\theta}_t$
Construct a mini-batch $\mathcal{B}_v^m$ of nodes
Server samples a subset of $L$-hop neighbors $\mathcal{S} = \left\{ \mathcal{S}^{(l)} \right\}_{l=0}^{L-1}$ for the training nodes in $\mathcal{B}_v^m$
**for** $l = 1$ **to** $L$ **do**
  /* Derive $l$-th layer embedding of node $v \in \mathcal{B}_v^m$ if $l = L$, otherwise $v \in \mathcal{S}^{(l)}$ */
  **for** *Remote client* $\bar{m}(v) \in \bar{\mathcal{M}}(v)$ **in parallel do**
    Aggregate the neighbor embeddings using Eq. (5)
    Send the aggregated embedding $\boldsymbol{h}_{\mathcal{N}(v)}^{(l),\bar{m}(v)}$ to server
  **Server:**
  Aggregate the neighbor embeddings from the remote clients using Eq. (6)
  Send the aggregated cross-client neighbor embedding $\boldsymbol{r}_{\mathcal{N}(v)}^{(l)}$ to Client $m(v)$
  **Client** $m(v)$**:** Compute node embeddings using Eq. (7) & (8)
Compute stochastic gradient $\nabla \widetilde{F}^m (\boldsymbol{\theta}_t^m)$ and send to server

**Algorithm 3:** Client $m$ in the $t$-th iteration: update with local graph data.

Receive global parameter $\boldsymbol{\theta}_t^m = \boldsymbol{\theta}_t$
Construct a mini-batch $\mathcal{B}_v^m$ of nodes
Sample a subset of $L$-hop neighbors $\mathcal{S} = \left\{ \mathcal{S}^{(l)} \right\}_{l=0}^{L-1}$ for the training nodes in $\mathcal{B}_v^m$
**for** $l = 1$ **to** $L$ **do**
  /* Derive $l$-th layer embedding of node $v \in \mathcal{B}_v^m$ if $l = L$, otherwise $v \in \mathcal{S}^{(l)}$ */
  Compute node embeddings using Eq. (3) and (4)
Compute stochastic gradient $\nabla \widetilde{F}^m (\boldsymbol{\theta}_t^m)$ and send to server

mini-batch of training nodes $\mathcal{B}_v^m$ and a subset of $L$-hop neighbors for the training nodes in $\mathcal{B}_v^m$, denote as $\mathcal{S} = \left\{ \mathcal{S}^{(l)} \right\}_{l=0}^{L-1}$, all from the local graph data. To compute the embedding of node $v$ in the $l$-th GNN layer ($v \in \mathcal{B}_v^m$ if $l = L$, otherwise $v \in \mathcal{S}^{(l)}$), client $m$ first conducts the neighbor aggregation for node $v$ based on the sampled neighbors using:

$$\boldsymbol{h}_{\mathcal{N}(v)}^{(l)} = \text{AGG} \left( \left\{ \boldsymbol{h}_u^{(l-1),m} \mid u \in \widetilde{\mathcal{N}}^m (v) \right\} \right), \tag{3}$$

where $\widetilde{\mathcal{N}}^m (v)$ denotes a set of the sampled neighbors located on client $m$ for node $v$, $\widetilde{\mathcal{N}}^m (v) \subseteq \mathcal{S}^{(l-1)}$, and $\widetilde{\mathcal{N}}^m (v) \subseteq \mathcal{N}^m (v)$. Then, client $m$ updates the embedding of node $v$ in the $l$-th GNN layer based on the aggregated neighbor information and the embedding of node $v$ from the $(l-1)$-th layer, as follows:

$$\boldsymbol{h}_v^{(l),m} = \sigma \left( \boldsymbol{W}_t^{(l),m} \cdot \text{COMB} \big( \boldsymbol{h}_v^{(l-1),m}, \boldsymbol{h}_{\mathcal{N}(v)}^{(l)} \big) \right). \tag{4}$$

At every $I$-th iteration, Swift-FedGNN allows a set of $K$ clients, uniformly sampled from $\mathcal{M}$, to conduct cross-client training that trains the local GNN models using both their local graph data and the cross-client neighbors. We use $\mathcal{K}$ to denote the set of $K$ clients, where $\mathcal{K} \subset \mathcal{M}$. The remaining clients perform local training as shown in Algorithm 3. Algorithm 2 details the cross-client training process for client $m \in \mathcal{K}$. Rather than directly exchanging node features between clients, Swift-FedGNN partitions GNN training between the clients and the server. We offload[1] the aggregation of node features and intermediate activations at each GNN layer to the server and remote

---

[1]The operation offloading in Swift-FedGNN only supports element-wise (*e.g.*, mean, sum, max) operations, *e.g.*, GCN, GraphSAGE, GIN, and SGCN. For non-element-wise operations (*e.g.*, GAT), which are fundamentally not a good fit in any communication-efficient FGL algorithm design, see Appendix E for detailed discussion.

clients corresponding to node $v$, thus reducing the communication overhead and eliminating the need for graph data sharing. This procedure helps preserve data privacy because the clients are unaware of the locations of neighbor nodes, and the embeddings of these neighbor nodes are aggregated before being transmitted to the clients. Operations performed on the server and the remote clients are colored using `server` and `remote client` respectively.

Specifically, client $m \in \mathcal{K}$ samples a mini-batch of training nodes $\mathcal{B}_v^m$. Then, with the cooperation of the server, a subset of $L$-hop neighbors for the training nodes in $\mathcal{B}_v^m$ is sampled and represented as $\mathcal{S} = \left\{ \mathcal{S}^{(l)} \right\}_{l=0}^{L-1}$. The nodes $v \in \mathcal{B}_v^m$ are on client $m$, while for $v \in \mathcal{S}^{(l)}$ with $l < L$, the nodes may be on clients other than $m$, denoting the client storing $v$ as $m(v)$. The set $\bar{\mathcal{M}}(v)$ represents remote clients with respect to $m(v)$, i.e., $\bar{\mathcal{M}}(v) \subseteq \mathcal{M} \setminus \{m(v)\}$, where the sampled cross-client neighbors of the training node $v$ are located. Each remote client $\bar{m}(v) \in \bar{\mathcal{M}}(v)$ may contain multiple sampled neighbors of the training node $v$, and the numbers of the sampled neighbors can vary across clients.

Computing the $l$-th layer embedding of node $v$ consists of four steps. Steps 1 to 3 below are used to aggregate the neighbor information of node $v$, and Step 4 is used to update the node $v$'s embedding at $l$-th GNN layer.

**Step 1)** Each remote client $\bar{m}(v)$ aggregates its sampled neighbors of node $v$ in *parallel,* using

$$\boldsymbol{h}_{\mathcal{N}(v)}^{(l),\bar{m}(v)} = \text{AGG}\left( \left\{ \boldsymbol{h}_u^{(l-1),\bar{m}(v)} \mid u \in \widetilde{\mathcal{N}}^{\bar{m}(v)}(v) \right\} \right). \quad (5)$$

We send only the aggregated results from each remote client $\bar{m}(v)$ to the server, which can help preserve data privacy and reduce communication overhead.

**Step 2)** Upon receiving the aggregated neighbor information from all the remote clients $\bar{m}(v) \in \bar{\mathcal{M}}(v)$, the server aggregates this information from different remote clients before sending it to client $m(v)$ as follows:

$$\boldsymbol{r}_{\mathcal{N}(v)}^{(l)} = \text{AGG}\left( \left\{ \boldsymbol{h}_{\mathcal{N}(v)}^{(l),\bar{m}(v)} \mid \bar{m}(v) \in \bar{\mathcal{M}}(v) \right\} \right). \quad (6)$$

This approach not only helps maintain data privacy[2] but also reduces communication costs by minimizing the amount of data transmitted between clients and the server.

**Step 3)** Neighbor information of node $v$ for both the sampled local neighbors and the sampled cross-client neighbors is aggregated as follows:

$$\boldsymbol{h}_{\mathcal{N}(v)}^{(l)} = \text{AGG}\left( \underbrace{\left\{ \boldsymbol{h}_u^{(l-1),m(v)} \mid u \in \widetilde{\mathcal{N}}^{m(v)}(v) \right\}}_{\text{local}} \cup \underbrace{\left\{ \boldsymbol{r}_{\mathcal{N}(v)}^{(l)} \right\}}_{\text{remote}} \right). \quad (7)$$

The cross-client neighbor information used here helps mitigate the information loss and reduce the performance degradation caused by connected nodes being distributed across different clients.

**Step 4)** The embedding of node $v$ in the $l$-th GNN layer is updated using the aggregated neighbor information and the embedding of node $v$ from the $(l-1)$-th layer as:

$$\boldsymbol{h}_v^{(l),m(v)} = \sigma\left( \boldsymbol{W}_t^{(l),m(v)} \cdot \text{COMB}\left( \boldsymbol{h}_v^{(l-1),m(v)}, \boldsymbol{h}_{\mathcal{N}(v)}^{(l)} \right) \right). \quad (8)$$

Using the embeddings of the training nodes in the mini-batch and the model parameters, the local stochastic gradients $\nabla \widetilde{F}^m(\boldsymbol{\theta}_t^m)$ are computed and used in the update of the global model parameters shown as $\boldsymbol{\theta}_{t+1} = \boldsymbol{\theta}_t - \alpha \frac{1}{|\mathcal{M}|} \sum_{m \in \mathcal{M}} \nabla \widetilde{F}^m(\boldsymbol{\theta}_t^m)$, where $\alpha$ is the learning rate.

## 5 THEORETICAL PERFORMANCE ANALYSIS

In this section, we establish the theoretical convergence guarantees for Swift-FedGNN using Graph Convolutional Network (GCN)[3] Kipf & Welling (2017) as the GNN architecture to solve Problem (1). The analysis of GNN convergence is significantly more challenging compared to the existing literature on deep neural networks (DNNs). The key difficulties stem from the fact that, unlike in DNNs, the stochastic gradients in GNNs are inherently *biased*. This bias is primarily caused by the

---

[2]To further enhance privacy, Swift-FedGNN is compatible with differential privacy techniques and federated encryption protocols, enabling formal privacy guarantees. See Appendix F for a detailed discussion.

[3]These convergence guarantees also extend to other element-wise operation-based GNNs, *e.g.*, GraphSAGE and GIN. See Appendix I for guidance on extending the analysis.

presence of cross-client neighbors and the neighbor sampling process. The errors from missing or unsampled neighbors propagate across layers, gradually getting amplified from the input layer to the output layer, complicating the overall convergence behavior.

For a graph $\mathcal{G}$, the structure can be represented by its adjacency matrix $\boldsymbol{A} \in \mathbb{R}^{N \times N}$, where $\boldsymbol{A}_{vu} = 1$ if $(v, u) \in \mathcal{E}$, otherwise $\boldsymbol{A}_{vu} = 0$. The propagation matrix can be computed as $\boldsymbol{P} = \boldsymbol{D}^{-1/2} \hat{\boldsymbol{A}} \boldsymbol{D}^{-1/2}$, where $\hat{\boldsymbol{A}} = \boldsymbol{A} + \boldsymbol{I}$, and $\boldsymbol{D} \in \mathbb{R}^{N \times N}$ corresponds to the degree matrix and $\boldsymbol{D}_{vv} = \sum_u \hat{\boldsymbol{A}}_{vu}$. For subgraph $\mathcal{G}^m$ located on client $m$, the adjacency matrix $\boldsymbol{A}^m$ can be denoted as $\boldsymbol{A}^m = \boldsymbol{A}^m_{local} + \boldsymbol{A}^m_{remote}$, where $\boldsymbol{A}^m_{local}$ corresponds to the nodes located on client $m$, and $\boldsymbol{A}^m_{remote}$ corresponds to their cross-client neighbors located on the remote clients other than $m$. Then, the propagation matrix can be calculated as $\boldsymbol{P}^m = \boldsymbol{D}_m^{-1/2} (\boldsymbol{A}^m + \boldsymbol{I}^m) \boldsymbol{D}_m^{-1/2}$, and can be represented as $\boldsymbol{P}^m = \boldsymbol{P}^m_{local} + \boldsymbol{P}^m_{remote}$, where $\boldsymbol{P}^m_{local} = \boldsymbol{D}_m^{-1/2} (\boldsymbol{A}^m_{local} + \boldsymbol{I}^m) \boldsymbol{D}_m^{-1/2}$ and $\boldsymbol{P}^m_{remote} = \boldsymbol{D}_m^{-1/2} (\boldsymbol{A}^m_{remote}) \boldsymbol{D}_m^{-1/2}$.

Given GCN as the GNN architecture, for client $m$ training using only the local graph data, Eq. (3) and (4) are equivalent to $\widetilde{\boldsymbol{H}}_t^{(l),m} = \sigma\big(\widetilde{\boldsymbol{P}}_{local}^{(l),m} \widetilde{\boldsymbol{H}}_{local}^{(l-1),m} \boldsymbol{W}_t^{(l),m}\big)$. For client $m$ training based on both the local graph data and the cross-client neighbors, Eq. (5)–(8) are equivalent to $\widetilde{\boldsymbol{H}}_t^{(l),m} = \sigma\big((\widetilde{\boldsymbol{P}}_{local}^{(l),m} \widetilde{\boldsymbol{H}}_{local}^{(l-1),m} + \widetilde{\boldsymbol{P}}_{remote}^{(l),m} \widetilde{\boldsymbol{H}}_{remote}^{(l-1),m}) \boldsymbol{W}_t^{(l),m}\big)$.

Before proceeding with the convergence analysis, we make the following standard assumptions.

**Assumption 5.1.** The loss function $\ell^m (\cdot, \cdot)$ is $C_l$-Lipschitz continuous and $L_l$-smooth with respect to the node embedding $\boldsymbol{h}^{(L)}$, i.e., $\|\ell^m(\boldsymbol{h}_1^{(L)}, y) - \ell^m(\boldsymbol{h}_2^{(L)}, y)\|_2 \leq C_l \|\boldsymbol{h}_1^{(L)} - \boldsymbol{h}_2^{(L)}\|_2$ and $\|\nabla \ell^m(\boldsymbol{h}_1^{(L)}, y) - \nabla \ell^m(\boldsymbol{h}_2^{(L)}, y)\|_2 \leq L_l \|\boldsymbol{h}_1^{(L)} - \boldsymbol{h}_2^{(L)}\|_2$.

**Assumption 5.2.** The activation function $\sigma(\cdot)$ is $C_\sigma$-Lipschitz continuous and $L_\sigma$-smooth, i.e., $\|\sigma(\boldsymbol{z}_1^{(l)}) - \sigma(\boldsymbol{z}_2^{(l)})\|_2 \leq C_\sigma \|\boldsymbol{z}_1^{(l)} - \boldsymbol{z}_2^{(l)}\|_2$ and $\|\nabla \sigma(\boldsymbol{z}_1^{(l)}) - \nabla \sigma(\boldsymbol{z}_2^{(l)})\|_2 \leq L_\sigma \|\boldsymbol{z}_1^{(l)} - \boldsymbol{z}_2^{(l)}\|_2$.

**Assumption 5.3.** For any $l \in [L]$, the norm of weight matrices, the propagation matrix, and the node feature matrix are bounded by $B_W$, $B_P$ and $B_X$, respectively, i.e., $\|\boldsymbol{W}^{(l)}\|_F \leq B_W$, $\|\boldsymbol{P}\|_F \leq B_P$, and $\|\boldsymbol{X}\|_F \leq B_X$. Note that this assumption is commonly used in the analysis of GNNs, e.g., Chen et al. (2018); Liao et al. (2020); Garg et al. (2020); Cong et al. (2021); Wan et al. (2022).

Different from DNNs with unbiased stochastic gradients, the stochastic gradients in sampling-based GNNs are *biased* due to neighbor sampling of the training nodes. This is one of the **key challenges** in the convergence analysis of Swift-FedGNN. Some existing works used strong assumptions to deal with these biased stochastic gradients in their analysis, e.g., Chen et al. (2018) adopts the unbiased stochastic gradient assumption, and Chen & Luss (2018) uses the consistent stochastic gradient assumption. However, these assumptions may not hold in reality. In this paper, without using the aforementioned strong assumptions, we are able to bound the errors between the stochastic gradients and the full gradients in the following lemma.

**Lemma 5.4.** *Under Assumptions 5.1–5.3, the errors between the stochastic gradients and the full gradients are bounded as $\|\nabla F_{local}^m (\boldsymbol{\theta}^m) - \nabla \tilde{F}_{local}^m (\boldsymbol{\theta}^m)\|_F \leq L B_{\Delta G}^l$ and $\|\nabla F_{full}^m (\boldsymbol{\theta}^m) - \nabla \tilde{F}_{full}^m (\boldsymbol{\theta}^m)\|_F \leq L B_{\Delta G}^f$, where $\nabla F_{local}^m (\boldsymbol{\theta}^m)$ and $\nabla \tilde{F}_{local}^m (\boldsymbol{\theta}^m)$ correspond to the full and stochastic gradients computed with only local graph data, respectively. $\nabla F_{full}^m (\boldsymbol{\theta}^m)$ and $\nabla \widetilde{F}_{full}^m (\boldsymbol{\theta}^m)$ include both local graph data and cross-client neighbors of the training nodes. $B_{\Delta G}^l$ and $B_{\Delta G}^f$ are defined in Eq. (12) and (13) in Appendix H.*

Furthermore, the dependencies of the nodes located on different clients can lead to additional errors in the gradient computations when client $m$ is updated only with its local graph data, since the cross-client neighbors are missed. This becomes another **key challenge** in the analysis of the convergence of Swift-FedGNN. We prove that such an error is upper-bounded as shown in the following lemma.

**Lemma 5.5.** *Under Assumptions 5.1–5.3, the error between the full gradient computed with both the local graph data and the cross-client neighbors of the training nodes ($\nabla F_{full}^m (\boldsymbol{\theta}^m)$) and the full gradient computed with only the local graph data ($\nabla F_{local}^m (\boldsymbol{\theta}^m)$) is upper-bounded as $\|\nabla F_{full}^m (\boldsymbol{\theta}^m) - \nabla F_{local}^m (\boldsymbol{\theta}^m)\|_F \leq L B_{\Delta G}^r$, where $B_{\Delta G}^r$ is defined in Eq. (14) in Appendix H.*

We note that all the errors mentioned in Lemmas 5.4 and 5.5 are correlated with the structure of GNNs, specifically showing a positive correlation with the number of layers in the networks. This finding is unique to GNNs, where each layer involves both neighbor aggregation and non-linear

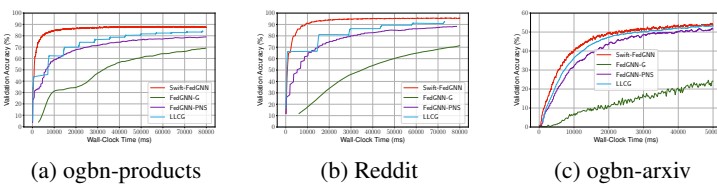

|(a) ogbn-products|(b) Reddit|(c) ogbn-arxiv|
|---|---|---|

Figure 4: Convergence performance in terms of validation accuracy of different algorithms.

Figure 5: Average communication cost per step.

transformation. As these two operations are interleaved across multiple layers, they create a structural entanglement that complicates the analysis.

Using Lemmas 5.4 and 5.5, we state the main convergence result of Swift-FedGNN solving an $L$-layer GNN in the following theorem:

**Theorem 5.6.** *Under Assumptions 5.1–5.3, choose step-size* $\alpha = \min\left\{\frac{\sqrt{M}}{\sqrt{T}}, \frac{1}{L_F}\right\}$, *where* $L_F$ *is the smoothness constant in Lemma H.2. The output of* Swift-FedGNN *solving an $L$-layer GNN satisfies:*

$$\frac{1}{T}\sum_{t=0}^{T-1}\|\nabla\mathcal{L}(\boldsymbol{\theta}_t)\|^2 \leq \frac{2\left(\mathcal{L}(\boldsymbol{\theta}_0)-\mathcal{L}(\boldsymbol{\theta}^*)\right)}{\sqrt{MT}} + L^2\left(B_{\Delta G}^l + B_{\Delta G}^r\right)^2 + \frac{KL^2}{IM}\left(\left(B_{\Delta G}^f\right)^2 - \left(B_{\Delta G}^l + B_{\Delta G}^r\right)^2\right).$$

The detailed proof of Theorem 5.6 can be found in Appendix H. We can see from Theorem 5.6 that the convergence rate of Swift-FedGNN is $\mathcal{O}\left(T^{-1/2}\right)$ to a neighborhood of the exact solution, which *matches* the SOTA convergence rate of sampling-based GNN algorithms, *e.g.*, Chen et al. (2018); Cong et al. (2021); Ramezani et al. (2022); Du & Wu (2022), even though Swift-FedGNN operates in the far more challenging federated setting.

Three important remarks on Theorem 5.6 are in order: (1) When choosing $I = 1$ and $K = M$, Swift-FedGNN performs fully cross-client training, ensuring no information loss in the graph data. In this scenario, Swift-FedGNN experiences minimal residual error. Such error is caused by sampling and is inevitable. However, Swift-FedGNN suffers from maximum sampling and communication overhead; (2) When choosing $K = 0$, Swift-FedGNN conducts fully local training, resulting in the information loss of all the cross-client neighbors. Consequently, Swift-FedGNN encounters maximum residual error. Nonetheless, the sampling and communication overhead is minimized; and (3) It can be shown that the last term of the convergence rate bound in Theorem 5.6 is negative. Hence, increasing $I$ or decreasing $K$ would increase the residual error due to more information loss of the cross-client neighbors. However, this would reduce the sampling and communication overhead. Thus, there is a trade-off between the information loss and the sampling and communication overhead. See Appendix G.1.1 for empirical evidence supporting our theoretical findings.

**Communication Complexity:** Assume each GNN layer uses a uniform neighbor sampling fan-out of $F$, with $F^l$ representing the worst-case number of neighbors sampled per training node at layer $l \in [1, L]$. Let $p_{(l)} \in (0, 1)$ be the fraction of neighbors at layer $l$ located on remote clients. If the $p_{(l)}F^l$ cross-client neighbors at layer $l$ are distributed across $C_{(l)} < M$ remote clients, then the total communication cost per cross-client training round in Swift-FedGNN for exchanging the aggregated cross-client neighbor embeddings is $\mathcal{O}\left(KB\sum_{l=1}^{L} C_{(l)}d_{(l-1)}^{emb}\right)$, where $B$ is the batch size per client, and $d_{(l)}^{\mathrm{emb}}$ is the embedding dimension at layer $l$. Since $C_{(l)} \ll p_{(l)}F^l$ due to Swift-FedGNN's aggregation mechanism, this result highlights the communication efficiency of Swift-FedGNN. For further discussion on both communication and computation complexity, see Appendix D.

## 6 NUMERICAL RESULTS

In this section, we conduct experiments to evaluate the performance of Swift-FedGNN. Due to space limitations, additional experimental details and results are provided in Appendix G.

**1) Experiment Settings:** We train a representative GNN model, GraphSAGE Hamilton et al. (2017), in the FL settings on five real-world node classification datasets: 1) ogbn-products Hu et al. (2020); 2) Reddit Hamilton et al. (2017); 3) ogbn-arxiv Hu et al. (2020); 4) flickr Zeng et al. (2020); and 5) citeseer Giles et al. (1998). The key statistics of the datasets are summarized in Table 14 in Appendix G. Note that ogbn-products dataset is the *largest* dataset one can find in the FGL literature,

Table 1: Total communication cost (GB) when achieving a target validation accuracy for each dataset.

| | OGBN-PRODUCTS | REDDIT | OGBN-ARXIV | FLICKR | CITESEER |
|---|---|---|---|---|---|
| **SWIFT-FEDGNN** | **0.66** | **5.89** | **0.95** | **1.07** | **35.32** |
| FEDGNN-G | 8.40 | 70.72 | 7.91 | 17.22 | 43.43 |
| LLCG | 4.47 | 23.89 | 1.46 | 1.30 | 25.80 |
| FEDGNN-PNS | 0.97 | 8.96 | 1.41 | 1.27 | 36.11 |

while the Reddit dataset is known for its *density*. In our FL simulations, we use 20 clients for the experiments with ogbn-products dataset and 10 clients for the experiments with the other datasets. All graphs are partitioned with METIS algorithm Karypis & Kumar (1998). In addition, we evaluate Swift-FedGNN on randomly partitioned graph data and on another widely used GNN model, GIN Xu et al. (2019). The corresponding results are provided in Appendix G.

**2) Baselines:** Since the goal of Swift-FedGNN is to reduce the sampling and communication time, we compare Swift-FedGNN with the algorithms most closely related to Swift-FedGNN, which mitigates the information loss of cross-client neighbors through periodical (sampling-based) full-neighbor training: **1)** LLCG Ramezani et al. (2022): A DGL framework that performs local training on each client independently, with periodic full-neighbor training conducted on a central server; **2)** FedGNN-PNS Du & Wu (2022): A FGL framework where each client periodically samples cross-client neighbors with an increasing sampling frequency. In the remaining iterations, clients reuse the most recently sampled cross-client neighbors; and **3)** FedGNN-G: A naive FGL algorithm where cross-client training is performed on each client in every iteration.

**3) Convergence Performance Comparisons:** In Figures 4a and 4b, we can see that for both the ogbn-products dataset and the Reddit dataset, Swift-FedGNN achieves substantially faster convergence than all baseline algorithms, which verifies the effectiveness of Swift-FedGNN in handling large or dense graphs. In addition, despite less frequent cross-client training, the validation accuracy of Swift-FedGNN is comparable to that of FedGNN-G, which trains a GNN model on the dataset without any information loss. Although LLCG performs periodic cross-client training on the server, it requires training over the full set of neighbors of the training nodes, leading to significant sampling and communication overhead. For instance, when training the ogbn-products dataset, LLCG takes over 5000 ms to perform cross-client training on the server, whereas Swift-FedGNN completes cross-client training within 200 ms due to neighbor sampling. FedGNN-PNS employs a dynamic cross-client sampling interval throughout training, gradually reducing the interval as training progresses. Consequently, FedGNN-PNS incurs extensive sampling and communication overhead during the later stages of training, slowing down the convergence process. As shown in Figure 4c, on the smaller ogbn-arxiv dataset the benefit of Swift-FedGNN is less pronounced. The dataset's limited size and sparsity reduce both neighbor sampling and communication overhead for all methods, narrowing the performance gap. This is also reflected in the following communication cost analysis. Nevertheless, Swift-FedGNN still delivers the best overall performance.

**4) Communication Cost Analysis:** Figure 5 shows the average communication cost per step for Swift-FedGNN and the baselines across five datasets, demonstrating that Swift-FedGNN consistently incurs the lowest communication cost on all of them. Specifically, our algorithm Swift-FedGNN incurs a communication cost that is $7\times$ to $21\times$ lower than that of FedGNN-G on four out of the five datasets (Reddit, ogbn-products, ogbn-arxiv, and flickr). On the smallest graph, citeseer, the gap narrows because the size of the cross-client neighbor information becomes negligible compared with the model size, yet Swift-FedGNN still maintains the lowest communication cost. For the largest dataset ogbn-products and the most dense dataset Reddit, Swift-FedGNN achieves communication costs that are $2\times$ and $5\times$ lower compared to FedGNN-PNS and LLCG, respectively. On the small datasets, ogbn-arxiv and flickr, the communication cost advantage of Swift-FedGNN remains evident, though closer to approximately $1\times$ lower than FedGNN-PNS and LLCG. These findings validate the superior communication efficiency of our proposed Swift-FedGNN algorithm.

Table 1 reports the total communication cost required to reach the same target validation accuracy on each dataset. The results demonstrate that our proposed Swift-FedGNN algorithm consistently incurs the lowest communication cost across all datasets except Citeseer. For example, to reach a target accuracy of 87% on the ogbn-products dataset, Swift-FedGNN achieves at least a 31.9% reduction in total communication cost compared to all baselines. Similarly, to reach a target accuracy of 55% on the smaller and sparser ogbn-arxiv dataset, Swift-FedGNN still delivers at least 32.2% communication savings, highlighting its robustness and efficiency across diverse graph structures.

Table 2: Validation accuracy of Swift-FedGNN with different correction frequencies ($I$) and client sampling sizes ($K$) on the ogbn-products dataset.

| # OF SAMPLED CLIENTS ($K$) | 10 | | | | 1 | 5 | 10 | 15 |
|---|---|---|---|---|---|---|---|---|
| CORRECTION FREQUENCY ($I$) | 5 | 10 | 20 | 40 | | | 10 | |
| VALIDATION ACCURACY (%) | 88.91 | 88.88 | 88.60 | 88.44 | 88.47 | 88.72 | 88.88 | 89.22 |

Table 3: Total communication cost (MB) of Swift-FedGNN with different $I$ and $K$ on the ogbn-products dataset when achieving a target validation accuracy of 87%.

| # OF SAMPLED CLIENTS ($K$) | 10 | | | | 1 | 5 | 10 | 15 |
|---|---|---|---|---|---|---|---|---|
| CORRECTION FREQUENCY ($I$) | 5 | 10 | 20 | 40 | | | 10 | |
| COMMUNICATION COST (MB) | 1344.0 | 675.5 | 324.5 | 275.0 | 57.8 | 342.2 | 675.5 | 1027.4 |

Table 4: Total communication cost (GB) on the ogbn-products dataset for two large-scale settings with 80 clients and 100 clients when achieving a target validation accuracy for each setting.

| | SWIFT-FEDGNN (FIXED $K = 10$) | SWIFT-FEDGNN (FIXED $K/M = 1/2$) | FEDGNN-G | LLCG | FEDGNN-PNS |
|---|---|---|---|---|---|
| $M = 80$ | 0.69 | 2.42 | 37.26 | 3.90 | 4.49 |
| $M = 100$ | 1.17 | 5.24 | 59.00 | 6.82 | 8.67 |

**5) Hyperparameter sensitivity analysis:** We explore the impact of different choices for key parameters in Swift-FedGNN (*i.e.*, the correction frequency $I$ and the client sampling size $K$) in Swift-FedGNN. Table 2 report the validation accuracy of Swift-FedGNN on ogbn-products dataset under various $I$ and $K$. The results show that: i) Increasing $I$ from 5 to 40 leads to only a minor accuracy degradation (0.47%), demonstrating that less frequent cross-client training still preserves model quality; and ii) Decreasing $K$ from 15 to 1 also results in a minor accuracy drop (0.75%), indicating that a small number of sampled clients is sufficient to maintain strong performance. These findings are consistent with our theoretical conclusion in Remark (3) of Theorem 5.6. Complementary to the these findings, Table 3 presents the total communication cost needed to achieve a target accuracy of 87% under the same parameter variations. These results show that increasing $I$ and decreasing $K$ substantially reduce communication cost. For example, increasing $I$ from 5 to 40 saves approximately 80% of the communication overhead, while reducing $K$ from 10 to 1 saves approximately 94%. Collectively, these results validate Swift-FedGNN's ability to reduce communication without incurring major information loss and demonstrate that Swift-FedGNN provides a tunable balance between communication efficiency and accuracy, and the trade-off can be controlled via $I$ and $K$.

**6) Evaluations of large-scale settings:** To evaluate the scalability of Swift-FedGNN, we extend our experiments to two large-scale settings with 80 clients and 100 clients on the ogbn-products dataset. Table 4 reports the total communication cost when achieving a target validation accuracy (*i.e.*, 83% for the 80-client setting and 84.3% for the 100-client setting). These results show that: i) With fixed $K = 10$, Swift-FedGNN reduces total communication cost by at least 82% in the 80-client setting and at least 83% in the 100-client setting compared to all baselines; and ii) with a client sampling ratio of $K/M = 50\%$, Swift-FedGNN still achieves at least 38% communication savings in the 80-client setting and at least 23% communication savings in the 100-client setting over all baselines. These findings highlight Swift-FedGNN's effectiveness and scalability, validating its communication efficiency even in large-scale FGL settings. Moreover, server-side aggregation in Swift-FedGNN will not be a bottleneck in large-scale settings as long as $K$ is adjusted appropriately (*e.g.*, $K \ll M$).

## 7 CONCLUSION

In this paper, we proposed the Swift-FedGNN algorithm, which is a mini-batch-based and sampling-based federated graph learning framework, for efficient federated GNN training. Swift-FedGNN reduces the cross-client neighbor sampling and communication overhead by *periodically* sampling a set of clients to conduct the local GNN training on local graph data and cross-client neighbors, which is time-consuming. The rest clients in these periodical iterations and all the clients in the remaining iterations perform efficient parallel local GNN training using only local graph data. We theoretically proved that the convergence rate of Swift-FedGNN is $\mathcal{O}\left(T^{-1/2}\right)$, matching the SOTA rate of sampling-based GNN methods, even in more challenging federated settings. We conducted extensive numerical experiments on real-world graph datasets and verified the effectiveness of Swift-FedGNN.

## ETHICS STATEMENT

We confirm that the ICLR Code of Ethics has been thoroughly reviewed and that this work fully adheres to it. The study does not involve human subjects, sensitive data, or any foreseeable risks, and it raises no ethical, legal, or conflict-of-interest concerns.

## REPRODUCIBILITY STATEMENT

We confirm the reproducibility of this work. Specifically, for the theoretical results, we state the assumptions in Section 5 and provide detailed proofs in Appendix H. For the experimental results, we submit the source code as a supplementary material and describe the implementation details in Appendix G.

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

## A  THE USE OF LARGE LANGUAGE MODELS (LLMS)

LLMs were used exclusively for grammar correction and language polishing during the writing process. They did not contribute to research ideation or any substantive aspects of the work.

## B  LIST OF NOTATIONS

| | |
|---|---|
| $\mathcal{G}\left(\mathcal{V}, \mathcal{E}\right)$ | Graph |
| $\mathcal{V}$ | Set of nodes |
| $\mathcal{E}$ | Set of edges |
| $N = \|\mathcal{V}\|$ | Number of nodes |
| $\mathcal{M}$ | Set of clients |
| $M = \|\mathcal{M}\|$ | Number of clients |
| $\mathcal{G}^m\left(\mathcal{V}^m, \mathcal{E}^m\right)$ | Subgraph at client $m$ |
| $\mathcal{V}^m$ | Set of nodes at client $m$ |
| $\mathcal{E}^m$ | Set of edges at client $m$ |
| $\boldsymbol{x}_v^m \in \mathbb{R}^d$ | Feature vector of node $v$ at client $m$ |
| $y_v^m$ | Label of node $v$ at client $m$ |
| $\ell^m$ | Loss function (*e.g.*, cross-entropy loss) at client $m$ |
| $\mathcal{V}_B^m$ | Mini-batch of training nodes |
| $\boldsymbol{\theta} = \left\{\boldsymbol{W}^{(l)}\right\}_{l=1}^L$ | Set of trainable model parameters |
| $m(v)$ | Local client of node $v$ |
| $\bar{m}(v)$ | Remote client of node $v$ |
| $\bar{\mathcal{M}}(v)$ | Set of the remote clients that host the neighbors of node $v$ |
| $\mathcal{N}^{m(v)}(v)$ | Set of the neighbors of node $v$ located on local client $m(v)$ |
| $\mathcal{N}^{\bar{m}(v)}(v)$ | Set of the neighbors of node $v$ located on remote client $\bar{m}(v)$ |
| $\boldsymbol{h}_v^{(l),m(v)}$ | Embedding of node $v$ located on client $m(v)$ |
| $\boldsymbol{h}_{\mathcal{N}(v)}^{(l)}$ | Aggregated embedding from node $v$'s neighbors |
| $\boldsymbol{W}^{(l)}$ | Weight matrix at $l$-th layer |
| $\sigma\left(\cdot\right)$ | Activation function (*e.g.*, ReLU) |
| $\mathrm{AGG}\left(\cdot\right)$ | Aggregation function (*e.g.*, mean) |
| $\mathrm{COMB}\left(\cdot\right)$ | Combination function (*e.g.*, concatenation) |
| $\mathcal{B}_v^m$ | Mini-batch of training nodes at client $m$ |
| $\mathcal{S} = \left\{\mathcal{S}^{(l)}\right\}_{l=0}^{L-1}$ | Subset of $L$-hop neighbors for the training nodes in $\mathcal{B}_v^m$ |
| $\widetilde{\mathcal{N}}^m\left(v\right)$ | Set of the sampled neighbors located on client $m$ for node $v$ |
| $\mathcal{K}$ | Set of sampled clients for cross-client training |
| $K = \|\mathcal{K}\|$ | Number of sampled clients for cross-client training |
| $\nabla\widetilde{F}^m\left(\boldsymbol{\theta}_t^m\right)$ | Stochastic gradient |
| $\alpha$ | Learning rate |
| $\boldsymbol{A} \in \mathbb{R}^{N \times N}$ | Adjacency matrix of graph $\mathcal{G}$ |
| $\boldsymbol{P}$ | Propagation matrix |
| $\boldsymbol{D}$ | Degree matrix |

| | |
|---|---|
| $\boldsymbol{A}^m$ | Adjacency matrix of subgraph $\mathcal{G}^m$ |
| $\boldsymbol{A}_{local}^m$ | Adjacency matrix corresponds to the nodes located on client $m$ |
| $\boldsymbol{A}_{remote}^m$ | Adjacency matrix corresponds to the cross-client neighbors located on the remote clients other than $m$ |
| $\boldsymbol{D}^m$ | Degree matrix of client $m$ |
| $\boldsymbol{P}^m$ | Propagation matrix of client $m$ |
| $\boldsymbol{P}_{local}^m$ | Propagation matrix corresponds to the nodes located on client $m$ |
| $\boldsymbol{P}_{remote}^m$ | Propagation matrix corresponds to the cross-client neighbors located on the remote clients other than $m$ |

## C  SINGLE-MACHINE GRAPH NEURAL NETWORKS TRAINING

We consider a graph $\mathcal{G}\left(\mathcal{V}, \mathcal{E}\right)$, where $\mathcal{V}$ is a set of nodes with $N = |\mathcal{V}|$ and $\mathcal{E}$ is a set of edges. Each node $v \in \mathcal{V}$ is associated with a feature vector $\boldsymbol{x}_v \in \mathbb{R}^d$, where $d$ is the dimension of the feature vector. Each node $v \in \mathcal{V}_{train}$ has a corresponding label $y_v$, where $\mathcal{V}_{train} \subseteq \mathcal{V}$.

GNNs aim to generate representations (embeddings) for each node in the graph by combining information from its neighboring nodes. Consider a GNN that consists of $L$ layers. The embedding of node $v$ at $l$-th layer, which is represented by $\boldsymbol{h}_v^{(l)}$, can be obtained through neighbor aggregation and node update, which are formulated as follows:

$$\boldsymbol{h}_{\mathcal{N}(v)}^{(l)} = \mathrm{AGG}\left(\left\{\boldsymbol{h}_u^{(l-1)} \mid u \in \mathcal{N}\left(v\right)\right\}\right), \quad \boldsymbol{h}_v^{(l)} = \sigma\left(\boldsymbol{W}^{(l)} \cdot \mathrm{COMB}\left(\boldsymbol{h}_v^{(l-1)}, \boldsymbol{h}_{\mathcal{N}(v)}^{(l)}\right)\right),$$

where $\boldsymbol{h}_v^{(0)}$ is initialized as the feature vector $\boldsymbol{x}_v$, $\mathcal{N}\left(v\right)$ denotes the set of neighbors of node $v$, $\boldsymbol{h}_{\mathcal{N}(v)}^{(l)}$ is the aggregated embedding from node $v$'s neighbors aggregated neighbor embedding for node $v$, $\boldsymbol{W}^{(l)}$ represents the weight matrix at $l$-th layer, $\sigma\left(\cdot\right)$ corresponds to an activation function (*e.g.*, ReLU), $\mathrm{AGG}\left(\cdot\right)$ is an aggregation function (*e.g.*, mean), and $\mathrm{COMB}\left(\cdot\right)$ is a combination function (*e.g.*, concatenation).

## D  COMMUNICATION AND COMPUTATION COMPLEXITY OF Swift-FedGNN

In this section, we provide asymptotic characterizations of the communication and computation complexities of our proposed Swift-FedGNN algorithm. Due to the complications in precisely analyzing the communication and computation costs, we provide a high-level asymptotic analysis based on the several key system parameters.

Throughout the analysis, we assume an $L$-layer GNN and the following parameters:

- $M$: The total number of clients;
- $F$: The same number of neighbor sampling fan-out used at each layer;
- $F^l$: The worst-case number of neighbors at each training node at each GNN layer $l \in [1, L]$ using $F$-fan-out;
- $p_{(l)} \in (0, 1)$: The fraction of the neighbors that are located on other clients.

### D.1  COMMUNICATION COMPLEXITY OF Swift-FedGNN

**1) Communication cost per iteration for exchanging cross-client neighbor information:** In Swift-FedGNN, every $I$ iterations, each of the $K$ sampled clients performs cross-client training and exchanges aggregated embeddings for its cross-client neighbors. The total communication cost per cross-client training iteration for exchanging these embeddings is on the order of:

$$\mathcal{O}\left(KB \sum_{l=1}^{L} p_{(l)} F^l d_{(l-1)}^{emb}\right),$$

where $B$ is the batch size per client, $d_{(l)}^{\mathrm{emb}}$ is the embedding (hidden) dimension at layer $l$, and $F^l$ reflects the exponential expansion in sampled neighborhoods as the layer depth increases.

Note that this estimate does not account for the two-stage aggregation in Swift-FedGNN, which will significantly reduce the size of transferred embeddings. Therefore, this expression only represents a conservative (worst-case) upper bound, and the actual communication overhead is likely to be much lower.

If the $p_{(l)}F^l$ cross-client neighbors at layer $l$ are distributed across $C_{(l)} < M$ remote clients, then after aggregation, the communication cost becomes:

$$\mathcal{O}\left(KB\sum_{l=1}^{L}C_{(l)}d_{(l-1)}^{emb}\right),$$

where $C_{(l)} \ll p_{(l)}F^l$ due to the aggregation mechanism in Swift-FedGNN.

For comparison, consider FedGNN-PNS Du & Wu (2022), the most closely related prior work, which reduces communication by reusing the same sampled training nodes and their sampled neighbors across multiple training iterations, but directly transmits raw input features for those cross-client neighbors. The total communication cost of FedGNN-PNS per cross-client neighbor update is approximately:

$$\mathcal{O}\left(MB\sum_{l=1}^{L}p_{(l)}F^l d_{(0)}^{emb}\right),$$

where $d_{(0)}^{emb}$ is the input feature dimension, typically larger than hidden dimensions in deeper layers.

When communication of cross-client neighbors occurs, Swift-FedGNN is more efficient than FedGNN-PNS due to three key reasons: i) It involves only $K < M$ clients per iteration; ii) It transmits lower-dimensional hidden embeddings (*i.e.*, $d_{(l-1)}^{emb} < d_{(0)}^{emb}$ for $l \geq 2$); and iii) It leverages two-stage aggregation to compress information prior to transmission (*i.e.*, $C_{(l)} \ll p_{(l)}F^l$).

Moreover, as training progresses, FedGNN-PNS increases the frequency of graph data communication, which can lead to significant cumulative overhead. In contrast, Swift-FedGNN maintains a fixed periodic communication schedule and reduces transferred data per iteration, resulting in substantially lower overall communication cost.

**2) Total communication cost over $T$ training iterations:** Given the per-iteration communication cost for exchanging cross-client neighbor embeddings, the total communication cost of Swift-FedGNN for exchanging these embeddings across $T$ training iterations is on the order of:

$$\mathcal{O}\left(\frac{T}{I}KB\sum_{l=1}^{L}C_{(l)}d_{(l-1)}^{emb}\right).$$

In addition, gradients and global model parameters are transmitted in every iteration. Let the model parameters at layer $l$ be $W_{(l)} \in \mathbb{R}^{d_{(l-1)}^{emb} \times d_{(l)}^{emb}}$. Then the total communication cost for gradients and model updates is on the order of:

$$\mathcal{O}\left(2TM\sum_{l=1}^{L}d_{(l-1)}^{emb}d_{(l)}^{emb}\right).$$

Combining both, the overall communication complexity of Swift-FedGNN is on the order of:

$$\mathcal{O}\left(2TM\sum_{l=1}^{L}d_{(l-1)}^{emb}d_{(l)}^{emb} + \frac{T}{I}KB\sum_{l=1}^{L}C_{(l)}d_{(l-1)}^{emb}\right).$$

### D.2 COMPUTATION COMPLEXITY OF Swift-FedGNN

At each GNN layer $l$, the per-node computational cost includes:

- Neighbor aggregation (*e.g.*, mean/sum/max): $\mathcal{O}(Fd_{(l-1)}^{emb})$.

- Linear transformation: $\mathcal{O}(d_{(l-1)}^{emb}d_{(l)}^{emb})$.

With a total of $F^{l-1}$ sampled nodes at layer $l$ (due to recursive fan-out), the total per-batch cost per client is on the order of:

$$\mathcal{O}\left( BF^{l-1}\left( Fd_{(l-1)}^{emb} + d_{(l-1)}^{emb}d_{(l)}^{emb}\right)\right).$$

Across $T$ training iterations and $M$ clients, the total computation complexity (including both forward and backward passes) of Swift-FedGNN can be expressed as:

$$\mathcal{O}\left( 2TM\sum_{l=1}^{L} BF^{l-1}\left( Fd_{(l-1)}^{emb} + d_{(l-1)}^{emb}d_{(l)}^{emb}\right)\right).$$

## E   DISCUSSION ON NON-ELEMENT-WISE OPERATIONS

In the design of our communication-efficient Swift-FedGNN, we do not consider non-element-wise operations (*e.g.*, GAT Veličković et al. (2017)), as such operations are fundamentally not a good fit in any communication-efficient FGL algorithm design.

Taking GAT as an example, GAT requires direct access to raw neighbor features/embeddings to compute attention weights based on nonlinear pairwise interactions (see Eq. (1) in Veličković et al. (2018)). This requirement necessitates transmitting raw neighbor features/embeddings across clients, which leads to significantly high communication overhead. In other words, it is impossible for GAT to leverage the same communication-efficient aggregated transmissions as in those GNN models based on element-wise operations (*e.g.*, GCN Kipf & Welling (2017), GraphSAGE Hamilton et al. (2017), GIN Xu et al. (2019), and SGCN Wu et al. (2019)). As a result, GAT is not an ideal GNN model choice in those FGL algorithm design settings, where communication efficiency is of utmost importance.

To further quantitatively understand GAT's communication efficiency limitation in FGL algorithm design, we analyze the communication cost of incorporating GAT into Swift-FedGNN (denoted as GAT-Swift-FedGNN) and compare it with our original Swift-FedGNN design. Throughout the analysis, we assume an $L$-layer GNN and the following parameters:

- $M$: The total number of clients;
- $F$: The same number of neighbor sampling fan-out used at each layer;
- $F^l$: The worst-case number of neighbors at each training node at each GNN layer $l \in [1, L]$ using $F$-fan-out;
- $p_{(l)} \in (0, 1)$: The fraction of the neighbors that are located on other clients.

**1) GAT-Swift-FedGNN:** Every $I$ iterations, each of the $K$ sampled clients performs cross-client training and exchanges raw features/embeddings for its cross-client neighbors. The total communication cost per cross-client training round for exchanging these embeddings is on the order of:

$$\mathcal{O}\left( KB\sum_{l=1}^{L} p_{(l)}F^l d_{(l-1)}^{emb}\right),$$

where $B$ is the batch size per client, $d_{(l)}^{emb}$ is the embedding (hidden) dimension at layer $l$, and $F^l$ reflects the exponential expansion in sampled neighborhoods as the layer depth increases.

**2) Swift-FedGNN:** In contrast, Swift-FedGNN avoids transferring raw features/embeddings by sharing aggregated neighbor features/embeddings. If the $p_{(l)}F^l$ cross-client neighbors at layer $l$ are distributed across $C_{(l)} < M$ remote clients, then after aggregation, the communication cost is on the order of:

$$\mathcal{O}\left( KB\sum_{l=1}^{L} C_{(l)} d_{(l-1)}^{emb}\right).$$

Since $C_{(l)} \ll p_{(l)}F^l$, Swift-FedGNN achieves significantly lower communication overhead than the GAT variant.

From the above analysis, we can see that GAT-Swift-FedGNN incurs a communication cost that is not only $F^l$ times higher than Swift-FedGNN, but the gap between them also grows exponentially as the number of layers increases.

Although non-element-wise operations (*e.g.*, GAT) are highly popular GNN models in both the literature and practice, how to reduce the communication cost and avoid raw neighbor feature transmission for non-element-wise operations in the federated setting is a fundamentally hard problem, which would require major architectural design changes in non-element-wise operations rather than straightforward adaptation. Therefore, exploring attention-based extensions is a valuable direction for future research.

## F    DISCUSSION ON PRIVACY IN Swift-FedGNN

In this work, our primary privacy motivation is to avoid the direct transmission of raw node features, which are often privacy-sensitive in real-world graph applications (*e.g.*, user attributes in social networks). Our "aggregate-then-transfer" design ensures that: i) Only aggregated neighbor embeddings (not raw features) are shared across clients; and ii) No raw node information is directly exposed to other clients or the server.

That said, we do not claim formal privacy guarantees (*e.g.*, differential privacy bounds) in this work, since simply using aggregation without Gaussian/Laplacian-type noise injection is unlikely to offer $(\epsilon, \delta)$-type differential privacy guarantee. Instead, our focus is on reducing communication overhead in federated graph learning while improving practical privacy-preserving behavior through communication-efficient design.

Importantly, the Swift-FedGNN framework is compatible with standard differential privacy techniques and federated encryption protocols, which can be integrated Gaussian/Laplacian-type noise injection to provide formal privacy guarantees.

## G    ADDITIONAL EXPERIMENTAL DETAILS AND RESULTS

### G.1    ADDITIONAL EXPERIMENTAL RESULTS

#### G.1.1    EXPERIMENTAL SUPPORT FOR THEORETICAL FINDINGS

Table 5: Gradient bias under varying GNN depths on the ogbn-arxiv dataset.

| GNN DEPTH | 2 | 14 | 16 |
|---|---|---|---|
| GRADIENT BIAS | 0.46 | 17.56 | 30.11 |

Table 6: Validation accuracy (%) under varying GNN depths on the ogbn-arxiv dataset.

| GNN DEPTH | 2 | 14 | 16 |
|---|---|---|---|
| VALIDATION ACCURACY (%) | 57.17 | 54.60 | 48.46 |

To empirically validate our theoretical findings, we use the gradient bias between the full gradient and the stochastic gradient as an empirical proxy. This quantity has a theoretical upper bound of $LB_{\Delta G}^l$ (see Lemma 5.4), making it a suitable example for analysis.

Table 5 presents the measured gradient bias on the ogbn-arxiv dataset across different GNN depths. These results clearly show that the gradient bias increases with the GNN depth, consistent with our theoretical result that deeper GNNs incur larger bias due to amplified sampling and cross-client neighbor errors.

Table 6 shows the validation accuracy on the ogbn-arxiv dataset under varying GNN depths. We observe that the validation accuracy degrades as the GNN depth increases. This behavior is consistent with our theoretical insight that deeper GNNs introduce larger gradient bias terms, which in turn lead to greater approximation error and reduced performance.

In summary, the empirical trends above corroborate the theoretical predictions in Theorem 5.6, confirming both the validity and practical relevance of the error bounds in Theorem 5.6.

It is worth emphasizing that the performance degradation observed with increasing GNN depth is a fundamental limitation of GNN architectures themselves, rather than a limitation our Swift-FedGNN algorithm design. This phenomenon is well-known to occur across GNNs regardless of the specific graph learning algorithm in use. Addressing it typically requires architecture-level enhancements, and many existing solutions (*e.g.*, Chen et al. (2022)) are fully compatible with our Swift-FedGNN design and can be integrated to mitigate depth-related degradation in practice.

### G.1.2 COMMUNICATION OVERHEAD

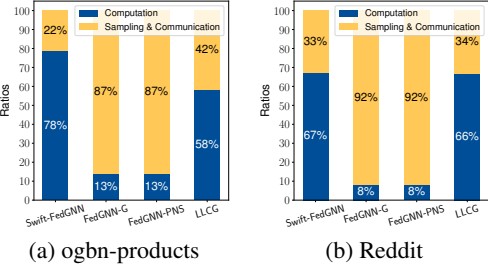

(a) ogbn-products                     (b) Reddit

Figure 6: Ratio of computation time to sampling and communication time for different algorithms.

**Communication and sample costs analysis:** Figure 6 illustrates the comparison between the ratios of the computation time and the sampling and communication time for Swift-FedGNN and the baseline algorithms. It can be seen that Swift-FedGNN significantly reduces the computation-(sampling & communication) ratio on the ogbn-products dataset. On the Reddit dataset, Swift-FedGNN also significantly reduces this ratio compared to FedGNN-PNS and FedGNN-G. While Swift-FedGNN achieves a comparable ratio to LLCG, it converges much faster and achieves higher validation accuracy than LLCG.

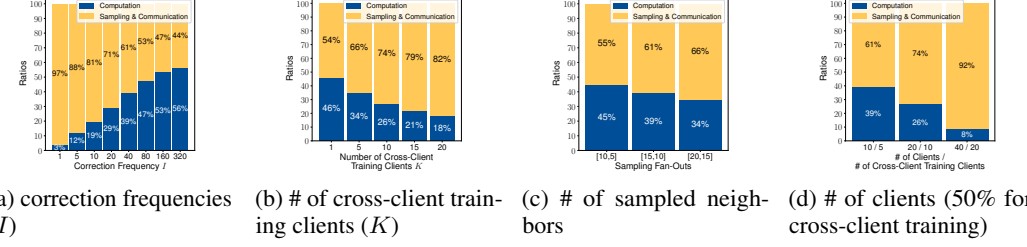

(a) correction frequencies ($I$)   (b) # of cross-client training clients ($K$)   (c) # of sampled neighbors   (d) # of clients (50% for cross-client training)

Figure 7: Ratio of computation time to sampling and communication time for Swift-FedGNN on ogbn-products dataset.

**Hyperparameter sensitivity analysis:** We explore the impact of the important hyperparameters in Swift-FedGNN. Figure 7a shows that when the correction frequency $I$ increases, the computation-(sampling & communication) ratio increases. Figure 7b and 7c indicate that as the number of cross-client training clients $K$, and the number of sampled neighbors increase, the computation-(sampling & communication) ratio decreases. Figure 7d evaluates Swift-FedGNN with different numbers of clients. In this experiment, 50% of clients periodically conduct cross-client training on both local and cross-client neighbors. We can see that as the number of clients increases, the computation-(sampling & communication) ratio decreases. These findings align with our expectations since sampling and communication overhead is significantly greater than computation overhead in GNN training.

Table 7: Communication overhead per iteration when communication occurs.

|  | Swift-FedGNN | LLCG | FedGNN-PNS | FedGNN-G |
|---|---|---|---|---|
| OGBN-PRODUCTS | 19.5 MB | 378.3 MB | 78.0 MB | 78.0 MB |
| REDDIT | 90.4 MB | 619.6 MB | 180.7 MB | 180.7 MB |

**Communication overhead when communication occurs:** Table 7 shows the communication overhead per iteration when cross-client sampling and communication occur for different algorithms. We can see that Swift-FedGNN significantly reduces the communication overhead compared to all baselines across both datasets. Specifically, on the ogbn-products dataset, Swift-FedGNN incurs 19.5

MB of overhead per iteration, which is approximately 20 times less than LLCG and 4 times less than both FedGNN-PNS and FedGNN-G. Similarly, for the Reddit dataset, due to its dense inter-node connections and larger feature size, Swift-FedGNN's overhead is 90.4 MB, which is still about 7 times less than LLCG and 2 times less than both FedGNN-PNS and FedGNN-G. This highlights the efficiency of Swift-FedGNN in reducing communication costs during cross-client training.

### G.1.3 VALIDATION ACCURACY

Table 8: Validation accuracy (%) of different algorithms using the GraphSAGE model.

|  | OGBN-PRODUCTS | REDDIT | OGBN-ARXIV | FLICKR | CITESEER |
|---|---|---|---|---|---|
| **SWIFT-FEDGNN** | **88.88** | **95.47** | **57.17** | **50.19** | **66.00** |
| LLCG | 87.66 | 95.27 | 56.78 | 50.12 | 68.40 |
| FEDGNN-PNS | 87.89 | 95.46 | 55.86 | 51.47 | 66.27 |
| FEDSAGE | 88.15 | 95.30 | 56.55 | 49.75 | 64.39 |
| FEDGNN-G | 88.71 | 95.96 | 56.78 | 51.57 | 66.08 |

**Validation accuracy comparisons:** Table 8 shows the validation accuracy of different algorithms. To assess the impact of cross-client neighbors, we include an additional baseline FedSage Zhang et al. (2021), an FGL algorithm that entirely ignores cross-client neighbors and performs purely local training in all iterations. The results demonstrate that despite incurring lower sampling and communication overhead, our Swift-FedGNN achieves validation accuracy comparable to that of the baseline algorithms. Moreover, compared to FedSage, which completely ignores cross-client neighbors, Swift-FedGNN achieves a higher validation accuracy, highlighting the importance of incorporating cross-client neighbor information. By minimizing sampling and communication overhead, Swift-FedGNN offers the highest efficiency in practical implementation.

It is worth noting that ogbn-arxiv, flickr, and citeseer are small datasets (Table 14), where graph partitioning leads to greater information loss. As a result, baselines that frequently exchange graph data can achieve slightly higher accuracy. However, these small datasets do not require federated graph learning in practice. Federated graph learning is primarily motivated by large-scale datasets like ogbn-products, where our method achieves the best performance.

Table 9: Validation accuracy (%) on the ogbn-products dataset for two large-scale settings with 80 clients and 100 clients.

|  | SWIFT-FEDGNN ($K = 10$) | SWIFT-FEDGNN ($K/M = 1/2$) | LLCG | FEDGNN-PNS | FEDSAGE | FEDGNN-G |
|---|---|---|---|---|---|---|
| $M = 80$ | 85.74 | 86.12 | 83.54 | 85.67 | 85.35 | 86.52 |
| $M = 100$ | 85.07 | 85.53 | 84.35 | 85.15 | 84.41 | 85.63 |

**Evaluations of large-scale settings:** Table 9 presents the validation accuracy on the ogbn-products dataset for two large-scale settings with 80 clients and 100 clients. These results show that Swift-FedGNN maintains comparable or better validation accuracy in both settings, with larger $K$ yielding slightly improved performance. These findings confirm that even with a small subset of sampled clients, Swift-FedGNN ensures stable convergence, while significantly lowering communication overhead.

### G.1.4 EVALUATION ACROSS DIFFERENT GNN MODELS

Table 10: Validation accuracy (%) of different algorithms using the GIN model.

|  | OGBN-PRODUCTS | OGBN-ARXIV | CITESEER |
|---|---|---|---|
| **SWIFT-FEDGNN** | **81.93** | **56.69** | **47.34** |
| LLCG | 80.72 | 57.32 | 46.60 |
| FEDGNN-PNS | 78.70 | 56.54 | 47.99 |
| FEDGNN-G | 83.76 | 57.01 | 50.76 |

**Evaluations using the GIN model:** To assess the adaptivity of Swift-FedGNN to different GNN models, we conduct experiments using the GIN Xu et al. (2019) model across multiple datasets. Table 10 shows that, similar to the results with the GraphSAGE model, Swift-FedGNN achieves

Table 11: Performance comparison using the GIN model when achieving a target validation accuracy for each dataset.

| | OGBN-PRODUCTS | | OGBN-ARXIV | |
|---|---|---|---|---|
| | TOTAL COMM. COST (GB) | WALL-CLOCK TIME (S) | TOTAL COMM. COST (GB) | WALL-CLOCK TIME (S) |
| **SWIFT-FEDGNN** | **0.74** | **65.18** | **2.29** | **75.80** |
| LLCG | 3.74 | 223.77 | 3.81 | 103.98 |
| FEDGNN-PNS | 5.75 | 113.46 | 3.62 | 131.92 |
| FEDGNN-G | 38.12 | 767.53 | 19.36 | 575.26 |

comparable validation accuracy to the baseline algorithms while significantly reducing sampling and communication overhead.

Table 11 reports the total communication cost and wall-clock time on the ogbn-products and ogbn-arxiv datasets when reaching a target validation accuracy of 80% and 56%, respectively. In both cases, Swift-FedGNN achieves the lowest wall-clock time to reach the target accuracy. Moreover, it reduces the total communication cost by at least 80% on ogbn-products and 37% on ogbn-arxiv compared to all baselines. These results demonstrate the effectiveness of Swift-FedGNN in significantly reducing communication overhead when using the GIN model.

### G.1.5 EVALUATION UNDER MORE HETEROGENEOUS SETTINGS

Table 12: Total communication cost (GB) using randomly partitioned ogbn-products dataset when achieving a target validation accuracy of 89.5%.

| | **SWIFT-FEDGNN** | FEDGNN-G | LLCG | FEDGNN-PNS |
|---|---|---|---|---|
| COMMUNICATION COST (GB) | **1.44** | 15.03 | 6.26 | 2.60 |

Table 13: Comparison of validation accuracy (%) using randomly partitioned ogbn-products dataset.

| | **SWIFT-FEDGNN** | FEDGNN-G | LLCG | FEDGNN-PNS |
|---|---|---|---|---|
| VALIDATION ACCURACY (%) | **89.94** | 91.23 | 89.92 | 89.91 |

**Evaluations using randomly partitioned ogbn-products dataset:** To evaluate the robustness of Swift-FedGNN under less structured scenarios, we conduct additional experiments using random partitioning instead of METIS on the ogbn-products dataset. Random partitioning introduces hetero-geneity by randomly assigning nodes to different subgraphs, thereby implicitly inducing non-identical and structurally unbalanced local subgraphs. Table 12 shows the total communication cost when achieving a target validation accuracy of 89.5%. Swift-FedGNN reduces total communication cost by at least 45% compared to all baselines. Table 13 reports the validation accuracy, demonstrating that Swift-FedGNN achieves the highest accuracy among methods that do not rely on full graph training. These results confirm that Swift-FedGNN maintains both communication efficiency and competitive performance even when the data is randomly partitioned, validating its applicability beyond well-partitioned settings.

### G.2 ADDITIONAL EXPERIMENTAL DETAILS

**Dataset.** Table 14 summarizes the key statistics of the datasets used in our experiments, includ-ing: 1) ogbn-products Hu et al. (2020), which is an Amazon product co-purchasing graph derived from Leskovec et al. (2007); 2) Reddit Hamilton et al. (2017), which consists of online forum posts within a month, where posts commented on by the same user are connected by an edge; 3) ogbn-arxiv Hu et al. (2020), which is a citation network between arXiv papers in the field of computer science, where nodes represent papers and directed edges indicate citation links; 4) flickr Zeng et al. (2020), which is an image network where each node represents an image and edges connect images that share common properties such as tags or visual similarity; and 5) citeseer Giles et al. (1998), which is a citation graph of research papers, where each node denotes a document and edges represent citation relationships between them.

**Implementation and testbed.** We implement Swift-FedGNN using Python on DGL 2.0.0 Wang et al. (2019b) and PyTorch 2.2.1 Paszke et al. (2019). Our implementation includes a custom GPU-based

Table 14: Benchmark datasets and key parameters.

| DATASET | # OF NODES | # OF EDGES | EDGES PER NODE |
|---|---|---|---|
| OGBN-PRODUCTS | 2.4 M | 61.9 M | 25.8 |
| REDDIT | 233 K | 114.6 M | 491.4 |
| OGBN-ARXIV | 169 K | 1.2 M | 7.1 |
| FLICKR | 89 K | 900 K | 10.1 |
| CITESEER | 3327 | 9228 | 2.8 |

sampler built on top of DGL's native sampler, which is designed to sequentially sample local and remote neighbors for each client at every layer. Additionally, we customized the GraphSAGE layer and GIN layer to facilitate model-parallel training within Swift-FedGNN . In this setup, the server handles the sampling and aggregation of node features and intermediate activations, while the clients are responsible for executing the nonlinear computations associated with the GraphSAGE layer.

We simulate a real-world federated learning scenario using a single machine equipped with NVIDIA Tesla V100 GPUs and 64GB memory. In our setup, both the clients and the server operate on the GPU, and data communication between them is simulated using shared memory. We monitor the data transfer size between the server and clients and set a simulated cross-client network bandwidth at 1Gbps, aligning with real-world measurements reported in Yuan et al. (2022).

**GNN model.** We train a two-layer GraphSAGE model and a two-layer GIN model with a hidden dimension of 256. Uniform sampling is employed for neighbor sampling, with fan-outs—*i.e.*, the number of sampled neighbors—set according to the official training script provided by the DGL team. The fan-out values are set to $[20, 15]$ for the ogbn-products dataset, and $[15, 10]$ for all other datasets.

**Hyperparameters.** The training mini-batch size is set at 256. For optimization, we use the Adam optimizer with a weight decay of $5 \times 10^{-4}$. We use a learning rate of 0.01 for the ogbn-products dataset, 0.0001 for the flickr dataset, 0.00001 for the citeseer dataset, and 0.001 for both the ogbn-arxiv and Reddit datasets. In Swift-FedGNN, we set $K = 10$ for the ogbn-products dataset and $K = 5$ for all other datasets. We choose $I = 5$ for the citeseer dataset and $I = 10$ for the remaining datasets.

# H    PROOF OF THEOREM 5.6

## H.1    GRADIENT COMPUTATIONS IN Swift-FedGNN

Recall that Swift-FedGNN uses GCN Kipf & Welling (2017) as the architecture of GNN to prove the convergence performance. When client $m$ performs local training that updates the local GNN model using only the local graph data, Each sampling-based GCN layer executes one feature propagation step, defined as:

$$\widetilde{\boldsymbol{H}}_{local}^{(l),m} = \left[ \widetilde{f}^{(l),m} \left( \widetilde{\boldsymbol{H}}_{local}^{(l-1),m}, \boldsymbol{W}^{(l),m} \right) \triangleq \sigma \left( \widetilde{\boldsymbol{P}}_{local}^{(l),m} \widetilde{\boldsymbol{H}}_{local}^{(l-1),m} \boldsymbol{W}^{(l),m} \right) \right].$$

Using the chain rule, the stochastic gradient can be computed as $\nabla \widetilde{F}^m \left( \boldsymbol{\theta}^m \right) = \left\{ \widetilde{\boldsymbol{G}}_{local}^{(l),m} \right\}_{l=1}^{L}$, where

$$\widetilde{\boldsymbol{G}}_{local}^{(l),m} = \left[ \nabla_W \widetilde{f}^{(l),m} \left( \widetilde{\boldsymbol{D}}_{local}^{(l),m}, \widetilde{\boldsymbol{H}}_{local}^{(l-1),m}, \boldsymbol{W}^{(l),m} \right) \right.$$

$$\left. \triangleq \left[ \widetilde{\boldsymbol{P}}_{local}^{(l),m} \widetilde{\boldsymbol{H}}_{local}^{(l-1),m} \right]^{\top} \widetilde{\boldsymbol{D}}_{local}^{(l),m} \circ \nabla \sigma \left( \widetilde{\boldsymbol{Z}}_{local}^{(l),m} \right) \right],$$

$$\widetilde{\boldsymbol{D}}_{local}^{(l),m} = \left[ \nabla_H \widetilde{f}^{(l+1),m} \left( \widetilde{\boldsymbol{D}}_{local}^{(l+1),m}, \widetilde{\boldsymbol{H}}_{local}^{(l),m}, \boldsymbol{W}^{(l+1),m} \right) \right.$$

$$\left. \triangleq \left[ \widetilde{\boldsymbol{P}}_{local}^{(l+1),m} \right]^{\top} \widetilde{\boldsymbol{D}}_{local}^{(l+1),m} \circ \nabla \sigma \left( \widetilde{\boldsymbol{Z}}_{local}^{(l+1),m} \right) \left[ \boldsymbol{W}^{(l+1),m} \right]^{\top} \right],$$

in which $\widetilde{\boldsymbol{Z}}_{local}^{(l),m} = \widetilde{\boldsymbol{P}}_{local}^{(l),m} \widetilde{\boldsymbol{H}}_{local}^{(l-1),m} \boldsymbol{W}^{(l),m}$, $\widetilde{\boldsymbol{D}}_{local}^{(L),m} = \partial \ell^m \left( \widetilde{\boldsymbol{H}}_{local}^{(L),m}, \boldsymbol{Y}_{local}^m \right) / \partial \widetilde{\boldsymbol{H}}_{local}^{(L),m}$, and $\circ$ represents Hadamard product.

Similarly, when client $m$ conducts cross-client training that updates the local GNN model based on the local graph data and the cross-client neighbors, each sampling-based GNN layer can be defined as:

$$\widetilde{\boldsymbol{H}}_{full}^{(l),m} = \left[ \widetilde{f}^{(l),m} \left( \widetilde{\boldsymbol{H}}_{full}^{(l-1),m}, \boldsymbol{W}^{(l),m} \right) \triangleq \sigma \left( \left( \widetilde{\boldsymbol{P}}_{local}^{(l),m} \widetilde{\boldsymbol{H}}_{local}^{(l-1),m} + \widetilde{\boldsymbol{P}}_{remote}^{(l),m} \widetilde{\boldsymbol{H}}_{remote}^{(l-1),m} \right) \boldsymbol{W}^{(l),m} \right) \right].$$

Using the chain rule, the stochastic gradient can be calculated as $\nabla \widetilde{F}^m \left( \boldsymbol{\theta}^m \right) = \left\{ \widetilde{\boldsymbol{G}}_{full}^{(l),m} \right\}_{l=1}^{L}$, where

$$\widetilde{\boldsymbol{G}}_{full}^{(l),m} = \left[ \nabla_W \widetilde{f}^{(l),m} \left( \widetilde{\boldsymbol{D}}_{full}^{(l),m}, \widetilde{\boldsymbol{H}}_{full}^{(l-1),m}, \boldsymbol{W}^{(l),m} \right) \right.$$

$$\triangleq \left[ \widetilde{\boldsymbol{P}}_{local}^{(l),m} \widetilde{\boldsymbol{H}}_{local}^{(l-1),m} + \widetilde{\boldsymbol{P}}_{remote}^{(l),m} \widetilde{\boldsymbol{H}}_{remote}^{(l-1),m} \right]^{\top} \widetilde{\boldsymbol{D}}_{full}^{(l),m} \circ \nabla \sigma \left( \widetilde{\boldsymbol{Z}}_{full}^{(l),m} \right) \right],$$

$$\widetilde{\boldsymbol{D}}_{full}^{(l),m} = \left[ \nabla_H \widetilde{f}^{(l+1),m} \left( \widetilde{\boldsymbol{D}}_{full}^{(l+1),m}, \widetilde{\boldsymbol{H}}_{full}^{(l),m}, \boldsymbol{W}^{(l+1),m} \right) \right.$$

$$\triangleq \left[ \widetilde{\boldsymbol{P}}_{local}^{(l+1),m} + \widetilde{\boldsymbol{P}}_{remote}^{(l+1),m} \right]^{\top} \widetilde{\boldsymbol{D}}_{full}^{(l+1),m} \circ \nabla \sigma \left( \widetilde{\boldsymbol{Z}}_{full}^{(l+1),m} \right) \left[ \boldsymbol{W}^{(l+1),m} \right]^{\top} \right],$$

in which $\widetilde{\boldsymbol{Z}}_{full}^{(l),m} = \left( \widetilde{\boldsymbol{P}}_{local}^{(l),m} \widetilde{\boldsymbol{H}}_{local}^{(l-1),m} + \widetilde{\boldsymbol{P}}_{remote}^{(l),m} \widetilde{\boldsymbol{H}}_{remote}^{(l-1),m} \right) \boldsymbol{W}^{(l),m}$, and $\widetilde{\boldsymbol{D}}_{full}^{(L),m} = \partial \ell^m \left( \widetilde{\boldsymbol{H}}_{full}^{(L),m}, \boldsymbol{Y}_{full}^m \right) / \partial \widetilde{\boldsymbol{H}}_{full}^{(L),m}$.

### H.2 Useful Propositions and Lemmas

**Proposition H.1.** *Under Assumption 5.3, the inequalities in Table 15 and Table 16 are hold.*

Table 15: Upper-bound for the norms of the propagation matrix and the node feature matrix.

|  | Propagation Matrix | Node Feature Matrix |
|---|---|---|
| Full Graph | $\|\boldsymbol{P}_{full}\|_F \leq B_P$ | $\|\boldsymbol{X}_{full}\|_F \leq B_X$ |
| Local Graph | $\|\boldsymbol{P}_{local}\|_F \leq B_P^l \leq B_P$ | $\|\boldsymbol{X}_{local}\|_F \leq B_X^l \leq B_X$ |
| Cross-Client Neighbors | $\|\boldsymbol{P}_{remote}\|_F \leq B_P^r \leq B_P$ | $\|\boldsymbol{X}_{remote}\|_F \leq B_X^r \leq B_X$ |

Table 16: Relationships for the norms of the propagation matrix and the node feature matrix before and after sampling.

|  | Propagation Matrix | Node Feature Matrix |
|---|---|---|
| Full Graph | $\left\| \widetilde{\boldsymbol{P}}_{full} - \boldsymbol{P}_{full} \right\|_F \leq B_{\Delta P}^f$ | $\left\| \widetilde{\boldsymbol{X}}_{full} - \boldsymbol{X}_{full} \right\|_F \leq B_{\Delta X}^f$ |
| Local Graph | $\left\| \widetilde{\boldsymbol{P}}_{local} - \boldsymbol{P}_{local} \right\|_F \leq B_{\Delta P}^l$ | $\left\| \widetilde{\boldsymbol{X}}_{local} - \boldsymbol{X}_{local} \right\|_F \leq B_{\Delta X}^l$ |
| Cross-Client Neighbors | $\left\| \widetilde{\boldsymbol{P}}_{remote} - \boldsymbol{P}_{remote} \right\|_F \leq B_{\Delta P}^r$ | $\left\| \widetilde{\boldsymbol{X}}_{remote} - \boldsymbol{X}_{remote} \right\|_F \leq B_{\Delta X}^r$ |

**Lemma H.2.** *[Lemma 1 in Cong et al. (2021)] An L-later GCN is $L_F$-Lipschitz smooth, i.e.,* $\|\nabla \mathcal{L} \left( \boldsymbol{\theta}_1 \right) - \nabla \mathcal{L} \left( \boldsymbol{\theta}_2 \right)\|_F \leq L_F \|\boldsymbol{\theta}_1 - \boldsymbol{\theta}_2\|_F$.

**Lemma H.3.** *Under Assumptions 5.1–5.3, and for any $l \in [L]$, the Frobenius norm of node embedding matrices, gradient passing from the l-th layer node embeddings to the $(l-1)$-th are bounded, i.e.,*

$$\left\| \boldsymbol{H}_{local}^{(l),m} \right\|_F, \left\| \widetilde{\boldsymbol{H}}_{local}^{(l),m} \right\|_F \leq B_H^l, \qquad \left\| \boldsymbol{H}_{full}^{(l),m} \right\|_F, \left\| \widetilde{\boldsymbol{H}}_{full}^{(l),m} \right\|_F \leq B_H^f,$$

$$\left\| \boldsymbol{D}_{local}^{(l),m} \right\|_F, \left\| \widetilde{\boldsymbol{D}}_{local}^{(l),m} \right\|_F \leq B_D^l, \qquad \left\| \boldsymbol{D}_{full}^{(l),m} \right\|_F, \left\| \widetilde{\boldsymbol{D}}_{full}^{(l),m} \right\|_F \leq B_D^f,$$

*where*

$$B_H^l, B_H^f = \max_{1 \leq l \leq L} \left( C_\sigma B_P B_W \right)^l B_X, \qquad B_D^l, B_D^f = \max_{1 \leq l \leq L} \left( B_P B_W C_\sigma \right)^{L-l} C_l.$$

*Proof.*

$$\left\| \boldsymbol{H}_{local}^{(l),m} \right\|_F = \left\| \sigma \left( \boldsymbol{P}_{local}^{(l),m} \boldsymbol{H}_{local}^{(l-1),m} \boldsymbol{W}^{(l),m} \right) \right\|_F$$

$$\overset{(a)}{\leq} C_\sigma B_W \left\| \boldsymbol{P}_{local}^{(l),m} \boldsymbol{H}_{local}^{(l-1),m} \right\|_F \leq C_\sigma B_W \left\| \boldsymbol{P}_{local}^{(l),m} \right\| \left\| \boldsymbol{H}_{local}^{(l-1),m} \right\|_F$$

$$\overset{(b)}{\leq} C_\sigma B_W B_P \left\| \boldsymbol{H}_{local}^{(l-1),m} \right\|_F \leq (C_\sigma B_W B_P)^l \left\| \boldsymbol{X}^m \right\|_F$$

$$\overset{(c)}{\leq} (C_\sigma B_W B_P)^l B_X \leq \max_{1 \leq l \leq L} (C_\sigma B_W B_P)^l B_X,$$

where (a)–(c) results from Assumptions 5.2 and 5.3.

$$\left\| \widetilde{\boldsymbol{H}}_{local}^{(l),m} \right\|_F = \left\| \sigma \left( \widetilde{\boldsymbol{P}}_{local}^{(l),m} \widetilde{\boldsymbol{H}}_{local}^{(l-1),m} \boldsymbol{W}^{(l),m} \right) \right\|_F$$

$$\overset{(a)}{\leq} C_\sigma B_W \left\| \widetilde{\boldsymbol{P}}_{local}^{(l),m} \widetilde{\boldsymbol{H}}_{local}^{(l-1),m} \right\|_F \leq C_\sigma B_W \left\| \widetilde{\boldsymbol{P}}_{local}^{(l),m} \right\| \left\| \widetilde{\boldsymbol{H}}_{local}^{(l-1),m} \right\|_F$$

$$\overset{(b)}{\leq} C_\sigma B_W B_P \left\| \widetilde{\boldsymbol{H}}_{local}^{(l-1),m} \right\|_F \leq (C_\sigma B_W B_P)^l \left\| \boldsymbol{X}^m \right\|_F$$

$$\overset{(c)}{\leq} (C_\sigma B_W B_P)^l B_X \leq \max_{1 \leq l \leq L} (C_\sigma B_W B_P)^l B_X,$$

where (a)–(c) follow from Assumptions 5.2 and 5.3.

$$\left\| \boldsymbol{H}_{full}^{(l),m} \right\|_F = \left\| \sigma \left( \boldsymbol{P}_{full}^{(l),m} \boldsymbol{H}_{full}^{(l-1),m} \boldsymbol{W}^{(l),m} \right) \right\|_F$$

$$\overset{(a)}{\leq} C_\sigma B_P B_W \left\| \boldsymbol{H}_{full}^{(l-1),m} \right\|_F \leq (C_\sigma B_P B_W)^l \left\| \boldsymbol{X}^m \right\|_F$$

$$\overset{(b)}{\leq} (C_\sigma B_P B_W)^l B_X \leq \max_{1 \leq l \leq L} (C_\sigma B_P B_W)^l B_X,$$

where (a) and (b) are because of Assumptions 5.2 and 5.3.

$$\left\| \widetilde{\boldsymbol{H}}_{full}^{(l),m} \right\|_F = \left\| \sigma \left( \left( \widetilde{\boldsymbol{P}}_{local}^{(l),m} \widetilde{\boldsymbol{H}}_{local}^{(l-1),m} + \widetilde{\boldsymbol{P}}_{remote}^{(l),m} \widetilde{\boldsymbol{H}}_{remote}^{(l-1),m} \right) \boldsymbol{W}^{(l),m} \right) \right\|_F$$

$$\overset{(a)}{\leq} C_\sigma B_W \left\| \widetilde{\boldsymbol{P}}_{local}^{(l),m} \widetilde{\boldsymbol{H}}_{local}^{(l-1),m} + \widetilde{\boldsymbol{P}}_{remote}^{(l),m} \widetilde{\boldsymbol{H}}_{remote}^{(l-1),m} \right\|_F = C_\sigma B_W \left\| \widetilde{\boldsymbol{P}}_{full}^{(l),m} \widetilde{\boldsymbol{H}}_{full}^{(l-1),m} \right\|_F$$

$$\overset{(b)}{\leq} C_\sigma B_W B_P \left\| \widetilde{\boldsymbol{H}}_{full}^{(l-1),m} \right\|_F \leq (C_\sigma B_W B_P)^l \left\| \boldsymbol{X}^m \right\|_F$$

$$\overset{(c)}{\leq} (C_\sigma B_W B_P)^l B_X \leq \max_{1 \leq l \leq L} (C_\sigma B_W B_P)^l B_X,$$

where (a)–(c) follow from Assumptions 5.2 and 5.3.

$$\left\| \boldsymbol{D}_{local}^{(l),m} \right\|_F = \left\| \left[ \boldsymbol{P}_{local}^{(l+1),m} \right]^\top \boldsymbol{D}_{local}^{(l+1),m} \circ \nabla \sigma \left( \boldsymbol{Z}_{local}^{(l+1),m} \right) \left[ \boldsymbol{W}^{(l+1),m} \right]^\top \right\|_F$$

$$\overset{(a)}{\leq} B_W C_\sigma \left\| \boldsymbol{P}_{local}^{(l+1),m} \right\|_F \left\| \boldsymbol{D}_{local}^{(l+1),m} \right\|_F$$

$$\overset{(b)}{\leq} B_P B_W C_\sigma \left\| \boldsymbol{D}_{local}^{(l+1),m} \right\|_F \leq (B_P B_W C_\sigma)^{L-l} \left\| \boldsymbol{D}_{local}^{(L),m} \right\|_F$$

$$\overset{(c)}{\leq} (B_P B_W C_\sigma)^{L-l} C_l \leq \max_{1 \leq l \leq L} (B_P B_W C_\sigma)^{L-l} C_l,$$

where (a)–(c) are because of Assumptions 5.1–5.3.

$$\left\| \widetilde{\boldsymbol{D}}_{local}^{(l),m} \right\|_F = \left\| \left[ \widetilde{\boldsymbol{P}}_{local}^{(l+1),m} \right]^\top \widetilde{\boldsymbol{D}}_{local}^{(l+1),m} \circ \nabla \sigma \left( \widetilde{\boldsymbol{Z}}_{local}^{(l+1),m} \right) \left[ \boldsymbol{W}^{(l+1),m} \right]^\top \right\|_F$$

$$
\stackrel{(a)}{\leq} B_W C_\sigma \left\| \widetilde{\boldsymbol{P}}_{local}^{(l+1),m} \right\|_F \left\| \widetilde{\boldsymbol{D}}_{local}^{(l+1),m} \right\|_F
$$

$$
\stackrel{(b)}{\leq} B_P B_W C_\sigma \left\| \widetilde{\boldsymbol{D}}_{local}^{(l+1),m} \right\|_F \leq (B_P B_W C_\sigma)^{L-l} \left\| \widetilde{\boldsymbol{D}}_{local}^{(L),m} \right\|_F
$$

$$
\stackrel{(c)}{\leq} (B_P B_W C_\sigma)^{L-l} C_l \leq \max_{1 \leq l \leq L} (B_P B_W C_\sigma)^{L-l} C_l,
$$

where (a)–(c) follow from Assumptions 5.1–5.3.

$$
\left\| \boldsymbol{D}_{full}^{(l),m} \right\|_F = \left\| \left[ \boldsymbol{P}_{full}^{(l+1),m} \right]^\top \boldsymbol{D}_{full}^{(l+1),m} \circ \nabla \sigma \left( \boldsymbol{Z}_{full}^{(l+1),m} \right) \left[ \boldsymbol{W}^{(l+1),m} \right]^\top \right\|_F
$$

$$
\stackrel{(a)}{\leq} B_P B_W C_\sigma \left\| \boldsymbol{D}_{full}^{(l+1),m} \right\|_F \leq (B_P B_W C_\sigma)^{L-l} \left\| \boldsymbol{D}_{full}^{(L),m} \right\|_F
$$

$$
\stackrel{(b)}{\leq} (B_P B_W C_\sigma)^{L-l} C_l \leq \max_{1 \leq l \leq L} (B_P B_W C_\sigma)^{L-l} C_l,
$$

where (a) and (b) use Assumptions 5.1–5.3.

$$
\left\| \widetilde{\boldsymbol{D}}_{full}^{(l),m} \right\|_F = \left\| \left[ \widetilde{\boldsymbol{P}}_{local}^{(l+1),m} + \widetilde{\boldsymbol{P}}_{remote}^{(l+1),m} \right]^\top \widetilde{\boldsymbol{D}}_{full}^{(l+1),m} \circ \nabla \sigma \left( \widetilde{\boldsymbol{Z}}_{full}^{(l+1),m} \right) \left[ \boldsymbol{W}^{(l+1),m} \right]^\top \right\|_F
$$

$$
= \left\| \left[ \widetilde{\boldsymbol{P}}_{full}^{(l+1),m} \right]^\top \widetilde{\boldsymbol{D}}_{full}^{(l+1),m} \circ \nabla \sigma \left( \widetilde{\boldsymbol{Z}}_{full}^{(l+1),m} \right) \left[ \boldsymbol{W}^{(l+1),m} \right]^\top \right\|_F
$$

$$
\stackrel{(a)}{\leq} B_P B_W C_\sigma \left\| \widetilde{\boldsymbol{D}}_{full}^{(l+1),m} \right\|_F \leq (B_P B_W C_\sigma)^{L-l} \left\| \widetilde{\boldsymbol{D}}_{full}^{(L),m} \right\|_F
$$

$$
\stackrel{(b)}{\leq} (B_P B_W C_\sigma)^{L-l} C_l \leq \max_{1 \leq l \leq L} (B_P B_W C_\sigma)^{L-l} C_l,
$$

where (a) and (b) utilize Assumptions 5.1–5.3.

$\square$

**Lemma H.4.** *Under Assumptions 5.1–5.3, and for any $l \in [L]$, the errors caused by sampling are bounded, i.e.,*

$$
\left\| \widetilde{\boldsymbol{H}}_{local}^{(l),m} - \boldsymbol{H}_{local}^{(l),m} \right\|_F \leq B_{\Delta H}^l, \qquad \left\| \widetilde{\boldsymbol{H}}_{full}^{(l),m} - \boldsymbol{H}_{full}^{(l),m} \right\|_F \leq B_{\Delta H}^f,
$$

$$
\left\| \widetilde{\boldsymbol{D}}_{local}^{(l),m} - \boldsymbol{D}_{local}^{(l),m} \right\|_F \leq B_{\Delta D}^l, \qquad \left\| \widetilde{\boldsymbol{D}}_{full}^{(l),m} - \boldsymbol{D}_{full}^{(l),m} \right\|_F \leq B_{\Delta D}^f,
$$

*where*

$$
B_{\Delta H}^l = \max_{1 \leq l \leq L} \left( \left( C_\sigma B_W B_H^l B_{\Delta P}^l \right)^l + \left( C_\sigma B_W B_P \right)^l B_{\Delta X}^l \right),
$$

$$
B_{\Delta H}^f = \max_{1 \leq l \leq L} \left( \left( C_\sigma B_W B_H^f B_{\Delta P}^f \right)^l + \left( C_\sigma B_W B_P \right)^l B_{\Delta X}^f \right),
$$

$$
B_{\Delta D}^l = \max_{1 \leq l \leq L} \left( \left( B_W B_D^l C_\sigma B_{\Delta P}^l + B_W^2 B_P B_D^l L_\sigma B_H^l B_{\Delta P}^l + B_W^2 B_P^2 B_D^l L_\sigma B_{\Delta H}^l \right)^{L-l} \right.
$$

$$
\left. + \left( B_W B_P C_\sigma \right)^{L-l} L_l B_{\Delta H}^l \right),
$$

$$
B_{\Delta D}^f = \max_{1 \leq l \leq L} \left( \left( B_W B_D^f C_\sigma B_{\Delta P}^f + B_W^2 B_P B_D^f L_\sigma B_H^f B_{\Delta P}^f + B_W^2 B_P^2 B_D^f L_\sigma B_{\Delta H}^f \right)^{L-l} \right.
$$

$$
\left. + \left( B_W B_P C_\sigma \right)^{L-l} L_l B_{\Delta H}^f \right).
$$

*Proof.*

$$\left\| \widetilde{\boldsymbol{H}}_{local}^{(l),m} - \boldsymbol{H}_{local}^{(l),m} \right\|_F$$

$$= \left\| \sigma \left( \widetilde{\boldsymbol{P}}_{local}^{(l),m} \widetilde{\boldsymbol{H}}_{local}^{(l-1),m} \boldsymbol{W}^{(l),m} \right) - \sigma \left( \boldsymbol{P}_{local}^{(l),m} \boldsymbol{H}_{local}^{(l-1),m} \right) \boldsymbol{W}^{(l),m} \right\|_F$$

$$\overset{(a)}{\leq} C_\sigma B_W \left\| \widetilde{\boldsymbol{P}}_{local}^{(l),m} \widetilde{\boldsymbol{H}}_{local}^{(l-1),m} - \boldsymbol{P}_{local}^{(l),m} \boldsymbol{H}_{local}^{(l-1),m} \right\|_F$$

$$\leq C_\sigma B_W \left\| \widetilde{\boldsymbol{P}}_{local}^{(l),m} \widetilde{\boldsymbol{H}}_{local}^{(l-1),m} - \boldsymbol{P}_{local}^{(l),m} \widetilde{\boldsymbol{H}}_{local}^{(l-1),m} \right\|_F + C_\sigma B_W \left\| \boldsymbol{P}_{local}^{(l),m} \widetilde{\boldsymbol{H}}_{local}^{(l-1),m} - \boldsymbol{P}_{local}^{(l),m} \boldsymbol{H}_{local}^{(l-1),m} \right\|_F$$

$$\overset{(b)}{\leq} C_\sigma B_W B_H^l \left\| \widetilde{\boldsymbol{P}}_{local}^{(l),m} - \boldsymbol{P}_{local}^{(l),m} \right\|_F + C_\sigma B_W B_P \left\| \widetilde{\boldsymbol{H}}_{local}^{(l-1),m} - \boldsymbol{H}_{local}^{(l-1),m} \right\|_F$$

$$\overset{(c)}{\leq} C_\sigma B_W B_H^l B_{\Delta P}^l + C_\sigma B_W B_P \left\| \widetilde{\boldsymbol{H}}_{local}^{(l-1),m} - \boldsymbol{H}_{local}^{(l-1),m} \right\|_F$$

$$\leq \left( C_\sigma B_W B_H^l B_{\Delta P}^l \right)^l + \left( C_\sigma B_W B_P \right)^l \left\| \widetilde{\boldsymbol{X}}_{local}^m - \boldsymbol{X}_{local}^m \right\|_F$$

$$\overset{(d)}{\leq} \left( C_\sigma B_W B_H^l B_{\Delta P}^l \right)^l + \left( C_\sigma B_W B_P \right)^l B_{\Delta X}^l$$

$$\leq \max_{1 \leq l \leq L} \left( \left( C_\sigma B_W B_H^l B_{\Delta P}^l \right)^l + \left( C_\sigma B_W B_P \right)^l B_{\Delta X}^l \right), \tag{9}$$

where (a) uses Assumptions 5.2 and 5.3, (b) is because of Assumption 5.3 and Lemma H.3, and (c) and (d) follow from Proposition H.1.

$$\left\| \widetilde{\boldsymbol{H}}_{full}^{(l),m} - \boldsymbol{H}_{full}^{(l),m} \right\|_F$$

$$= \left\| \sigma \left( \left( \widetilde{\boldsymbol{P}}_{local}^{(l),m} \widetilde{\boldsymbol{H}}_{local}^{(l-1),m} + \widetilde{\boldsymbol{P}}_{remote}^{(l),m} \widetilde{\boldsymbol{H}}_{remote}^{(l-1),m} \right) \boldsymbol{W}^{(l),m} \right) - \sigma \left( \boldsymbol{P}_{full}^{(l),m} \boldsymbol{H}_{full}^{(l-1),m} \right) \boldsymbol{W}^{(l),m} \right\|_F$$

$$\overset{(a)}{\leq} C_\sigma B_W \left\| \widetilde{\boldsymbol{P}}_{full}^{(l),m} \widetilde{\boldsymbol{H}}_{full}^{(l-1),m} - \boldsymbol{P}_{full}^{(l),m} \boldsymbol{H}_{full}^{(l-1),m} \right\|_F$$

$$\leq C_\sigma B_W \left\| \widetilde{\boldsymbol{P}}_{full}^{(l),m} \widetilde{\boldsymbol{H}}_{full}^{(l-1),m} - \boldsymbol{P}_{full}^{(l),m} \widetilde{\boldsymbol{H}}_{full}^{(l-1),m} \right\|_F + C_\sigma B_W \left\| \boldsymbol{P}_{full}^{(l),m} \widetilde{\boldsymbol{H}}_{full}^{(l-1),m} - \boldsymbol{P}_{full}^{(l),m} \boldsymbol{H}_{full}^{(l-1),m} \right\|_F$$

$$\overset{(b)}{\leq} C_\sigma B_W B_H^f \left\| \widetilde{\boldsymbol{P}}_{full}^{(l),m} - \boldsymbol{P}_{full}^{(l),m} \right\|_F + C_\sigma B_W B_P \left\| \widetilde{\boldsymbol{H}}_{full}^{(l-1),m} - \boldsymbol{H}_{full}^{(l-1),m} \right\|_F$$

$$\overset{(c)}{\leq} C_\sigma B_W B_H^f B_{\Delta P}^f + C_\sigma B_W B_P \left\| \widetilde{\boldsymbol{H}}_{full}^{(l-1),m} - \boldsymbol{H}_{full}^{(l-1),m} \right\|_F$$

$$\leq \left( C_\sigma B_W B_H^f B_{\Delta P}^f \right)^l + \left( C_\sigma B_W B_P \right)^l \left\| \widetilde{\boldsymbol{X}}_{full}^m - \boldsymbol{X}_{full}^m \right\|_F$$

$$\overset{(d)}{\leq} \left( C_\sigma B_W B_H^f B_{\Delta P}^f \right)^l + \left( C_\sigma B_W B_P \right)^l B_{\Delta X}^f$$

$$\leq \max_{1 \leq l \leq L} \left( \left( C_\sigma B_W B_H^f B_{\Delta P}^f \right)^l + \left( C_\sigma B_W B_P \right)^l B_{\Delta X}^f \right), \tag{10}$$

where (a) follows from Assumptions 5.2 and 5.3, (b) is due to Assumption 5.3 and Lemma H.3, and (c) and (d) are because of Proposition H.1.

$$\left\| \widetilde{\boldsymbol{D}}_{local}^{(l),m} - \boldsymbol{D}_{local}^{(l),m} \right\|_F$$

$$= \left\| \left[ \widetilde{\boldsymbol{P}}_{local}^{(l+1),m} \right]^\top \widetilde{\boldsymbol{D}}_{local}^{(l+1),m} \circ \nabla \sigma \left( \widetilde{\boldsymbol{Z}}_{local}^{(l+1),m} \right) \left[ \boldsymbol{W}^{(l+1),m} \right]^\top \right.$$

$$\left. - \left[ \boldsymbol{P}_{local}^{(l+1),m} \right]^\top \boldsymbol{D}_{local}^{(l+1),m} \circ \nabla \sigma \left( \boldsymbol{Z}_{local}^{(l+1),m} \right) \left[ \boldsymbol{W}^{(l+1),m} \right]^\top \right\|_F$$

$$\overset{(a)}{\leq} B_W \left\| \left[ \widetilde{\boldsymbol{P}}_{local}^{(l+1),m} \right]^\top \widetilde{\boldsymbol{D}}_{local}^{(l+1),m} \circ \nabla \sigma \left( \widetilde{\boldsymbol{Z}}_{local}^{(l+1),m} \right) - \left[ \boldsymbol{P}_{local}^{(l+1),m} \right]^\top \boldsymbol{D}_{local}^{(l+1),m} \circ \nabla \sigma \left( \boldsymbol{Z}_{local}^{(l+1),m} \right) \right\|_F$$

$$\leq B_W \left\| \left[ \widetilde{\boldsymbol{P}}_{local}^{(l+1),m} \right]^\top \widetilde{\boldsymbol{D}}_{local}^{(l+1),m} \circ \nabla\sigma\left( \widetilde{\boldsymbol{Z}}_{local}^{(l+1),m} \right) - \left[ \boldsymbol{P}_{local}^{(l+1),m} \right]^\top \widetilde{\boldsymbol{D}}_{local}^{(l+1),m} \circ \nabla\sigma\left( \widetilde{\boldsymbol{Z}}_{local}^{(l+1),m} \right) \right\|_F$$

$$+ B_W \left\| \left[ \boldsymbol{P}_{local}^{(l+1),m} \right]^\top \widetilde{\boldsymbol{D}}_{local}^{(l+1),m} \circ \nabla\sigma\left( \widetilde{\boldsymbol{Z}}_{local}^{(l+1),m} \right) - \left[ \boldsymbol{P}_{local}^{(l+1),m} \right]^\top \boldsymbol{D}_{local}^{(l+1),m} \circ \nabla\sigma\left( \widetilde{\boldsymbol{Z}}_{local}^{(l+1),m} \right) \right\|_F$$

$$+ B_W \left\| \left[ \boldsymbol{P}_{local}^{(l+1),m} \right]^\top \boldsymbol{D}_{local}^{(l+1),m} \circ \nabla\sigma\left( \widetilde{\boldsymbol{Z}}_{local}^{(l+1),m} \right) - \left[ \boldsymbol{P}_{local}^{(l+1),m} \right]^\top \boldsymbol{D}_{local}^{(l+1),m} \circ \nabla\sigma\left( \boldsymbol{Z}_{local}^{(l+1),m} \right) \right\|_F$$

$$\overset{(b)}{\leq} B_W B_D^l C_\sigma \left\| \widetilde{\boldsymbol{P}}_{local}^{(l+1),m} - \boldsymbol{P}_{local}^{(l+1),m} \right\|_F + B_W B_P C_\sigma \left\| \widetilde{\boldsymbol{D}}_{local}^{(l+1),m} - \boldsymbol{D}_{local}^{(l+1),m} \right\|_F$$

$$+ B_W B_P B_D^l \left\| \nabla\sigma\left( \widetilde{\boldsymbol{Z}}_{local}^{(l+1),m} \right) - \nabla\sigma\left( \boldsymbol{Z}_{local}^{(l+1),m} \right) \right\|_F$$

$$\overset{(c)}{\leq} B_W B_D^l C_\sigma \left\| \widetilde{\boldsymbol{P}}_{local}^{(l+1),m} - \boldsymbol{P}_{local}^{(l+1),m} \right\|_F + B_W B_P C_\sigma \left\| \widetilde{\boldsymbol{D}}_{local}^{(l+1),m} - \boldsymbol{D}_{local}^{(l+1),m} \right\|_F$$

$$+ B_W^2 B_P B_D^l L_\sigma \left\| \widetilde{\boldsymbol{P}}_{local}^{(l),m} \widetilde{\boldsymbol{H}}_{local}^{(l-1),m} - \boldsymbol{P}_{local}^{(l),m} \boldsymbol{H}_{local}^{(l-1),m} \right\|_F$$

$$\leq B_W B_D^l C_\sigma \left\| \widetilde{\boldsymbol{P}}_{local}^{(l+1),m} - \boldsymbol{P}_{local}^{(l+1),m} \right\|_F + B_W B_P C_\sigma \left\| \widetilde{\boldsymbol{D}}_{local}^{(l+1),m} - \boldsymbol{D}_{local}^{(l+1),m} \right\|_F$$

$$+ B_W^2 B_P B_D^l L_\sigma \left\| \widetilde{\boldsymbol{P}}_{local}^{(l),m} \widetilde{\boldsymbol{H}}_{local}^{(l-1),m} - \boldsymbol{P}_{local}^{(l),m} \widetilde{\boldsymbol{H}}_{local}^{(l-1),m} \right\|_F$$

$$+ B_W^2 B_P B_D^l L_\sigma \left\| \boldsymbol{P}_{local}^{(l),m} \widetilde{\boldsymbol{H}}_{local}^{(l-1),m} - \boldsymbol{P}_{local}^{(l),m} \boldsymbol{H}_{local}^{(l-1),m} \right\|_F$$

$$\overset{(d)}{\leq} B_W B_D^l C_\sigma \left\| \widetilde{\boldsymbol{P}}_{local}^{(l+1),m} - \boldsymbol{P}_{local}^{(l+1),m} \right\|_F + B_W B_P C_\sigma \left\| \widetilde{\boldsymbol{D}}_{local}^{(l+1),m} - \boldsymbol{D}_{local}^{(l+1),m} \right\|_F$$

$$+ B_W^2 B_P B_D^l L_\sigma B_H^l \left\| \widetilde{\boldsymbol{P}}_{local}^{(l),m} - \boldsymbol{P}_{local}^{(l),m} \right\|_F + B_W^2 B_P^2 B_D^l L_\sigma \left\| \widetilde{\boldsymbol{H}}_{local}^{(l-1),m} - \boldsymbol{H}_{local}^{(l-1),m} \right\|_F$$

$$\overset{(e)}{\leq} B_W B_D^l C_\sigma B_{\Delta P}^l + B_W^2 B_P B_D^l L_\sigma B_H^l B_{\Delta P}^l + B_W^2 B_P^2 B_D^l L_\sigma B_{\Delta H}^l$$

$$+ B_W B_P C_\sigma \left\| \widetilde{\boldsymbol{D}}_{local}^{(l+1),m} - \boldsymbol{D}_{local}^{(l+1),m} \right\|_F$$

$$\leq \left( B_W B_D^l C_\sigma B_{\Delta P}^l + B_W^2 B_P B_D^l L_\sigma B_H^l B_{\Delta P}^l + B_W^2 B_P^2 B_D^l L_\sigma B_{\Delta H}^l \right)^{L-l}$$

$$+ \left( B_W B_P C_\sigma \right)^{L-l} \left\| \widetilde{\boldsymbol{D}}_{local}^{(L),m} - \boldsymbol{D}_{local}^{(L),m} \right\|_F$$

$$\overset{(f)}{\leq} \left( B_W B_D^l C_\sigma B_{\Delta P}^l + B_W^2 B_P B_D^l L_\sigma B_H^l B_{\Delta P}^l + B_W^2 B_P^2 B_D^l L_\sigma B_{\Delta H}^l \right)^{L-l}$$

$$+ \left( B_W B_P C_\sigma \right)^{L-l} L_l \left\| \widetilde{\boldsymbol{H}}_{local}^{(L),m} - \boldsymbol{H}_{local}^{(L),m} \right\|_F$$

$$\overset{(g)}{\leq} \left( B_W B_D^l C_\sigma B_{\Delta P}^l + B_W^2 B_P B_D^l L_\sigma B_H^l B_{\Delta P}^l + B_W^2 B_P^2 B_D^l L_\sigma B_{\Delta H}^l \right)^{L-l}$$

$$+ \left( B_W B_P C_\sigma \right)^{L-l} L_l B_{\Delta H}^l$$

$$\leq \max_{1 \leq l \leq L} \left( \left( B_W B_D^l C_\sigma B_{\Delta P}^l + B_W^2 B_P B_D^l L_\sigma B_H^l B_{\Delta P}^l + B_W^2 B_P^2 B_D^l L_\sigma B_{\Delta H}^l \right)^{L-l} \right.$$

$$+ \left. \left( B_W B_P C_\sigma \right)^{L-l} L_l B_{\Delta H}^l \right),$$

where (a) uses Assumption 5.3, (b) is because of Assumptions 5.2 and 5.3 and Lemma H.3, (c) follows from Assumptions 5.2 and 5.3, (d) utilizes Assumption 5.3 and Lemma H.3, (e) results from Eq. (9) and Proposition H.1, (f) is because of Assumption 5.1, and (g) is due to Eq. (9).

$$\left\| \widetilde{\boldsymbol{D}}_{full}^{(l),m} - \boldsymbol{D}_{full}^{(l),m} \right\|_F$$

$$= \left\| \left[ \widetilde{\boldsymbol{P}}_{local}^{(l+1),m} + \widetilde{\boldsymbol{P}}_{remote}^{(l+1),m} \right]^\top \widetilde{\boldsymbol{D}}_{full}^{(l+1),m} \circ \nabla\sigma\left( \widetilde{\boldsymbol{Z}}_{full}^{(l+1),m} \right) \left[ \boldsymbol{W}^{(l+1),m} \right]^\top \right.$$

$$- \left. \left[ \boldsymbol{P}_{full}^{(l+1),m} \right]^\top \boldsymbol{D}_{full}^{(l+1),m} \circ \nabla\sigma\left( \boldsymbol{Z}_{full}^{(l+1),m} \right) \left[ \boldsymbol{W}^{(l+1),m} \right]^\top \right\|_F$$

$$\overset{(a)}{\leq} B_W \left\| \left[ \widetilde{\boldsymbol{P}}_{full}^{(l+1),m} \right]^\top \widetilde{\boldsymbol{D}}_{full}^{(l+1),m} \circ \nabla\sigma\left( \widetilde{\boldsymbol{Z}}_{full}^{(l+1),m} \right) - \left[ \boldsymbol{P}_{full}^{(l+1),m} \right]^\top \boldsymbol{D}_{full}^{(l+1),m} \circ \nabla\sigma\left( \boldsymbol{Z}_{full}^{(l+1),m} \right) \right\|_F$$

$$\leq B_W \left\| \left[ \widetilde{\boldsymbol{P}}_{full}^{(l+1),m} \right]^\top \widetilde{\boldsymbol{D}}_{full}^{(l+1),m} \circ \nabla\sigma\left( \widetilde{\boldsymbol{Z}}_{full}^{(l+1),m} \right) - \left[ \boldsymbol{P}_{full}^{(l+1),m} \right]^\top \widetilde{\boldsymbol{D}}_{full}^{(l+1),m} \circ \nabla\sigma\left( \widetilde{\boldsymbol{Z}}_{full}^{(l+1),m} \right) \right\|_F$$

$$+ B_W \left\| \left[ \boldsymbol{P}_{full}^{(l+1),m} \right]^\top \widetilde{\boldsymbol{D}}_{full}^{(l+1),m} \circ \nabla\sigma\left( \widetilde{\boldsymbol{Z}}_{full}^{(l+1),m} \right) - \left[ \boldsymbol{P}_{full}^{(l+1),m} \right]^\top \boldsymbol{D}_{full}^{(l+1),m} \circ \nabla\sigma\left( \widetilde{\boldsymbol{Z}}_{full}^{(l+1),m} \right) \right\|_F$$

$$+ B_W \left\| \left[ \boldsymbol{P}_{full}^{(l+1),m} \right]^\top \boldsymbol{D}_{full}^{(l+1),m} \circ \nabla\sigma\left( \widetilde{\boldsymbol{Z}}_{full}^{(l+1),m} \right) - \left[ \boldsymbol{P}_{full}^{(l+1),m} \right]^\top \boldsymbol{D}_{full}^{(l+1),m} \circ \nabla\sigma\left( \boldsymbol{Z}_{full}^{(l+1),m} \right) \right\|_F$$

$$\overset{(b)}{\leq} B_W B_D^f C_\sigma \left\| \widetilde{\boldsymbol{P}}_{full}^{(l+1),m} - \boldsymbol{P}_{full}^{(l+1),m} \right\|_F + B_W B_P C_\sigma \left\| \widetilde{\boldsymbol{D}}_{full}^{(l+1),m} - \boldsymbol{D}_{full}^{(l+1),m} \right\|_F$$

$$+ B_W B_P B_D^f \left\| \nabla\sigma\left( \widetilde{\boldsymbol{Z}}_{full}^{(l+1),m} \right) - \nabla\sigma\left( \boldsymbol{Z}_{full}^{(l+1),m} \right) \right\|_F$$

$$\overset{(c)}{\leq} B_W B_D^f C_\sigma \left\| \widetilde{\boldsymbol{P}}_{full}^{(l+1),m} - \boldsymbol{P}_{full}^{(l+1),m} \right\|_F + B_W B_P C_\sigma \left\| \widetilde{\boldsymbol{D}}_{full}^{(l+1),m} - \boldsymbol{D}_{full}^{(l+1),m} \right\|_F$$

$$+ B_W^2 B_P B_D^f L_\sigma \left\| \widetilde{\boldsymbol{P}}_{full}^{(l+1),m} \widetilde{\boldsymbol{H}}_{full}^{(l),m} - \boldsymbol{P}_{full}^{(l+1),m} \boldsymbol{H}_{full}^{(l),m} \right\|_F$$

$$\leq B_W B_D^f C_\sigma \left\| \widetilde{\boldsymbol{P}}_{full}^{(l+1),m} - \boldsymbol{P}_{full}^{(l+1),m} \right\|_F + B_W B_P C_\sigma \left\| \widetilde{\boldsymbol{D}}_{full}^{(l+1),m} - \boldsymbol{D}_{full}^{(l+1),m} \right\|_F$$

$$+ B_W^2 B_P B_D^f L_\sigma \left\| \widetilde{\boldsymbol{P}}_{full}^{(l+1),m} \widetilde{\boldsymbol{H}}_{full}^{(l),m} - \boldsymbol{P}_{full}^{(l+1),m} \widetilde{\boldsymbol{H}}_{full}^{(l),m} \right\|_F$$

$$+ B_W^2 B_P B_D^f L_\sigma \left\| \boldsymbol{P}_{full}^{(l+1),m} \widetilde{\boldsymbol{H}}_{full}^{(l),m} - \boldsymbol{P}_{full}^{(l+1),m} \boldsymbol{H}_{full}^{(l),m} \right\|_F$$

$$\overset{(d)}{\leq} B_W B_D^f C_\sigma \left\| \widetilde{\boldsymbol{P}}_{full}^{(l+1),m} - \boldsymbol{P}_{full}^{(l+1),m} \right\|_F + B_W B_P C_\sigma \left\| \widetilde{\boldsymbol{D}}_{full}^{(l+1),m} - \boldsymbol{D}_{full}^{(l+1),m} \right\|_F$$

$$+ B_W^2 B_P B_D^f L_\sigma B_H^f \left\| \widetilde{\boldsymbol{P}}_{full}^{(l+1),m} - \boldsymbol{P}_{full}^{(l+1),m} \right\|_F + B_W^2 B_P^2 B_D^f L_\sigma \left\| \widetilde{\boldsymbol{H}}_{full}^{(l),m} - \boldsymbol{H}_{full}^{(l),m} \right\|_F$$

$$\overset{(e)}{\leq} B_W B_D^f C_\sigma B_{\Delta P}^f + B_W^2 B_P B_D^f L_\sigma B_H^f B_{\Delta P}^f + B_W^2 B_P^2 B_D^f L_\sigma B_{\Delta H}^f$$

$$+ B_W B_P C_\sigma \left\| \widetilde{\boldsymbol{D}}_{full}^{(l+1),m} - \boldsymbol{D}_{full}^{(l+1),m} \right\|_F$$

$$\leq \left( B_W B_D^f C_\sigma B_{\Delta P}^f + B_W^2 B_P B_D^f L_\sigma B_H^f B_{\Delta P}^f + B_W^2 B_P^2 B_D^f L_\sigma B_{\Delta H}^f \right)^{L-l}$$

$$+ (B_W B_P C_\sigma)^{L-l} \left\| \widetilde{\boldsymbol{D}}_{full}^{(L),m} - \boldsymbol{D}_{full}^{(L),m} \right\|_F$$

$$\overset{(f)}{\leq} \left( B_W B_D^f C_\sigma B_{\Delta P}^f + B_W^2 B_P B_D^f L_\sigma B_H^f B_{\Delta P}^f + B_W^2 B_P^2 B_D^f L_\sigma B_{\Delta H}^f \right)^{L-l}$$

$$+ (B_W B_P C_\sigma)^{L-l} L_l \left\| \widetilde{\boldsymbol{H}}_{full}^{(L),m} - \boldsymbol{H}_{full}^{(L),m} \right\|_F$$

$$\overset{(g)}{\leq} \left( B_W B_D^f C_\sigma B_{\Delta P}^f + B_W^2 B_P B_D^f L_\sigma B_H^f B_{\Delta P}^f + B_W^2 B_P^2 B_D^f L_\sigma B_{\Delta H}^f \right)^{L-l}$$

$$+ (B_W B_P C_\sigma)^{L-l} L_l B_{\Delta H}^f$$

$$\leq \max_{1 \leq l \leq L} \left( \left( B_W B_D^f C_\sigma B_{\Delta P}^f + B_W^2 B_P B_D^f L_\sigma B_H^f B_{\Delta P}^f + B_W^2 B_P^2 B_D^f L_\sigma B_{\Delta H}^f \right)^{L-l} \right.$$

$$\left. + (B_W B_P C_\sigma)^{L-l} L_l B_{\Delta H}^f \right),$$

where (a) is because of Assumption 5.3, (b) results from Assumptions 5.2 and 5.3 and Lemma H.3, (c) uses Assumptions 5.2 and 5.3, (d) is due to Assumption 5.3 and Lemma H.3, (e) follows from Eq. (10) and Proposition H.1, (f) utilizes Assumption 5.1, and (g) is because of Eq. (10).

$$\square$$

**Lemma H.5.** *Under Assumptions 5.1–5.3, and for any $l \in [L]$, the errors caused by the information loss of the cross-client neighbors are bounded, i.e.,*

$$\left\| \boldsymbol{H}_{local}^{(l),m} - \boldsymbol{H}_{full}^{(l),m} \right\|_F \leq B_{\Delta H}^r, \qquad \left\| \boldsymbol{D}_{local}^{(l),m} - \boldsymbol{D}_{full}^{(l),m} \right\|_F \leq B_{\Delta D}^r,$$

*where*

$$B_{\Delta H}^r = \max_{1 \leq l \leq L} \left( (C_\sigma B_W B_P)^l B_X^r + \left( C_\sigma B_W B_H^f B_P \right)^l \right),$$

$$B_{\Delta D}^r = \max_{1 \leq l \leq L} \left( \left( B_W B_D^l C_\sigma B_P + B_W^2 B_P^2 B_D^f L_\sigma B_H^l + B_W^2 B_P^2 B_D^f L_\sigma B_{\Delta H}^r \right)^{L-l} \right.$$

$$\left. + (B_W B_P C\sigma)^{L-l} L_l B_{\Delta H}^r \right).$$

*Proof.*

$$\left\| \boldsymbol{H}_{local}^{(l),m} - \boldsymbol{H}_{full}^{(l),m} \right\|_F = \left\| \sigma \left( \boldsymbol{P}_{local}^{(l),m} \boldsymbol{H}_{local}^{(l-1),m} \right) \boldsymbol{W}^{(l),m} - \sigma \left( \boldsymbol{P}_{full}^{(l),m} \boldsymbol{H}_{full}^{(l-1),m} \right) \boldsymbol{W}^{(l),m} \right\|_F$$

$$\overset{(a)}{\leq} C_\sigma B_W \left\| \boldsymbol{P}_{local}^{(l),m} \boldsymbol{H}_{local}^{(l-1),m} - \boldsymbol{P}_{full}^{(l),m} \boldsymbol{H}_{full}^{(l-1),m} \right\|_F$$

$$\leq C_\sigma B_W \left\| \boldsymbol{P}_{local}^{(l),m} \boldsymbol{H}_{local}^{(l-1),m} - \boldsymbol{P}_{local}^{(l),m} \boldsymbol{H}_{full}^{(l-1),m} \right\|_F$$

$$+ C_\sigma B_W \left\| \boldsymbol{P}_{local}^{(l),m} \boldsymbol{H}_{full}^{(l-1),m} - \boldsymbol{P}_{full}^{(l),m} \boldsymbol{H}_{full}^{(l-1),m} \right\|_F$$

$$\overset{(b)}{\leq} C_\sigma B_W B_P \left\| \boldsymbol{H}_{local}^{(l-1),m} - \boldsymbol{H}_{full}^{(l-1),m} \right\|_F + C_\sigma B_W B_H^f \left\| \boldsymbol{P}_{local}^{(l),m} - \boldsymbol{P}_{full}^{(l),m} \right\|_F$$

$$\leq C_\sigma B_W B_P \left\| \boldsymbol{H}_{local}^{(l-1),m} - \boldsymbol{H}_{full}^{(l-1),m} \right\|_F + C_\sigma B_W B_H^f \left\| \boldsymbol{P}_{remote}^{(l),m} \right\|_F$$

$$\overset{(c)}{\leq} C_\sigma B_W B_P \left\| \boldsymbol{H}_{local}^{(l-1),m} - \boldsymbol{H}_{full}^{(l-1),m} \right\|_F + C_\sigma B_W B_H^f B_P$$

$$\leq (C_\sigma B_W B_P)^l \left\| \boldsymbol{X}_{local}^m - \boldsymbol{X}_{full}^m \right\|_F + \left( C_\sigma B_W B_H^f B_P \right)^l$$

$$\overset{(d)}{\leq} (C_\sigma B_W B_P)^l B_X^r + \left( C_\sigma B_W B_H^f B_P \right)^l$$

$$\leq \max_{1 \leq l \leq L} \left( (C_\sigma B_W B_P)^l B_X^r + \left( C_\sigma B_W B_H^f B_P \right)^l \right), \tag{11}$$

where (a) uses Assumptions 5.2 and 5.3, (b) is because of Assumption 5.3 and Lemma H.3, (c) follows from Assumption 5.3, and (d) is due to Proposition H.1.

$$\left\| \boldsymbol{D}_{local}^{(l),m} - \boldsymbol{D}_{full}^{(l),m} \right\|_F$$

$$= \left\| \left[ \boldsymbol{P}_{local}^{(l+1),m} \right]^\top \boldsymbol{D}_{local}^{(l+1),m} \circ \nabla\sigma \left( \boldsymbol{Z}_{local}^{(l+1),m} \right) \left[ \boldsymbol{W}^{(l+1),m} \right]^\top \right.$$

$$\left. - \left[ \boldsymbol{P}_{full}^{(l+1),m} \right]^\top \boldsymbol{D}_{full}^{(l+1),m} \circ \nabla\sigma \left( \boldsymbol{Z}_{full}^{(l+1),m} \right) \left[ \boldsymbol{W}^{(l+1),m} \right]^\top \right\|_F$$

$$\overset{(a)}{\leq} B_W \left\| \left[ \boldsymbol{P}_{local}^{(l+1),m} \right]^\top \boldsymbol{D}_{local}^{(l+1),m} \circ \nabla\sigma \left( \boldsymbol{Z}_{local}^{(l+1),m} \right) - \left[ \boldsymbol{P}_{full}^{(l+1),m} \right]^\top \boldsymbol{D}_{full}^{(l+1),m} \circ \nabla\sigma \left( \boldsymbol{Z}_{full}^{(l+1),m} \right) \right\|_F$$

$$\leq B_W \left\| \left[ \boldsymbol{P}_{local}^{(l+1),m} \right]^\top \boldsymbol{D}_{local}^{(l+1),m} \circ \nabla\sigma \left( \boldsymbol{Z}_{local}^{(l+1),m} \right) - \left[ \boldsymbol{P}_{full}^{(l+1),m} \right]^\top \boldsymbol{D}_{local}^{(l+1),m} \circ \nabla\sigma \left( \boldsymbol{Z}_{local}^{(l+1),m} \right) \right\|_F$$

$$+ B_W \left\| \left[ \boldsymbol{P}_{full}^{(l+1),m} \right]^\top \boldsymbol{D}_{local}^{(l+1),m} \circ \nabla\sigma \left( \boldsymbol{Z}_{local}^{(l+1),m} \right) - \left[ \boldsymbol{P}_{full}^{(l+1),m} \right]^\top \boldsymbol{D}_{full}^{(l+1),m} \circ \nabla\sigma \left( \boldsymbol{Z}_{local}^{(l+1),m} \right) \right\|_F$$

$$+ B_W \left\| \left[ \boldsymbol{P}_{full}^{(l+1),m} \right]^\top \boldsymbol{D}_{full}^{(l+1),m} \circ \nabla\sigma \left( \boldsymbol{Z}_{local}^{(l+1),m} \right) - \left[ \boldsymbol{P}_{full}^{(l+1),m} \right]^\top \boldsymbol{D}_{full}^{(l+1),m} \circ \nabla\sigma \left( \boldsymbol{Z}_{full}^{(l+1),m} \right) \right\|_F$$

$$\overset{(b)}{\leq} B_W B_D^l C_\sigma \left\| \boldsymbol{P}_{local}^{(l+1),m} - \boldsymbol{P}_{full}^{(l+1),m} \right\|_F + B_W B_P C\sigma \left\| \boldsymbol{D}_{local}^{(l+1),m} - \boldsymbol{D}_{full}^{(l+1),m} \right\|_F$$

$$+ B_W B_P B_D^f \left\| \nabla\sigma\left(\boldsymbol{Z}_{local}^{(l+1),m}\right) - \nabla\sigma\left(\boldsymbol{Z}_{full}^{(l+1),m}\right) \right\|_F$$

$$\overset{(c)}{\leq} B_W B_D^l C_\sigma \left\| \boldsymbol{P}_{local}^{(l+1),m} - \boldsymbol{P}_{full}^{(l+1),m} \right\|_F + B_W B_P C\sigma \left\| \boldsymbol{D}_{local}^{(l+1),m} - \boldsymbol{D}_{full}^{(l+1),m} \right\|_F$$

$$+ B_W^2 B_P B_D^f L_\sigma \left\| \boldsymbol{P}_{local}^{(l+1),m} \boldsymbol{H}_{local}^{(l),m} - \boldsymbol{P}_{full}^{(l+1),m} \boldsymbol{H}_{full}^{(l),m} \right\|_F$$

$$\leq B_W B_D^l C_\sigma \left\| \boldsymbol{P}_{local}^{(l+1),m} - \boldsymbol{P}_{full}^{(l+1),m} \right\|_F + B_W B_P C\sigma \left\| \boldsymbol{D}_{local}^{(l+1),m} - \boldsymbol{D}_{full}^{(l+1),m} \right\|_F$$

$$+ B_W^2 B_P B_D^f L_\sigma \left\| \boldsymbol{P}_{local}^{(l+1),m} \boldsymbol{H}_{local}^{(l),m} - \boldsymbol{P}_{full}^{(l+1),m} \boldsymbol{H}_{local}^{(l),m} \right\|_F$$

$$+ B_W^2 B_P B_D^f L_\sigma \left\| \boldsymbol{P}_{full}^{(l+1),m} \boldsymbol{H}_{local}^{(l),m} - \boldsymbol{P}_{full}^{(l+1),m} \boldsymbol{H}_{full}^{(l),m} \right\|_F$$

$$\overset{(d)}{\leq} B_W B_D^l C_\sigma \left\| \boldsymbol{P}_{local}^{(l+1),m} - \boldsymbol{P}_{full}^{(l+1),m} \right\|_F + B_W B_P C\sigma \left\| \boldsymbol{D}_{local}^{(l+1),m} - \boldsymbol{D}_{full}^{(l+1),m} \right\|_F$$

$$+ B_W^2 B_P B_D^f L_\sigma B_H^l \left\| \boldsymbol{P}_{local}^{(l+1),m} - \boldsymbol{P}_{full}^{(l+1),m} \right\|_F + B_W^2 B_P^2 B_D^f L_\sigma \left\| \boldsymbol{H}_{local}^{(l),m} - \boldsymbol{H}_{full}^{(l),m} \right\|_F$$

$$\overset{(e)}{\leq} B_W B_D^l C_\sigma B_P + B_W^2 B_P^2 B_D^f L_\sigma B_H^l + B_W^2 B_P^2 B_D^f L_\sigma B_{\Delta H}^r$$

$$+ B_W B_P C\sigma \left\| \boldsymbol{D}_{local}^{(l+1),m} - \boldsymbol{D}_{full}^{(l+1),m} \right\|_F$$

$$\leq \left( B_W B_D^l C_\sigma B_P + B_W^2 B_P^2 B_D^f L_\sigma B_H^l + B_W^2 B_P^2 B_D^f L_\sigma B_{\Delta H}^r \right)^{L-l}$$

$$+ (B_W B_P C\sigma)^{L-l} \left\| \boldsymbol{D}_{local}^{(L),m} - \boldsymbol{D}_{full}^{(L),m} \right\|_F$$

$$\overset{(f)}{\leq} \left( B_W B_D^l C_\sigma B_P + B_W^2 B_P^2 B_D^f L_\sigma B_H^l + B_W^2 B_P^2 B_D^f L_\sigma B_{\Delta H}^r \right)^{L-l}$$

$$+ (B_W B_P C\sigma)^{L-l} L_l \left\| \boldsymbol{H}_{local}^{(L),m} - \boldsymbol{H}_{full}^{(L),m} \right\|_F$$

$$\overset{(g)}{\leq} \left( B_W B_D^l C_\sigma B_P + B_W^2 B_P^2 B_D^f L_\sigma B_H^l + B_W^2 B_P^2 B_D^f L_\sigma B_{\Delta H}^r \right)^{L-l}$$

$$+ (B_W B_P C\sigma)^{L-l} L_l B_{\Delta H}^r$$

$$\leq \max_{1 \leq l \leq L} \left( \left( B_W B_D^l C_\sigma B_P + B_W^2 B_P^2 B_D^f L_\sigma B_H^l + B_W^2 B_P^2 B_D^f L_\sigma B_{\Delta H}^r \right)^{L-l} \right.$$

$$\left. + (B_W B_P C\sigma)^{L-l} L_l B_{\Delta H}^r \right),$$

where (a) follows from Assumption 5.3, (b) uses Assumptions 5.2 and 5.3 and Lemma H.3, (c) is because of Assumptions 5.2 and 5.3, (d) results from Assumption 5.3 and Lemma H.3, (e) is due to Assumption 5.3 and Eq. (11), (f) utilizes Assumption 5.1, and (g) uses Eq. (11).

□

### H.3 ERRORS OF STOCHASTIC GRADIENTS

**Lemma H.6.** *Under Assumptions 5.1–5.3, the errors between the stochastic gradients and the full gradients are bounded as follows:*

$$\left\| \nabla F_{local}^m(\boldsymbol{\theta}^m) - \nabla \widetilde{F}_{local}^m(\boldsymbol{\theta}^m) \right\|_F \leq L B_{\Delta G}^l, \quad \left\| \nabla F_{full}^m(\boldsymbol{\theta}^m) - \nabla \widetilde{F}_{full}^m(\boldsymbol{\theta}^m) \right\|_F \leq L B_{\Delta G}^f,$$

*where*

$$B_{\Delta G}^l = \max_{1 \leq l \leq L} \left( \left( B_D^l C_\sigma + B_P B_H^l B_D^l L_\sigma B_W \right) B_H^l B_{\Delta P}^l + B_P B_H^l C_\sigma B_{\Delta D}^l \right.$$

$$\left. + \left( B_D^l C_\sigma + B_P B_H^l B_D^l L_\sigma B_W \right) B_P B_{\Delta H}^l \right), \tag{12}$$

$$B_{\Delta G}^f = \max_{1 \le l \le L} \left( \left( B_D^f C_\sigma + B_P B_H^f B_D^f L_\sigma B_W \right) B_H^f B_{\Delta P}^f + B_P B_H^f C_\sigma B_{\Delta D}^f \right.$$

$$\left. + \left( B_D^f C_\sigma + B_P B_H^f B_D^f L_\sigma B_W \right) B_P B_{\Delta H}^f \right) \tag{13}$$

*Proof.*

$$\left\| \widetilde{\boldsymbol{G}}_{local}^{(l),m} - \boldsymbol{G}_{local}^{(l),m} \right\|_F$$

$$= \left\| \left[ \widetilde{\boldsymbol{P}}_{local}^{(l),m} \widetilde{\boldsymbol{H}}_{local}^{(l-1),m} \right]^\top \widetilde{\boldsymbol{D}}_{local}^{(l),m} \circ \nabla\sigma \left( \widetilde{\boldsymbol{Z}}_{local}^{(l),m} \right) - \left[ \boldsymbol{P}_{local}^{(l),m} \boldsymbol{H}_{local}^{(l-1),m} \right]^\top \boldsymbol{D}_{local}^{(l),m} \circ \nabla\sigma \left( \boldsymbol{Z}_{local}^{(l),m} \right) \right\|_F$$

$$\le \left\| \left[ \widetilde{\boldsymbol{P}}_{local}^{(l),m} \widetilde{\boldsymbol{H}}_{local}^{(l-1),m} \right]^\top \widetilde{\boldsymbol{D}}_{local}^{(l),m} \circ \nabla\sigma \left( \widetilde{\boldsymbol{Z}}_{local}^{(l),m} \right) - \left[ \boldsymbol{P}_{local}^{(l),m} \boldsymbol{H}_{local}^{(l-1),m} \right]^\top \widetilde{\boldsymbol{D}}_{local}^{(l),m} \circ \nabla\sigma \left( \widetilde{\boldsymbol{Z}}_{local}^{(l),m} \right) \right\|_F$$

$$+ \left\| \left[ \boldsymbol{P}_{local}^{(l),m} \boldsymbol{H}_{local}^{(l-1),m} \right]^\top \widetilde{\boldsymbol{D}}_{local}^{(l),m} \circ \nabla\sigma \left( \widetilde{\boldsymbol{Z}}_{local}^{(l),m} \right) - \left[ \boldsymbol{P}_{local}^{(l),m} \boldsymbol{H}_{local}^{(l-1),m} \right]^\top \boldsymbol{D}_{local}^{(l),m} \circ \nabla\sigma \left( \widetilde{\boldsymbol{Z}}_{local}^{(l),m} \right) \right\|_F$$

$$+ \left\| \left[ \boldsymbol{P}_{local}^{(l),m} \boldsymbol{H}_{local}^{(l-1),m} \right]^\top \boldsymbol{D}_{local}^{(l),m} \circ \nabla\sigma \left( \widetilde{\boldsymbol{Z}}_{local}^{(l),m} \right) - \left[ \boldsymbol{P}_{local}^{(l),m} \boldsymbol{H}_{local}^{(l-1),m} \right]^\top \boldsymbol{D}_{local}^{(l),m} \circ \nabla\sigma \left( \boldsymbol{Z}_{local}^{(l),m} \right) \right\|_F$$

$$\overset{(a)}{\le} B_D^l C_\sigma \left\| \widetilde{\boldsymbol{P}}_{local}^{(l),m} \widetilde{\boldsymbol{H}}_{local}^{(l-1),m} - \boldsymbol{P}_{local}^{(l),m} \boldsymbol{H}_{local}^{(l-1),m} \right\|_F + B_P B_H^l C_\sigma \left\| \widetilde{\boldsymbol{D}}_{local}^{(l),m} - \boldsymbol{D}_{local}^{(l),m} \right\|_F$$

$$+ B_P B_H^l B_D^l \left\| \nabla\sigma \left( \widetilde{\boldsymbol{Z}}_{local}^{(l),m} \right) - \nabla\sigma \left( \boldsymbol{Z}_{local}^{(l),m} \right) \right\|_F$$

$$\overset{(b)}{\le} B_D^l C_\sigma \left\| \widetilde{\boldsymbol{P}}_{local}^{(l),m} \widetilde{\boldsymbol{H}}_{local}^{(l-1),m} - \boldsymbol{P}_{local}^{(l),m} \boldsymbol{H}_{local}^{(l-1),m} \right\|_F + B_P B_H^l C_\sigma \left\| \widetilde{\boldsymbol{D}}_{local}^{(l),m} - \boldsymbol{D}_{local}^{(l),m} \right\|_F$$

$$+ B_P B_H^l B_D^l L_\sigma B_W \left\| \widetilde{\boldsymbol{P}}_{local}^{(l),m} \widetilde{\boldsymbol{H}}_{local}^{(l-1),m} - \boldsymbol{P}_{local}^{(l),m} \boldsymbol{H}_{local}^{(l-1),m} \right\|_F$$

$$\le \left( B_D^l C_\sigma + B_P B_H^l B_D^l L_\sigma B_W \right) \left\| \widetilde{\boldsymbol{P}}_{local}^{(l),m} \widetilde{\boldsymbol{H}}_{local}^{(l-1),m} - \boldsymbol{P}_{local}^{(l),m} \widetilde{\boldsymbol{H}}_{local}^{(l-1),m} \right\|_F$$

$$+ \left( B_D^l C_\sigma + B_P B_H^l B_D^l L_\sigma B_W \right) \left\| \boldsymbol{P}_{local}^{(l),m} \widetilde{\boldsymbol{H}}_{local}^{(l-1),m} - \boldsymbol{P}_{local}^{(l),m} \boldsymbol{H}_{local}^{(l-1),m} \right\|_F$$

$$+ B_P B_H^l C_\sigma \left\| \widetilde{\boldsymbol{D}}_{local}^{(l),m} - \boldsymbol{D}_{local}^{(l),m} \right\|_F$$

$$\overset{(c)}{\le} \left( B_D^l C_\sigma + B_P B_H^l B_D^l L_\sigma B_W \right) B_H^l \left\| \widetilde{\boldsymbol{P}}_{local}^{(l),m} - \boldsymbol{P}_{local}^{(l),m} \right\|_F + B_P B_H^l C_\sigma \left\| \widetilde{\boldsymbol{D}}_{local}^{(l),m} - \boldsymbol{D}_{local}^{(l),m} \right\|_F$$

$$+ \left( B_D^l C_\sigma + B_P B_H^l B_D^l L_\sigma B_W \right) B_P \left\| \widetilde{\boldsymbol{H}}_{local}^{(l-1),m} - \boldsymbol{H}_{local}^{(l-1),m} \right\|_F$$

$$\overset{(d)}{\le} \left( B_D^l C_\sigma + B_P B_H^l B_D^l L_\sigma B_W \right) B_H^l B_{\Delta P}^l + B_P B_H^l C_\sigma B_{\Delta D}^l$$

$$+ \left( B_D^l C_\sigma + B_P B_H^l B_D^l L_\sigma B_W \right) B_P B_{\Delta H}^l$$

$$\le \max_{1 \le l \le L} \left( \left( B_D^l C_\sigma + B_P B_H^l B_D^l L_\sigma B_W \right) B_H^l B_{\Delta P}^l + B_P B_H^l C_\sigma B_{\Delta D}^l \right.$$

$$\left. + \left( B_D^l C_\sigma + B_P B_H^l B_D^l L_\sigma B_W \right) B_P B_{\Delta H}^l \right) := B_{\Delta G}^l,$$

where (a) follows from Assumptions 5.2 and 5.3 and Lemma H.3, (b) is because of Assumptions 5.2 and 5.3, (c) uses Assumption 5.3 and Lemma H.3, and (d) results from Lemma H.4 and Proposition H.1.

When client $m$ performs local training with only its local data, the error between the stochastic gradient and the full-gradient can be bounded as:

$$\left\| \nabla F_{local}^m \left( \boldsymbol{\theta}^m \right) - \nabla \widetilde{F}_{local}^m \left( \boldsymbol{\theta}^m \right) \right\|_F = \sum_{l=1}^{L} \left\| \boldsymbol{G}_{local}^{(l),m} - \widetilde{\boldsymbol{G}}_{local}^{(l),m} \right\|_F \le L B_{\Delta G}^l.$$

$$\left\| \widetilde{\boldsymbol{G}}_{full}^{(l),m} - \boldsymbol{G}_{full}^{(l),m} \right\|_F$$

$$= \left\| \left[ \widetilde{\boldsymbol{P}}_{local}^{(l),m} \widetilde{\boldsymbol{H}}_{local}^{(l-1),m} + \widetilde{\boldsymbol{P}}_{remote}^{(l),m} \widetilde{\boldsymbol{H}}_{remote}^{(l-1),m} \right]^\top \widetilde{\boldsymbol{D}}_{full}^{(l),m} \circ \nabla\sigma \left( \widetilde{\boldsymbol{Z}}_{full}^{(l),m} \right) \right.$$

$$\left. - \left[ \boldsymbol{P}_{full}^{(l),m} \boldsymbol{H}_{full}^{(l-1),m} \right]^\top \boldsymbol{D}_{full}^{(l),m} \circ \nabla\sigma \left( \boldsymbol{Z}_{full}^{(l),m} \right) \right\|_F$$

$$\leq \left\| \left[ \widetilde{\boldsymbol{P}}_{full}^{(l),m} \widetilde{\boldsymbol{H}}_{full}^{(l-1),m} \right]^\top \widetilde{\boldsymbol{D}}_{full}^{(l),m} \circ \nabla\sigma \left( \widetilde{\boldsymbol{Z}}_{full}^{(l),m} \right) - \left[ \boldsymbol{P}_{full}^{(l),m} \boldsymbol{H}_{full}^{(l-1),m} \right]^\top \widetilde{\boldsymbol{D}}_{full}^{(l),m} \circ \nabla\sigma \left( \widetilde{\boldsymbol{Z}}_{full}^{(l),m} \right) \right\|_F$$

$$+ \left\| \left[ \boldsymbol{P}_{full}^{(l),m} \boldsymbol{H}_{full}^{(l-1),m} \right]^\top \widetilde{\boldsymbol{D}}_{full}^{(l),m} \circ \nabla\sigma \left( \widetilde{\boldsymbol{Z}}_{full}^{(l),m} \right) - \left[ \boldsymbol{P}_{full}^{(l),m} \boldsymbol{H}_{full}^{(l-1),m} \right]^\top \boldsymbol{D}_{full}^{(l),m} \circ \nabla\sigma \left( \widetilde{\boldsymbol{Z}}_{full}^{(l),m} \right) \right\|_F$$

$$+ \left\| \left[ \boldsymbol{P}_{full}^{(l),m} \boldsymbol{H}_{full}^{(l-1),m} \right]^\top \boldsymbol{D}_{full}^{(l),m} \circ \nabla\sigma \left( \widetilde{\boldsymbol{Z}}_{full}^{(l),m} \right) - \left[ \boldsymbol{P}_{full}^{(l),m} \boldsymbol{H}_{full}^{(l-1),m} \right]^\top \boldsymbol{D}_{full}^{(l),m} \circ \nabla\sigma \left( \boldsymbol{Z}_{full}^{(l),m} \right) \right\|_F$$

$$\overset{(a)}{\leq} B_D^f C_\sigma \left\| \widetilde{\boldsymbol{P}}_{full}^{(l),m} \widetilde{\boldsymbol{H}}_{full}^{(l-1),m} - \boldsymbol{P}_{full}^{(l),m} \boldsymbol{H}_{full}^{(l-1),m} \right\|_F + B_P B_H^f C_\sigma \left\| \widetilde{\boldsymbol{D}}_{full}^{(l),m} - \boldsymbol{D}_{full}^{(l),m} \right\|_F$$

$$+ B_P B_H^f B_D^f \left\| \nabla\sigma \left( \widetilde{\boldsymbol{Z}}_{full}^{(l),m} \right) - \nabla\sigma \left( \boldsymbol{Z}_{full}^{(l),m} \right) \right\|_F$$

$$\overset{(b)}{\leq} B_D^f C_\sigma \left\| \widetilde{\boldsymbol{P}}_{full}^{(l),m} \widetilde{\boldsymbol{H}}_{full}^{(l-1),m} - \boldsymbol{P}_{full}^{(l),m} \boldsymbol{H}_{full}^{(l-1),m} \right\|_F + B_P B_H^f C_\sigma \left\| \widetilde{\boldsymbol{D}}_{full}^{(l),m} - \boldsymbol{D}_{full}^{(l),m} \right\|_F$$

$$+ B_P B_H^f B_D^f L_\sigma B_W \left\| \widetilde{\boldsymbol{P}}_{full}^{(l),m} \widetilde{\boldsymbol{H}}_{full}^{(l-1),m} - \boldsymbol{P}_{full}^{(l),m} \boldsymbol{H}_{full}^{(l-1),m} \right\|_F$$

$$\leq \left( B_D^f C_\sigma + B_P B_H^f B_D^f L_\sigma B_W \right) \left\| \widetilde{\boldsymbol{P}}_{full}^{(l),m} \widetilde{\boldsymbol{H}}_{full}^{(l-1),m} - \boldsymbol{P}_{full}^{(l),m} \widetilde{\boldsymbol{H}}_{full}^{(l-1),m} \right\|_F$$

$$+ \left( B_D^f C_\sigma + B_P B_H^f B_D^f L_\sigma B_W \right) \left\| \boldsymbol{P}_{full}^{(l),m} \widetilde{\boldsymbol{H}}_{full}^{(l-1),m} - \boldsymbol{P}_{full}^{(l),m} \boldsymbol{H}_{full}^{(l-1),m} \right\|_F$$

$$+ B_P B_H^f C_\sigma \left\| \widetilde{\boldsymbol{D}}_{full}^{(l),m} - \boldsymbol{D}_{full}^{(l),m} \right\|_F$$

$$\overset{(c)}{\leq} \left( B_D^f C_\sigma + B_P B_H^f B_D^f L_\sigma B_W \right) B_H^f \left\| \widetilde{\boldsymbol{P}}_{full}^{(l),m} - \boldsymbol{P}_{full}^{(l),m} \right\|_F + B_P B_H^f C_\sigma \left\| \widetilde{\boldsymbol{D}}_{full}^{(l),m} - \boldsymbol{D}_{full}^{(l),m} \right\|_F$$

$$+ \left( B_D^f C_\sigma + B_P B_H^f B_D^f L_\sigma B_W \right) B_P \left\| \widetilde{\boldsymbol{H}}_{full}^{(l-1),m} - \boldsymbol{H}_{full}^{(l-1),m} \right\|_F$$

$$\overset{(d)}{\leq} \left( B_D^f C_\sigma + B_P B_H^f B_D^f L_\sigma B_W \right) B_H^f B_{\Delta P}^f + B_P B_H^f C_\sigma B_{\Delta D}^f$$

$$+ \left( B_D^f C_\sigma + B_P B_H^f B_D^f L_\sigma B_W \right) B_P B_{\Delta H}^f$$

$$\leq \max_{1 \leq l \leq L} \left( \left( B_D^f C_\sigma + B_P B_H^f B_D^f L_\sigma B_W \right) B_H^f B_{\Delta P}^f + B_P B_H^f C_\sigma B_{\Delta D}^f \right.$$

$$+ \left. \left( B_D^f C_\sigma + B_P B_H^f B_D^f L_\sigma B_W \right) B_P B_{\Delta H}^f \right) := B_{\Delta G}^f,$$

where (a) results from Assumptions 5.2 and 5.3 and Lemma H.3, (b) uses Assumptions 5.2 and 5.3, (c) is due to Assumption 5.3 and Lemma H.3, and (d) is because of Lemma H.4 and Proposition H.1.

When client $m$ conducts cross-client training using its local data and the cross-client neighbors, the error between the stochastic gradient and the full-gradient can be bounded as:

$$\left\| \nabla F_{full}^m (\boldsymbol{\theta}^m) - \nabla \widetilde{F}_{full}^m (\boldsymbol{\theta}^m) \right\|_F = \sum_{l=1}^L \left\| \boldsymbol{G}_{full}^{(l),m} - \widetilde{\boldsymbol{G}}_{full}^{(l),m} \right\|_F \leq L B_{\Delta G}^f.$$

$\square$

**Lemma H.7.** *Under Assumptions 5.1–5.3, the error between the full gradient computed with both the local graph data and the cross-client neighbors and the full gradient computed with only the local graph data is upper-bounded as follows:*

$$\left\| \nabla F_{full}^m (\boldsymbol{\theta}^m) - \nabla F_{local}^m (\boldsymbol{\theta}^m) \right\|_F \leq L B_{\Delta G}^r,$$

*where*

$$B_{\Delta G}^r = \max_{1 \leq l \leq L} \left( \left( B_D^l C_\sigma + B_P B_H^f B_D^f L_\sigma B_W \right) B_P B_{\Delta H}^r + B_P B_H^f C_\sigma B_{\Delta D}^r \right.$$

$$+ \left( B_D^l C_\sigma + B_P B_H^f B_D^f L_\sigma B_W \right) B_H^f B_P \right) \tag{14}$$

*Proof.*

$$\left\| \boldsymbol{G}_{local}^{(l),m} - \boldsymbol{G}_{full}^{(l),m} \right\|_F$$

$$= \left\| \left[ \boldsymbol{P}_{local}^{(l),m} \boldsymbol{H}_{local}^{(l-1),m} \right]^\top \boldsymbol{D}_{local}^{(l),m} \circ \nabla\sigma\left( \boldsymbol{Z}_{local}^{(l),m} \right) - \left[ \boldsymbol{P}_{full}^{(l),m} \boldsymbol{H}_{full}^{(l-1),m} \right]^\top \boldsymbol{D}_{full}^{(l),m} \circ \nabla\sigma\left( \boldsymbol{Z}_{full}^{(l),m} \right) \right\|_F$$

$$\leq \left\| \left[ \boldsymbol{P}_{local}^{(l),m} \boldsymbol{H}_{local}^{(l-1),m} \right]^\top \boldsymbol{D}_{local}^{(l),m} \circ \nabla\sigma\left( \boldsymbol{Z}_{local}^{(l),m} \right) - \left[ \boldsymbol{P}_{full}^{(l),m} \boldsymbol{H}_{full}^{(l-1),m} \right]^\top \boldsymbol{D}_{local}^{(l),m} \circ \nabla\sigma\left( \boldsymbol{Z}_{local}^{(l),m} \right) \right\|_F$$

$$+ \left\| \left[ \boldsymbol{P}_{full}^{(l),m} \boldsymbol{H}_{full}^{(l-1),m} \right]^\top \boldsymbol{D}_{local}^{(l),m} \circ \nabla\sigma\left( \boldsymbol{Z}_{local}^{(l),m} \right) - \left[ \boldsymbol{P}_{full}^{(l),m} \boldsymbol{H}_{full}^{(l-1),m} \right]^\top \boldsymbol{D}_{full}^{(l),m} \circ \nabla\sigma\left( \boldsymbol{Z}_{local}^{(l),m} \right) \right\|_F$$

$$+ \left\| \left[ \boldsymbol{P}_{full}^{(l),m} \boldsymbol{H}_{full}^{(l-1),m} \right]^\top \boldsymbol{D}_{full}^{(l),m} \circ \nabla\sigma\left( \boldsymbol{Z}_{local}^{(l),m} \right) - \left[ \boldsymbol{P}_{full}^{(l),m} \boldsymbol{H}_{full}^{(l-1),m} \right]^\top \boldsymbol{D}_{full}^{(l),m} \circ \nabla\sigma\left( \boldsymbol{Z}_{full}^{(l),m} \right) \right\|_F$$

$$\overset{(a)}{\leq} B_D^l C_\sigma \left\| \boldsymbol{P}_{local}^{(l),m} \boldsymbol{H}_{local}^{(l-1),m} - \boldsymbol{P}_{full}^{(l),m} \boldsymbol{H}_{full}^{(l-1),m} \right\|_F + B_P B_H^f C_\sigma \left\| \boldsymbol{D}_{local}^{(l),m} - \boldsymbol{D}_{full}^{(l),m} \right\|_F$$

$$+ B_P B_H^f B_D^f \left\| \nabla\sigma\left( \boldsymbol{Z}_{local}^{(l),m} \right) - \nabla\sigma\left( \boldsymbol{Z}_{full}^{(l),m} \right) \right\|_F$$

$$\overset{(b)}{\leq} B_D^l C_\sigma \left\| \boldsymbol{P}_{local}^{(l),m} \boldsymbol{H}_{local}^{(l-1),m} - \boldsymbol{P}_{full}^{(l),m} \boldsymbol{H}_{full}^{(l-1),m} \right\|_F + B_P B_H^f C_\sigma \left\| \boldsymbol{D}_{local}^{(l),m} - \boldsymbol{D}_{full}^{(l),m} \right\|_F$$

$$+ B_P B_H^f B_D^f L_\sigma B_W \left\| \boldsymbol{P}_{local}^{(l),m} \boldsymbol{H}_{local}^{(l-1),m} - \boldsymbol{P}_{full}^{(l),m} \boldsymbol{H}_{full}^{(l-1),m} \right\|_F$$

$$\leq \left( B_D^l C_\sigma + B_P B_H^f B_D^f L_\sigma B_W \right) \left\| \boldsymbol{P}_{local}^{(l),m} \boldsymbol{H}_{local}^{(l-1),m} - \boldsymbol{P}_{local}^{(l),m} \boldsymbol{H}_{full}^{(l-1),m} \right\|_F$$

$$+ \left( B_D^l C_\sigma + B_P B_H^f B_D^f L_\sigma B_W \right) \left\| \boldsymbol{P}_{local}^{(l),m} \boldsymbol{H}_{full}^{(l-1),m} - \boldsymbol{P}_{full}^{(l),m} \boldsymbol{H}_{full}^{(l-1),m} \right\|_F$$

$$+ B_P B_H^f C_\sigma \left\| \boldsymbol{D}_{local}^{(l),m} - \boldsymbol{D}_{full}^{(l),m} \right\|_F$$

$$\overset{(c)}{\leq} \left( B_D^l C_\sigma + B_P B_H^f B_D^f L_\sigma B_W \right) B_P \left\| \boldsymbol{H}_{local}^{(l-1),m} - \boldsymbol{H}_{full}^{(l-1),m} \right\|_F$$

$$+ \left( B_D^l C_\sigma + B_P B_H^f B_D^f L_\sigma B_W \right) B_H^f \left\| \boldsymbol{P}_{local}^{(l),m} - \boldsymbol{P}_{full}^{(l),m} \right\|_F + B_P B_H^f C_\sigma \left\| \boldsymbol{D}_{local}^{(l),m} - \boldsymbol{D}_{full}^{(l),m} \right\|_F$$

$$\overset{(d)}{\leq} \left( B_D^l C_\sigma + B_P B_H^f B_D^f L_\sigma B_W \right) B_P B_{\Delta H}^r + B_P B_H^f C_\sigma B_{\Delta D}^r$$

$$+ \left( B_D^l C_\sigma + B_P B_H^f B_D^f L_\sigma B_W \right) B_H^f B_P$$

$$\leq \max_{1 \leq l \leq L} \left( \left( B_D^l C_\sigma + B_P B_H^f B_D^f L_\sigma B_W \right) B_P B_{\Delta H}^r + B_P B_H^f C_\sigma B_{\Delta D}^r \right.$$

$$\left. + \left( B_D^l C_\sigma + B_P B_H^f B_D^f L_\sigma B_W \right) B_H^f B_P \right) = B_{\Delta G}^r,$$

where (a) is because of Assumptions 5.2 and 5.3 and Lemma H.3, (b) uses Assumptions 5.2 and 5.3, (c) follow from Assumption 5.3 and Lemma H.3, and (d) results from Assumption 5.3 and Lemma H.5.

The error between the full gradient computed with both the local graph data and the cross-client neighbors and the full gradient computed with only the local graph data is bounded as follows:

$$\left\| \nabla F_{full}^m \left( \boldsymbol{\theta}^m \right) - \nabla F_{local}^m \left( \boldsymbol{\theta}^m \right) \right\|_F = \sum_{l=1}^L \left\| \boldsymbol{G}_{local}^{(l),m} - \boldsymbol{G}_{full}^{(l),m} \right\|_F \leq L B_{\Delta G}^r.$$

$\square$

### H.4 MAIN PROOF OF THEOREM 5.6

**Theorem H.8.** *Under Assumptions 5.1–5.3, choose step-size $\alpha = \min\left\{\sqrt{M}/\sqrt{T}, 1/L_F\right\}$, where $L_F$ is the smoothness constant given in Lemma H.2. The output of* Swift-FedGNN *with a L-layer GNN satisfies:*

$$\frac{1}{T}\sum_{t=0}^{T-1}\|\nabla\mathcal{L}(\boldsymbol{\theta}_t)\|^2 \leq \frac{2}{\sqrt{MT}}(\mathcal{L}(\boldsymbol{\theta}_0) - \mathcal{L}(\boldsymbol{\theta}^*)) + \left(1 - \frac{K}{IM}\right)L^2\left(B_{\Delta G}^l + B_{\Delta G}^r\right)^2 + \frac{K}{IM}L^2(B_{\Delta G}^f)^2.$$

*Proof.*

$$\mathcal{L}(\boldsymbol{\theta}_{t+1}) - \mathcal{L}(\boldsymbol{\theta}_t)$$

$$\overset{(a)}{\leq} \langle\nabla\mathcal{L}(\boldsymbol{\theta}_t), \boldsymbol{\theta}_{t+1} - \boldsymbol{\theta}_t\rangle + \frac{L_F}{2}\|\boldsymbol{\theta}_{t+1} - \boldsymbol{\theta}_t\|^2$$

$$\overset{(b)}{=} -\alpha\left\langle\nabla\mathcal{L}(\boldsymbol{\theta}_t), \frac{1}{M}\sum_{m\in\mathcal{M}}\nabla\widetilde{F}^m(\boldsymbol{\theta}_t^m)\right\rangle + \frac{L_F}{2}\alpha^2\left\|\frac{1}{M}\sum_{m\in\mathcal{M}}\nabla\widetilde{F}^m(\boldsymbol{\theta}_t^m)\right\|^2$$

$$\overset{(c)}{=} -\frac{\alpha}{2}\left\|\nabla\mathcal{L}(\boldsymbol{\theta}_t)\right\|^2 - \frac{\alpha}{2}\left\|\frac{1}{M}\sum_{m\in\mathcal{M}}\nabla\widetilde{F}^m(\boldsymbol{\theta}_t^m)\right\|^2 + \frac{\alpha}{2}\left\|\nabla\mathcal{L}(\boldsymbol{\theta}_t) - \frac{1}{M}\sum_{m\in\mathcal{M}}\nabla\widetilde{F}^m(\boldsymbol{\theta}_t^m)\right\|^2$$

$$+ \frac{L_F}{2}\alpha^2\left\|\frac{1}{M}\sum_{m\in\mathcal{M}}\nabla\widetilde{F}^m(\boldsymbol{\theta}_t^m)\right\|^2$$

$$= -\frac{\alpha}{2}\left\|\nabla\mathcal{L}(\boldsymbol{\theta}_t)\right\|^2 - \frac{\alpha}{2}\left\|\frac{1}{M}\sum_{m\in\mathcal{M}}\nabla\widetilde{F}^m(\boldsymbol{\theta}_t^m)\right\|^2 + \frac{\alpha}{2}\left\|\frac{1}{M}\sum_{m\in\mathcal{M}}\left(\nabla F^m(\boldsymbol{\theta}_t^m) - \nabla\widetilde{F}^m(\boldsymbol{\theta}_t^m)\right)\right\|^2$$

$$+ \frac{L_F}{2}\alpha^2\left\|\frac{1}{M}\sum_{m\in\mathcal{M}}\nabla\widetilde{F}^m(\boldsymbol{\theta}_t^m)\right\|^2$$

$$\overset{(d)}{\leq} -\frac{\alpha}{2}\left\|\nabla\mathcal{L}(\boldsymbol{\theta}_t)\right\|^2 - \frac{\alpha}{2}\left\|\frac{1}{M}\sum_{m\in\mathcal{M}}\nabla\widetilde{F}^m(\boldsymbol{\theta}_t^m)\right\|^2 + \frac{\alpha}{2}\frac{1}{M}\sum_{m\in\mathcal{M}}\left\|\nabla F^m(\boldsymbol{\theta}_t^m) - \nabla\widetilde{F}^m(\boldsymbol{\theta}_t^m)\right\|^2$$

$$+ \frac{L_F}{2}\alpha^2\left\|\frac{1}{M}\sum_{m\in\mathcal{M}}\nabla\widetilde{F}^m(\boldsymbol{\theta}_t^m)\right\|^2$$

$$\overset{(e)}{\leq} -\frac{\alpha}{2}\left\|\nabla\mathcal{L}(\boldsymbol{\theta}_t)\right\|^2 + \frac{\alpha}{2}\frac{1}{M}\sum_{m\in\mathcal{M}}\left\|\nabla F^m(\boldsymbol{\theta}_t^m) - \nabla\widetilde{F}^m(\boldsymbol{\theta}_t^m)\right\|^2, \tag{15}$$

where (a) follows from Lemma H.2, (b) is because of the update rule in Swift-FedGNN, (c) uses $\langle\boldsymbol{x}, \boldsymbol{y}\rangle = \frac{1}{2}\|\boldsymbol{x}\|^2 + \frac{1}{2}\|\boldsymbol{y}\|^2 - \frac{1}{2}\|\boldsymbol{x} - \boldsymbol{y}\|^2$, (d) utilizes $\|\sum_{i=1}^n\boldsymbol{x}_i\|^2 \leq n\sum_{i=1}^n\|\boldsymbol{x}_i\|^2$, and (e) is due to the choice of $\alpha \leq 1/L_F$.

When $t \in [(n_t - 1)I + 1, n_t I - 1] \cap \mathbb{Z}$, where $n_t = \{1, 2, \cdots\}$, Swift-FedGNN conducts local training for all clients $m \in \mathcal{M}$. Thus,

$$\left\|\nabla F^m(\boldsymbol{\theta}_t^m) - \nabla\widetilde{F}^m(\boldsymbol{\theta}_t^m)\right\| = \left\|\nabla F_{full}^m(\boldsymbol{\theta}_t^m) - \nabla\widetilde{F}_{local}^m(\boldsymbol{\theta}_t^m)\right\|$$

$$\leq \left\|\nabla F_{full}^m(\boldsymbol{\theta}_t^m) - \nabla F_{local}^m(\boldsymbol{\theta}_t^m)\right\| + \left\|\nabla F_{local}^m(\boldsymbol{\theta}_t^m) - \nabla\widetilde{F}_{local}^m(\boldsymbol{\theta}_t^m)\right\|$$

$$\overset{(a)}{\leq} LB_{\Delta G}^r + LB_{\Delta G}^l, \tag{16}$$

where (a) follows from Lemmas H.6 and H.7.

When $t = n_t I$, where $n_t = \{1, 2, \cdots\}$, Swift-FedGNN performs local training for clients $m \in \mathcal{M}\backslash\mathcal{K}$, and thus the inequality (16) holds for these clients. The randomly sampled clients $m \in \mathcal{K}$ conduct cross-client training, and thus

$$\left\|\nabla F^m(\boldsymbol{\theta}_t^m) - \nabla\widetilde{F}^m(\boldsymbol{\theta}_t^m)\right\| = \left\|\nabla F_{full}^m(\boldsymbol{\theta}_t^m) - \nabla\widetilde{F}_{full}^m(\boldsymbol{\theta}_t^m)\right\| \overset{(a)}{\leq} LB_{\Delta G}^f,$$

where (a) uses Lemma H.6.

Telescoping (15) from $i = (n_t - 1) I + 1$ to $n_t I$, we have

$$
\sum_{i=(n_t-1)I+1}^{n_t I} \left( \mathcal{L} \left( \boldsymbol{\theta}_{i+1} \right) - \mathcal{L} \left( \boldsymbol{\theta}_i \right) \right)
$$

$$
\leq -\frac{\alpha}{2} \sum_{i=(n_t-1)I+1}^{n_t I} \left\| \nabla \mathcal{L} \left( \boldsymbol{\theta}_i \right) \right\|^2 + \frac{\alpha}{2}(I-1)L^2 \left( B_{\Delta G}^l + B_{\Delta G}^r \right)^2 + \frac{\alpha}{2M} K L^2 \left( B_{\Delta G}^f \right)^2
$$

$$
+ \frac{\alpha}{2M}(M-K)L^2 \left( B_{\Delta G}^l + B_{\Delta G}^r \right)^2 .
$$

Choosing $T = n_t I$ yields

$$
\sum_{t=0}^{T-1} \left( \mathcal{L} \left( \boldsymbol{\theta}_{t+1} \right) - \mathcal{L} \left( \boldsymbol{\theta}_t \right) \right)
$$

$$
\leq -\frac{\alpha}{2} \sum_{t=0}^{T-1} \left\| \nabla \mathcal{L} \left( \boldsymbol{\theta}_t \right) \right\|^2 + \frac{\alpha}{2}(T-n_t)L^2 \left( B_{\Delta G}^l + B_{\Delta G}^r \right)^2 + n_t \frac{\alpha}{2M} K L^2 \left( B_{\Delta G}^f \right)^2
$$

$$
+ n_t \frac{\alpha}{2M}(M-K)L^2 \left( B_{\Delta G}^l + B_{\Delta G}^r \right)^2 .
$$

Rearranging the terms and multiplying both sides by $2/\alpha$, we get

$$
\sum_{t=0}^{T-1} \left\| \nabla \mathcal{L} \left( \boldsymbol{\theta}_t \right) \right\|^2
$$

$$
\leq \frac{2}{\alpha} \sum_{t=0}^{T-1} \left( \mathcal{L} \left( \boldsymbol{\theta}_t \right) - \mathcal{L} \left( \boldsymbol{\theta}_{t+1} \right) \right) + (T-n_t)L^2 \left( B_{\Delta G}^l + B_{\Delta G}^r \right)^2 + \frac{n_t}{M} K L^2 \left( B_{\Delta G}^f \right)^2
$$

$$
+ \frac{n_t}{M}(M-K)L^2 \left( B_{\Delta G}^l + B_{\Delta G}^r \right)^2 .
$$

Dividing both sides by $T$ and choosing $\alpha = \sqrt{M}/\sqrt{T}$ completes the proof of Theorem 5.6.

$\square$

# I  THEORETICAL ANALYSIS EXTENSIONS FOR GRAPHSAGE AND GIN

While our theoretical analysis is presented under the GCN architecture for mathematical tractability, the core convergence results of Swift-FedGNN extend naturally to a broader class of element-wise operation-based GNNs, including GraphSAGE and GIN. In particular, our convergence bounds remain applicable to these models under similar assumptions.

The main challenge in extending the theoretical analysis to GraphSAGE and GIN lies in handling non-linear and heterogeneous aggregation functions, which are more prominent in GraphSAGE (e.g., max-pooling, LSTM) and GIN (e.g., MLP-based injective updates). These functions introduce additional sources of nonlinearity and variance in the layer-wise error propagation, making it harder to tightly bound the bias and variance of the resulting stochastic gradients.

Below, we describe the respective update rules and outline the required modifications to adapt our proof strategy for GraphSAGE and GIN.

## I.1  UPDATE RULES FOR GRAPHSAGE AND GIN

**1) GraphSAGE:** The propagation matrices for GraphSAGE are given by $\boldsymbol{K}_{local}^m = \boldsymbol{D}_m^{-1} \hat{A}_{local}^m$ and $\boldsymbol{K}_{remote}^m = \boldsymbol{D}_m^{-1} \hat{A}_{remote}^m$. Similar to GCN, when client $m$ trains using only the

local graph data, the update rule for GraphSAGE (*i.e.*, Eq. (3) and (4)) is: $\widetilde{\boldsymbol{H}}_t^{(l),m} = \sigma\left(\left[\widetilde{\boldsymbol{H}}_t^{(l-1),m} \parallel \widetilde{\boldsymbol{K}}_{local}^{(l),m} \widetilde{\boldsymbol{H}}_{local}^{(l-1),m}\right] \boldsymbol{W}_t^{(l),m}\right)$. When client $m$ trains based on both the local graph data and the cross-client neighbors, the update rule for GraphSAGE (*i.e.*, Eq. (5)–(8)) becomes $\widetilde{\boldsymbol{H}}_t^{(l),m} = \sigma\left(\left[\widetilde{\boldsymbol{H}}_t^{(l-1),m} \parallel \left(\widetilde{\boldsymbol{K}}_{local}^{(l),m} \widetilde{\boldsymbol{H}}_{local}^{(l-1),m} + \widetilde{\boldsymbol{K}}_{remote}^{(l),m} \widetilde{\boldsymbol{H}}_{remote}^{(l-1),m}\right)\right] \boldsymbol{W}_t^{(l),m}\right)$.

**2) GIN:** The propagation matrices for GIN are defined as $\boldsymbol{S}_{local}^{(l),m} = \boldsymbol{A}_{local}^m + \left(1 + \epsilon^{(l),m}\right)\boldsymbol{I}$ and $\boldsymbol{S}_{remote}^{(l),m} = \boldsymbol{A}_{remote}^m + \left(1 + \epsilon^{(l),m}\right)\boldsymbol{I}$. When client $m$ trains on local graph data only, the update rule for GIN (*i.e.*, Eq. (3) and (4)) is: $\widetilde{\boldsymbol{H}}_t^{(l),m} = \text{MLP}^{(l),m}\left(\widetilde{\boldsymbol{S}}_{local}^{(l),m} \widetilde{\boldsymbol{H}}_{local}^{(l-1),m}\right)$. When client $m$ trains using both the local graph data and the cross-client neighbors, the update rule for GIN (*i.e.*, Eq. (5)–(8)) becomes: $\widetilde{\boldsymbol{H}}_t^{(l),m} = \text{MLP}^{(l),m}\left(\widetilde{\boldsymbol{S}}_{local}^{(l),m} \widetilde{\boldsymbol{H}}_{local}^{(l-1),m} + \widetilde{\boldsymbol{S}}_{remote}^{(l),m} \widetilde{\boldsymbol{H}}_{remote}^{(l-1),m}\right)$.

### I.2 Proof Sketch: Extending Theoretical Analysis to GraphSAGE and GIN

Extending the convergence analysis in Theorem 5.6 to GraphSAGE and GIN follows a similar proof strategy as that for GCN, with the GCN-specific lemmas replaced by their respective counterparts for GraphSAGE or GIN.

**1) Modified Bias Bounding Strategy:** The original convergence proof (Theorem 5.6) relies on bounding the gradient bias introduced by (i) stochastic neighbor sampling and (ii) the absence of cross-client neighbors. These bounds are formalized in Lemmas 5.4 and 5.5, supported by Lemmas H.3–H.5, all of which are based on GCN-specific updates.

To generalize the analysis, we replace the GCN-specific update rules with the corresponding rules for GraphSAGE or GIN, and re-derive the associated bounds in Lemmas 5.4 and 5.5 and their supporting lemmas (Lemmas H.3–H.5). This yields modified upper bounds on the gradient bias, where the constants depend on the respective GNN architectures.

**Lemma I.1.** *Under Assumptions 5.1–5.3, the errors between the stochastic gradients and the full gradients are bounded as follows:*

$$\left\|\nabla F_{local}^m\left(\boldsymbol{\theta}^m\right) - \nabla \widetilde{F}_{local}^m\left(\boldsymbol{\theta}^m\right)\right\|_F \leq C_{\Delta G}^l, \quad \left\|\nabla F_{full}^m\left(\boldsymbol{\theta}^m\right) - \nabla \widetilde{F}_{full}^m\left(\boldsymbol{\theta}^m\right)\right\|_F \leq C_{\Delta G}^f,$$

*where $C_{\Delta G}^l$ and $C_{\Delta G}^f$ are constants that depend on the respective GNN architectures (e.g., Graph-SAGE or GIN) and are positively correlated with the GNN depth.*

**Lemma I.2.** *Under Assumptions 5.1–5.3, the error between the full gradient computed with both the local graph data and the cross-client neighbors and the full gradient computed with only the local graph data is upper-bounded as follows:*

$$\left\|\nabla F_{full}^m\left(\boldsymbol{\theta}^m\right) - \nabla F_{local}^m\left(\boldsymbol{\theta}^m\right)\right\|_F \leq C_{\Delta G}^r,$$

*where $C_{\Delta G}^r$ is a constant that depends on the respective GNN architectures (e.g., GraphSAGE or GIN) and is positively correlated with the GNN depth.*

**2) Generalized Convergence Result:** By substituting the updated gradient bias bounds (Lemmas I.1 and I.2) into the main convergence proof (Theorem 5.6), we obtain the following generalized convergence result for Swift-FedGNN with GraphSAGE or GIN:

$$\frac{1}{T}\sum_{t=0}^{T-1}\|\nabla\mathcal{L}\left(\boldsymbol{\theta}_t\right)\|^2 \leq \frac{2\left(\mathcal{L}\left(\boldsymbol{\theta}_0\right) - \mathcal{L}\left(\boldsymbol{\theta}^*\right)\right)}{\sqrt{MT}} + \left(C_{\Delta G}^l + C_{\Delta G}^r\right)^2 + \frac{K}{IM}\left(\left(C_{\Delta G}^f\right)^2 - \left(C_{\Delta G}^l + C_{\Delta G}^r\right)^2\right),$$

where the residual error terms depend on the specific GNN architecture used.

This extension demonstrates that Swift-FedGNN 's convergence guarantees are not limited to GCN, but remain valid for other element-wise operation-based GNNs such as GraphSAGE and GIN under similar assumptions. Importantly, the key theoretical insights (e.g., the residual error scaling with the correction frequency $I$ and the client sampling size $K$) persist across architectures, supporting the broad applicability of our framework.

