# OpenReview forum: "Swift-FedGNN: Federated Graph Learning with Low Communication and Sample Complexities"
_ICLR.cc/2026/Conference — Submitted to ICLR 2026_

### Official Review · Reviewer_qPZs · 2025-10-29

**Soundness:** 3
**Presentation:** 2
**Contribution:** 2
**Rating:** 6
**Confidence:** 4

**Summary:**

The paper introduces Swift-FedGNN, a novel federated graph learning (FGL) framework designed to address the high communication and sampling costs inherent in training graph neural networks (GNNs) on geo-distributed, privacy-sensitive graph data. The core contribution is a hybrid training strategy that primarily relies on efficient parallel local training, punctuated by periodic, time-consuming cross-client training performed only on a randomly sampled subset of clients. To minimize communication and enhance privacy, Swift-FedGNN offloads the aggregation of intermediate node features from remote clients to the central server, thereby avoiding direct client-to-client exchange of raw graph features. The authors provide a rigorous theoretical convergence analysis, demonstrating that Swift-FedGNN achieves an $O(T^{-1/2})$ convergence rate, which matches state-of-the-art sampling-based GNN methods despite the challenging federated setting and the presence of biased stochastic gradients (whose approximation errors are bounded for the first time in this context). Extensive experiments on large-scale real-world datasets show that the proposed method significantly outperforms existing SOTA FGL baselines in efficiency, achieving up to 4x speed-up and a substantially lower total communication cost while maintaining competitive accuracy.

**Strengths:**

1. This approach constitutes a significant and non-trivial technical modification to existing DGL and FGL paradigms.

2. The quality of this research is high in both its theoretical and empirical contributions.

3. This work holds substantial practical significance as it addresses the key barrier to deploying large-scale GNN training in privacy-sensitive and resource-constrained geo-distributed environments: high communication and sampling complexities.

**Weaknesses:**

1. The paper's main claim rests on successfully balancing communication overhead and information loss, controlled by $I$ and $K$. Although Figures 7a and 7b show the impact of $I$ and $K$ on the computation-to-communication time ratio, they completely lack the corresponding analysis for the final validation accuracy. This omission prevents a full assessment of the accuracy degradation trade-off for efficiency gains, making it impossible to determine the optimal operating point for $I$ and $K$. Thus, the experiments are insufficient to fully support the claim of "preserving minimal information loss."

2. Analysis of Performance-Efficiency Trade-off for $I$ and $K$: The paper discusses the relationship between the correction frequency $I$ and the number of sampled cross-client training clients $K$ with the computation-to-communication ratio (Figure 7a, 7b), but it does not show their impact on the final model validation accuracy. Please provide experimental results showing the sensitivity of the validation accuracy as a function of $I$ and $K$ (e.g., plots of accuracy versus communication cost or wall-clock time for varying $(I, K)$ pairs). This is crucial for fully assessing the method's core trade-off and for guiding optimal hyperparameter settings in practice.

**Questions:**

Please see the above weakness.

---

> ### Author Response · Authors · 2025-11-22
>
> > **Weakness 1:** The paper's main claim rests on successfully balancing communication overhead and information loss, controlled by $I$ and $K$. Although Figures 7a and 7b show the impact of $I$ and $K$ on the computation-to-communication time ratio, they completely lack the corresponding analysis for the final validation accuracy. This omission prevents a full assessment of the accuracy degradation trade-off for efficiency gains, making it impossible to determine the optimal operating point for $I$ and $K$. Thus, the experiments are insufficient to fully support the claim of "preserving minimal information loss."
>
> **Response:** Thanks for your comment. We agree that the trade-off between communication overhead and validation accuracy is critical to assessing the effectiveness of Swift-FedGNN. As you correctly pointed out, our original Figures 7a and 7b focus on the computation-to-communication time ratio under varying correction frequency ($I$) and number of sampled clients ($K$), but did not explicitly present the corresponding accuracy trends.
>
> To address this, in this rebuttal period, we have conducted additional experiments to  validate the accuracy of Swift-FedGNN on the ogbn-products dataset under different settings of $I$ and $K$ in Tables 1 and 2, respectively. The results demonstrate that:
> * Increasing the correction interval ($I$) from 5 to 40 (Table 1) leads to only a minor accuracy degradation (from 88.91% to 88.44%), demonstrating that less frequent cross-client training still preserves model quality.
> * Decreasing the number of sampled clients ($K$) from 15 to 1 (Table 2) also results in a minor accuracy drop (from 89.22% to 88.47%), indicating that a small number of sampled clients is sufficient to maintain strong performance.
>
> These findings are consistent with our theoretical conclusion in Remark (3) in Line 406 on Page 8.
>
> Complementary to these findings, Tables 3 and 4 present the total communication cost needed to achieve a target accuracy of **87%** under the same parameter variations. The results show that:
> * Increasing $I$ and decreasing $K$ substantially reduce communication cost. For example, raising $I$ from 5 to 40 saves approximately **80%** of the communication overhead, while reducing $K$ from 15 to 1 saves approximately **94%**.
>
> Collectively, these results validate Swift-FedGNN’s ability to reduce communication without incurring major information loss and demonstrate that Swift-FedGNN provides a tunable balance between communication efficiency and accuracy, and the trade-off can be controlled via $I$ and $K$.
>
> In practice, we observe that a wide range of $I$ and $K$ settings yield strong accuracy while substantially reducing communication overhead. This flexibility of Swift-FedGNN having two "control knobs" in $I$ and $K$ makes it well-suited for practical deployment under varying resource constraints. Practitioners can choose $I$ and $K$ based on their specific system budgets and application requirements. Specifically, for resource-constrained clients, using a larger $I$-value (less frequent cross-client training) and a smaller $K$-value (fewer sampled clients) can significantly reduce communication without major accuracy loss. Conversely, for accuracy-critical applications with more communication budget, a smaller $I$-value and a larger $K$-value offer the best accuracy.
>
>
> Table 1: Final validation accuracy of Swift-FedGNN with different correction frequencies ($I$) on ogbn-products dataset, with the number of sampled clients fixed at $K = 10$.
> | Correction Frequencies ($I$) | 5| 10 | 20 | 40 |
> | -------- | -------- | -------- |-------- |-------- |
> | Validation Accuracy (%)     | 88.91 | 88.88     | 88.60     | 88.44 |
>
>
> Table 2: Final validation accuracy of Swift-FedGNN with different numbers of sampled clients ($K$) on ogbn-products dataset, with the correction frequency fixed at $I = 10$.
> | # of Sampled Clients ($K$) | 1 | 5 | 10| 15|
> | -------- | -------- | -------- |-------- |-------- |
> | Validation Accuracy (%)     | 88.47     | 88.72     | 88.88 |89.22|
>
>
> Table 3: Total communication cost of Swift-FedGNN with different correction frequencies ($I$) on ogbn-products dataset when achieving a target validation accuracy of 87%.
> | Correction Frequencies ($I$) | 5| 10 | 20 | 40 |
> | -------- | -------- | -------- |-------- |-------- |
> |Total Communication Cost (MB)|1344.0|675.5|324.5|275.0|
>
>
> Table 4: Total communication cost of Swift-FedGNN with different numbers of sampled clients ($K$) on ogbn-products dataset when achieving a target validation accuracy of 87%.
> | # of Sampled Clients ($K$) | 1 | 5 | 10|15|
> | -------- | -------- | -------- |-------- |-------- |
> |Total Communication Cost (MB)|57.8|342.2|675.5|1027.4|

---

> ### Author Response · Authors · 2025-11-22
>
> > **Weakness 2:** Analysis of Performance-Efficiency Trade-off for $I$ and $K$: The paper discusses the relationship between the correction frequency $I$ and the number of sampled cross-client training clients $K$ with the computation-to-communication ratio (Figure 7a, 7b), but it does not show their impact on the final model validation accuracy. Please provide experimental results showing the sensitivity of the validation accuracy as a function of $I$ and $K$ (e.g., plots of accuracy versus communication cost or wall-clock time for varying $(I,K)$ pairs). This is crucial for fully assessing the method's core trade-off and for guiding optimal hyperparameter settings in practice.
>
> **Response:** Thanks for your comment. Since this comment is closely tied with the previous comment, to avoid repetition, please refer to our response to your *Weakness 1* for our detailed analysis of the performance-efficiency trade-off for $I$ and $K$.

---

> ### Comment · Reviewer_qPZs · 2025-11-25
>
> Thank you for your response. I have decided to maintain my original positive score.

---

### Official Review · Reviewer_uhnB · 2025-10-31

**Soundness:** 3
**Presentation:** 3
**Contribution:** 3
**Rating:** 6
**Confidence:** 3

**Summary:**

This paper proposes Swift-FedGNN, a new framework for federated graph learning that achieves efficient training on geo-distributed graphs while reducing both communication and sample complexities. The key idea is to perform frequent local updates on each client and periodic cross-client aggregation on a subset of sampled clients.
The paper provides a theoretical convergence guarantee under biased stochastic gradients (Theorem 5.6), and comprehensive experiments on large-scale benchmarks demonstrate faster convergence and lower communication cost compared with strong baselines such as FedSage, FedGNN-G, and LLCG. Overall, the manuscript is technically sound, well-structured, and addresses a practically relevant problem in federated graph learning.

**Strengths:**

1. The paper tackles a concrete and under-studied challenge: how to train GNNs efficiently when graph data are distributed across multiple institutions and cannot be shared due to privacy regulations.
2. Swift-FedGNN combines parallel local training with periodic cross-client aggregation. Randomly sampled clients share only aggregated neighbor embeddings rather than raw node features. Additionally, this work rigorously analyzes convergence under biased stochastic gradients.
3. Experiments on multiple datasets (Reddit, OGBN-Products, Cora, Citeseer) show that Swift-FedGNN achieves up to 4× speed-up and an order-of-magnitude reduction in communication with competitive accuracy.

**Weaknesses:**

1. Although the paper only shares aggregated embeddings, could the authors provide further analysis or experiments to verify that these embeddings cannot be reconstructed or used to infer sensitive information?
2. The communication and computation complexity is currently demonstrated only empirically. It is recommended that the authors provide asymptotic expressions as functions of the number of clients (M), sampled clients (K), communication interval (I), and fan-out (F) to enhance the theoretical completeness of the paper. (*I understand this may be challenging, but I hope the authors can improve this aspect as much as possible. This will not affect my decision, as empirical validation already provides partial support.*)
3. The offloading mechanism only supports element-wise aggregation (mean/sum/max), and attention-based or heterogeneous GNNs requiring non-element-wise operations (e.g., GAT, HAN) are not discussed.

**Questions:**

1. Could aggregated embeddings still leak information under small client sizes or sparse features? Any empirical test or mitigation (e.g., DP noise)?
2. What happens when M ≫ 20 clients? Does server-side aggregation become a bottleneck?
3. What are realistic magnitudes of constants such as $B_{\Delta G}$ in Theorem 5.6, and do empirical error curves align with theoretical rates?

**Details Of Ethics Concerns:**

No ethical concerns detected.

---

> ### Author Response · Authors · 2025-11-22
>
> > **Weakness 1:** Although the paper only shares aggregated embeddings, could the authors provide further analysis or experiments to verify that these embeddings cannot be reconstructed or used to infer sensitive information?
>
> **Response:** Thanks for your comment. While Swift-FedGNN avoids transmitting raw node features by only sharing aggregated neighbor embeddings, we acknowledge that, in theory, aggregated embeddings could still carry sensitive information that might be exploited for reconstruction or inference attacks, particularly in cases involving extremely small client sizes, sparse features, or low-degree nodes (e.g., when a node has only a single neighbor), making the aggregated embedding trivially informative.
>
> In this work, our primary privacy motivation is to avoid the direct transmission of raw node features, which are often privacy-sensitive in real-world graph applications (e.g., user attributes in social networks). Our “aggregate-then-transfer” design ensures that:
> * Only aggregated neighbor embeddings (not raw features) are shared across clients, and
> * No raw node information is directly exposed to other clients or the server.
>
> That said, we do *not* claim formal privacy guarantees (e.g., differential privacy bounds) in this work, since simply using aggregation without Gaussian/Laplacian-type noise injection is unlikely to offer $(\epsilon,\delta)$-type differential privacy (DP) guarantee. Instead, our focus is on reducing communication overhead in federated graph learning while improving practical privacy-preserving behavior through communication-efficient design.
>
> Importantly, the Swift-FedGNN framework is **compatible with standard DP techniques** and federated encryption protocols, which can be integrated Gaussian/Laplacian-type noise injection to provide formal privacy guarantees. Such extensions are orthogonal to our focus on communication efficiency for federated graph learning in this paper and are left for future work.

---

> ### Author Response · Authors · 2025-11-22
>
> > **Weakness 2:** The communication and computation complexity is currently demonstrated only empirically. It is recommended that the authors provide asymptotic expressions as functions of the number of clients (M), sampled clients (K), communication interval (I), and fan-out (F) to enhance the theoretical completeness of the paper.
>
> **Response:** Thanks for your comment. Following your suggestions, we provide asymptotic characterizations for the communication and computation complexities of our proposed Swift-FedGNN algorithm. Due to the complications in precisely analyzing the communication and computation costs, we provide a high-level worst-case asymptotic analysis based on the several key system parameters.
>
> Throughout the analysis, we assume an $L$-layer GNN and the following parameters:
> * $M$: The total number of clients;
> * $F$: The same number of neighbor sampling fan-out used at each layer;
> * $F^{l}$: The worst-case number of neighbors at each training node at each GNN layer $l\in[1,L]$ using $F$-fan-out;
> * $p_{(l)} \in (0, 1)$: The fraction of the neighbors that are located on other clients.
>
> **1) Communication Complexity:** Every $I$ iterations, each of the $K$ sampled clients performs cross-client training and exchanges **aggregated embeddings** for its cross-client neighbors. The total communication cost for exchanging these embeddings is on the order of:
> $$\mathcal{O}\left(\frac{T}{I}KB\sum_{l=1}^{L}p_{(l)}F^ld_{(l-1)}^{emb}\right),$$ where $T$ is the total number of training iterations, $B$ is the batch size per client, $d_{(l)}^{\text{emb}}$ is the embedding (hidden) dimension at layer $l$, and $F^l$ reflects the exponential expansion in sampled neighborhoods as the layer depth increases.
>
> Note that this estimate does **not** account for the two-stage aggregation in Swift-FedGNN, which will significantly reduce the size of transferred embeddings. Therefore, this expression only represents a **conservative (worst-case) upper bound**, and the actual communication overhead is likely to be much lower.
>
> If the $p_{(l)}F^l$ cross-client neighbors at layer $l$ are distributed across $C_{(l)} < M$ remote clients, then after aggregation, the communication cost becomes:
> $$\mathcal{O}\left(\frac{T}{I}KB\sum_{l=1}^{L}C_{(l)}d_{(l-1)}^{emb}\right),$$ where $C_{(l)}\ll p_{(l)}F^l$ due to the aggregation mechanism in Swift-FedGNN..
>
> In addition, gradients and global model parameters are transmitted in every iteration. Let the model parameters at layer $l$ be $W_{(l)} \in \mathbb{R}^{d_{(l-1)}^{\text{emb}} \times d_{(l)}^{\text{emb}}}$. Then the total communication cost for gradients and model updates is on the order of:
> $$\mathcal{O}\left(2TM\sum_{l=1}^{L}d_{(l-1)}^{emb} d_{(l)}^{emb}\right).$$
>
> Combining both, the total communication complexity is on the order of:
> $$\mathcal{O}\left(2TM\sum_{l=1}^{L}d_{(l-1)}^{emb} d_{(l)}^{emb}+\frac{T}{I}KB\sum_{l=1}^{L}C_{(l)}d_{(l-1)}^{emb}\right).$$
>
> **2) Computation Complexity:** At each GNN layer $l$, the per-node computational cost includes:
> * Neighbor aggregation (e.g., mean/sum/max): $\mathcal{O}(Fd_{(l-1)}^{emb})$.
> * Linear transformation: $\mathcal{O}(d_{(l-1)}^{emb}d_{(l)}^{emb})$.
>
> With a total of $F^{l-1}$ sampled nodes at layer $l$ (due to recursive fan-out), the total per-batch cost per client is on the order of:
> $$\mathcal{O}\left(BF^{l-1}\left(Fd_{(l-1)}^{emb}+d_{(l-1)}^{emb}d_{(l)}^{emb}\right)\right).$$
>
> Across $T$ training iterations and $M$ clients, the total computation complexity (including both forward and backward passes) can be expressed as:
> $$\mathcal{O}\left(2TM\sum_{l=1}^{L}BF^{l-1}\left(Fd_{(l-1)}^{emb}+d_{(l-1)}^{emb}d_{(l)}^{emb}\right)\right).$$

---

> ### Author Response · Authors · 2025-11-22
>
> > **Weakness 3:** The offloading mechanism only supports element-wise aggregation (mean/sum/max), and attention-based or heterogeneous GNNs requiring non-element-wise operations (e.g., GAT, HAN) are not discussed.
>
> **Response:** Thanks for your comment. We agree that our current offloading mechanism supports only element-wise aggregation operations (e.g., mean, sum, and max). This design choice is *intentional* and aligns with the goal of **reducing communication overhead and avoiding raw feature transmission**, which remains a central challenge in federated graph learning (FGL).
>
> We would like to clarify that we do not consider attention-based or heterogeneous GNNs that require non-element-wise operations in the design and experiments of our federated GNN algorithm, since the main focus of this work is on **communication-efficient FGL**. We are fully aware that attention-based or heterogeneous GNNs are highly popular GNN models, both in the literature and in practice. However, they are **fundamentally not a good fit** in any communication-efficient federated GNN algorithm design.
>
> Taking GAT as an example, GAT requires direct access to **raw neighbor features/embeddings** to compute attention weights based on **nonlinear** pairwise interactions (see Eq. (1) in [1]). This requirement necessitates transmitting raw neighbor features/embeddings across clients, which leads to significantly high communication overhead. In other words, it is **impossible** for GAT to leverage the same communication-efficient aggregated transmissions as in those GNN models based on linear weighted aggregations (e.g., GCN-type models). As a result, GAT is not an ideal GNN model choice in those federated GNN algorithm design settings, where communication efficiency is of utmost importance.
>
> To further quantitatively understand GAT's communication efficiency limitation in federated GNN algorithm design, we analyze the communication cost of incorporating GAT into Swift-FedGNN (denoted as GAT-Swift-FedGNN) in here and compare it with our original Swift-FedGNN design. Throughout the analysis, we assume an $L$-layer GNN and the following parameters:
> * $M$: The total number of clients;
> * $F$: The same number of neighbor sampling fan-out used at each layer;
> * $F^{l}$: The worst-case number of neighbors at each training node at each GNN layer $l\in[1,L]$ using $F$-fan-out;
> * $p_{(l)} \in (0, 1)$: The fraction of the neighbors that are located on other clients.
>
> **1. GAT-Swift-FedGNN:** Every $I$ iterations, each of the $K$ sampled clients performs cross-client training and exchanges **raw features/embeddings** for its cross-client neighbors. The total communication cost per cross-client training round for exchanging these embeddings is on the order of:
> $$\mathcal{O}\left(KB\sum_{l=1}^{L}p_{(l)}F^ld_{(l-1)}^{emb}\right),$$ where $B$ is the batch size per client, $d_{(l)}^{\text{emb}}$ is the embedding (hidden) dimension at layer $l$, and $F^l$ reflects the exponential expansion in sampled neighborhoods as the layer depth increases.
>
> **2. Swift-FedGNN:** In contrast, Swift-FedGNN avoids transferring raw features/embeddings by sharing **aggregated neighbor features/embeddings**. If the $p_{(l)}F^l$ cross-client neighbors at layer $l$ are distributed across $C_{(l)} < M$ remote clients, then after aggregation, the communication cost is on the order of:
> $$\mathcal{O}\left(KB\sum_{l=1}^{L}C_{(l)}d_{(l-1)}^{emb}\right).$$
>
> Since $C_{(l)} \ll p_{(l)}F^l$, Swift-FedGNN achieves significantly lower communication overhead than the GAT variant.
>
>
> From the above analysis, we can see that GAT-Swift-FedGNN incurs a communication cost that is not only $F^l$ times higher than Swift-FedGNN, but the gap between them also grows **exponentially** as the number of layers increases.
>
> Again, we acknowledge that attention-based or heterogeneous GNNs requiring non-element-wise operations are important GNN models. However, how to reduce the communication cost and avoid raw neighbor feature transmission for them in the federated setting is a fundamentally hard problem, which would require major architectural design changes in these attention-based or heterogeneous model rather than a straightforward adaptation. Clearly, this is far beyond the scope of this work, and even the feasibility of modifying attention-based or heterogeneous GNNs requiring non-element-wise operations for the federated setting itself is substantial enough to merit a separate study.
>
>
> [1] Veličković, P., Cucurull, G., Casanova, A., Romero, A., Liò, P., & Bengio, Y. (2018, February). Graph Attention Networks. In International Conference on Learning Representations.

---

> ### Author Response · Authors · 2025-11-22
>
> > **Question 1:** Could aggregated embeddings still leak information under small client sizes or sparse features? Any empirical test or mitigation (e.g., DP noise)?
>
> **Response:** Thanks for your question. We acknowledge that aggregated embeddings could potentially leak information, particularly in scenarios involving small client sizes or sparse features. To avoid repetition, for a more detailed discussion, please refer to our response to your *Weakness 1*, where we outline the privacy limitations of aggregated embedding sharing and possible mitigations (e.g., injecting Gaussian/ noise into the aggregated embeddings before transmission to further reduce the risk of privacy leakage and provide formal privacy guarantees).
>
>
> > **Question 2:** What happens when M ≫ 20 clients? Does server-side aggregation become a bottleneck?
>
> **Response:** Thanks for your questions. In this rebuttal period, we have extended our experiments to two large-scale settings with **80 clients** and **100 clients** to more thoroughly evaluate scalability. Tables 1 and 3 below report the total communication cost on ogbn-products dataset when achieving a target validation accuracy, and Tables 2 and 4 present the final validation accuracy. These results show that:
> * With $K=10$, Swift-FedGNN reduces total communication cost by **at least 82%** in the 80-client setting and **at least 83%** in the 100-client setting compared to all baselines.
> * With $50\\%$ client sampling, Swift-FedGNN still achieves **at least 38%** communication savings in the 80-client setting and **at least 23%** communication savings in the 100-client setting over all baselines.
> * In both settings, Swift-FedGNN maintains comparable or better validation accuracy, with larger $K$ yielding slightly improved performance.
>
> The above findings highlight Swift-FedGNN’s effectiveness and scalability, validating its communication efficiency even in large-scale federated graph learning settings. We also emphasize that Swift-FedGNN is designed to reduce communication overhead while maintaining competitive accuracy, not to surpass baselines in absolute accuracy.
>
> Regarding the question of whether server-side aggregation becomes a bottleneck when $M \gg 20$ clients, our response is that it depends on the number of sampled clients $K$ used for periodic cross-client training. Specifically:
> * When $K \ll M$, the server will not be an aggregation bottleneck: the server only needs to aggregate cross-client neighbor information from a small subset of clients in each round for cross-client training. As a result, server-side latency increases slowly with $M$, thus preserving the overall scalability of the system.
> * When $K$ is comparable or equal to $M$, the server-side aggregation would indeed become a bottleneck, as it must handle cross-client neighbor information from all or most clients. However, this can be mitigated by reducing the $K$-value, which is fully supported by our framework.
>
> In summary, the server-side aggregation in Swift-FedGNN will not be a bottleneck in the large-client settings as long as $K$ is adjusted appropriately.
>
>
> Table 1: Total communication cost on ogbn-products dataset with **80 clients** when achieving a target validation accuracy of 83%.
>
> ||Swift-FedGNN (K/M=10/80)|Swift-FedGNN (K/M=40/80)|FedGNN-G|LLCG|FedGNN-PNS|
> |-|-|-|-|-|-|
> |Total Communication Cost (MB)|711.5|2476.8|38154.2|3988.8|4601.8|
>
>
> Table 2: Final validation accuracy (%) on ogbn-products dataset with **80 clients**.
>
> ||Swift-FedGNN (K/M=10/80)|Swift-FedGNN (K/M=40/80)|FedGNN-G|LLCG|FedGNN-PNS|
> |-|-|-|-|-|-|
> |Final Validation Accuracy (%)|85.74|86.12|86.52|83.54|85.67|
>
>
> Table 3: Total communication cost on ogbn-products dataset with **100 clients** when achieving a target validation accuracy of 84.3%.
>
> ||Swift-FedGNN (K/M=10/100)|Swift-FedGNN (K/M=50/100)|FedGNN-G|LLCG|FedGNN-PNS|
> |-|-|-|-|-|-|
> |Total Communication Cost (MB)|1196.0|5367.7|60419.5|6987.4|8880.5|
>
>
> Table 4: Final validation accuracy (%) on ogbn-products dataset with **100 clients**.
>
> ||Swift-FedGNN (K/M=10/100)|Swift-FedGNN (K/M=50/100)|FedGNN-G|LLCG|FedGNN-PNS|
> |-|-|-|-|-|-|
> |Final Validation Accuracy (%)|85.07|85.53|85.63|84.35|85.15|

---

> ### Author Response · Authors · 2025-11-22
>
> > **Question 3:** What are realistic magnitudes of constants such as $B_{\Delta G}$ in Theorem 5.6, and do empirical error curves align with theoretical rates?
>
>
> **Response:** Thanks for your questions. To illustrate the realistic magnitudes of constants such as $B_{\Delta G}$ in Theorem 5.6, we use the gradient bias between the full gradient and the stochastic gradient as an empirical proxy. This quantity has a theoretical upper bound of $LB_{\Delta G}^{l}$, making it a suitable example for analysis.
>
> Table 1 presents the measured gradient bias on the ogbn-arxiv dataset under varying GNN depths. These results clearly show that gradient bias increases with the GNN depth, consistent with our theoretical result: *"Deeper GNNs incur larger bias due to amplified sampling and cross-client neighbor errors."*
>
> Table 2 shows the validation accuracy on the ogbn-arxiv dataset under varying GNN depths. We observe that validation accuracy degrades as the GNN depth increases. This behavior is consistent with our theory: *"Deeper GNNs introduce larger gradient bias terms, which in turn lead to greater approximation error and reduced performance."*
>
> In summary, the empirical trends above corroborate the theoretical predictions in Theorem 5.6, confirming both the validity and practical relevance of the error bounds in Theorem 5.6.
>
> Lastly, it is worth emphasizing that the performance degradation observed with increasing GNN depth is a **fundamental limitation of GNN architectures themselves**, rather than a limitation our Swift-FedGNN algorithm design. This phenomenon is well-known to occur across GNNs regardless of the specific graph learning algorithm in use. Addressing it typically requires architecture-level enhancements, and many existing solutions (e.g., [1]) are fully compatible with our Swift-FedGNN design and can be integrated to mitigate depth-related degradation in practice.
>
>
> Table 1: Gradient bias under varying GNN depths on the ogbn-arxiv dataset.
> | GNN Depth | 2 | 14 | 16 |
> | -------- | -------- | -------- | -------- |
> | Gradient Bias     | 0.46     | 17.56     | 30.11     |
>
>
> Table 2: Validation accuracy (%) under varying GNN depths on the ogbn-arxiv dataset.
> | GNN Depth | 2 | 14 | 16 |
> | -------- | -------- | -------- | -------- |
> | Validation accuracy (%)     | 57.17  | 54.60    |48.46|
>
>
> [1] Chen, T., Zhou, K., Duan, K., Zheng, W., Wang, P., Hu, X., & Wang, Z. (2022). Bag of tricks for training deeper graph neural networks: A comprehensive benchmark study. IEEE Transactions on Pattern Analysis and Machine Intelligence, 45(3), 2769-2781.

---

> ### Comment · Reviewer_uhnB · 2025-11-24
>
> Thanks to the authors for the very detailed and clear rebuttal. I’ve gone through the responses, and they address my earlier points well.
>
> * **W1:** The explanation of the privacy assumptions and the limits of aggregation-based protection is clear, and noting the compatibility with DP extensions helps set the right scope.
>
> * **W2:** The added asymptotic complexity analysis (in terms of M, K, I, F, and depth) gives the theoretical detail I felt was missing and fits well with the empirical results.
>
> * **W3:** The clarification on why attention-based or heterogeneous GNNs are difficult to support in a communication-efficient federated setup is reasonable, and the comparison with a hypothetical GAT variant makes the point clear.
>
> * **Q1:** The privacy clarification is consistent with what I expected.
>
> * **Q2:** The new experiments with 80 and 100 clients make the scalability argument more convincing, and the discussion of server-side bottlenecks is helpful.
>
> * **Q3:** The empirical check on the constants in Theorem 5.6 aligns with the theory and answers my question.
>
> Overall, the rebuttal resolves my concerns. I hope the authors can incorporate these clarifications into the final camera-ready version. I am raising my score from 6 to 8. My confidence remains the same.

---

> > ### Author Response · Authors · 2025-11-24
> >
> > Dear Reviewer uhnB,
> >
> > We sincerely thank you for the thoughtful feedback and generous score increase. We’re very glad to hear that our clarifications addressed your concerns.  We will incorporate all relevant clarifications into the revised version of our paper. Thank you again for your constructive comments throughout the review process.
> >
> > Sincerely,\
> > The Authors

---

### Official Review · Reviewer_uxRE · 2025-10-31

**Soundness:** 2
**Presentation:** 1
**Contribution:** 2
**Rating:** 2
**Confidence:** 3

**Summary:**

This paper proposes Swift-FedGNN, a communication- and sample-efficient framework for federated graph neural network (GNN) training on geo-distributed graphs where data privacy prevents direct feature sharing among clients. The method primarily performs efficient local training on each client and periodically selects a subset of clients to conduct cross-client training that accounts for dependencies between nodes across clients. By offloading partial aggregation operations to remote clients and the central server, Swift-FedGNN mitigates the information loss of cross-client neighbors while substantially reducing communication and memory overhead. The authors provide a rigorous theoretical convergence analysis, proving that Swift-FedGNN's convergence rate. Extensive experiments on large-scale real-world datasets demonstrate that Swift-FedGNN achieves comparable accuracy to full cross-client training baselines while significantly improving efficiency in terms of training time and communication cost.

**Strengths:**

The paper addresses an important and timely problem in federated graph learning by aiming to reduce communication and sampling overhead while preserving accuracy under privacy constraints. It provides a clear problem formulation and a well-structured methodological design combining local training and periodic cross-client aggregation. The theoretical analysis is good and extends convergence result to a biased-gradient federated GNN setting.

**Weaknesses:**

The paper’s main weakness lies in its limited novelty—while the proposed framework is well-motivated, it largely combines known techniques from federated learning and distributed GNN training, such as periodic communication and hierarchical aggregation, without introducing a clearly new conceptual or algorithmic component. The idea of alternating between local and cross-client updates has been explored in several prior works on communication-efficient federated optimization and graph learning, and the paper does not convincingly demonstrate how Swift-FedGNN differs fundamentally or advances beyond these methods. Moreover, the experimental evaluation, though conducted on two standard benchmarks, is not sufficiently comprehensive to support the claimed generality or superiority. The set of baselines is limited, omitting comparisons with stronger or more recent federated GNN approaches that also target efficiency and scalability. The ablation and sensitivity analyses are relatively shallow, leaving unclear how different design choices impact performance in heterogeneous settings. Finally, the paper’s writing and presentation could be improved for clarity and precision, as some methodological descriptions are verbose yet lack formal rigor or intuitive explanations, which diminishes the overall readability and perceived contribution.

**Questions:**

1. The paper claims to achieve low communication and sampling complexity, but the analysis remains largely empirical. A clearer quantitative or theoretical comparison of communication cost with existing methods such as FedSage+ or FedGNN-PNS would strengthen the claim.

2. The periodic cross-client training strategy relies on fixed intervals and random client subsets, which appear heuristic. The sensitivity of performance to these design choices and whether adaptive or data-driven scheduling could offer improvements remain unclear.

3. The experimental evaluation is limited in scope. Additional benchmarks and comparisons with more recent or stronger federated GNN baselines would help demonstrate the robustness and generality of the proposed approach.

---

> ### Author Response · Authors · 2025-11-22
>
> > **Weakness 1:** The paper’s main weakness lies in its limited novelty—while the proposed framework is well-motivated, it largely combines known techniques from federated learning and distributed GNN training, such as periodic communication and hierarchical aggregation, without introducing a clearly new conceptual or algorithmic component. The idea of alternating between local and cross-client updates has been explored in several prior works on communication-efficient federated optimization and graph learning, and the paper does not convincingly demonstrate how Swift-FedGNN differs fundamentally or advances beyond these methods.
>
> **Response:** Thanks for your comment. We would like to further clarify the novelty of this work (also acknowledged by several other reviewers). While Swift-FedGNN incorporates techniques concepts found in the broader literature on federated learning and distributed GNN training (e.g., periodic communication and local/global updates), its key novelty lies in the algorithmic design tailored specifically to the challenges of *federated graph learning* (FGL), along with theoretical and system-level innovations that go beyond prior work. We summarize the novelty of our work as follows:
>
> **1. New FGL Framework with Two-Stage Aggregation:** Unlike prior works that perform either direct cross-client neighbor exchange (e.g., LLCG [1]) or require full-neighbor aggregation and training at a central server (e.g., FedGNN-PNS [2]), Swift-FedGNN introduces a **two-stage aggregation** scheme:
> * First, remote clients aggregate their neighbor information locally,
> * Then, the server performs a second aggregation step before sending aggregated embeddings to the training client.
>
> This aggregated-then-transferred strategy avoids direct exchange of raw features, ensuring better privacy preservation and reducing both communication and memory costs.
>
>
> **2. New Theoretical Analysis of Biased Stochastic Gradients in GNNs:** To our knowledge, this is the first work to rigorously analyze the convergence of FGL with **biased stochastic gradients** arising from both neighbor sampling and missing cross-client neighbors (i.e., without relying on strong assumptions like unbiased or consistent gradients as in prior works). We explicitly bound the gradient approximation errors (Lemmas 5.4–5.5) and show their positive correlation with GNN depth, a structural property unique to graph models **not found** in the literature. This leads to a **new** convergence guarantee (Theorem 5.6) that matches the state-of-the-art convergence rate for sampling-based centralized GNN training, despite operating under the much more restrictive and challenging federated setting.
>
> **3. System-Level Novelty and Empirical Effectiveness:** Prior works either incur large memory/communication overhead (e.g., LLCG or FedGNN-PNS), rely on training nodes and neighbors reuse (FedGNN-PNS), or require full-neighbor GNN training at the server (LLCG). In contrast, Swift-FedGNN offers the following salient features:
> * Restricting cross-client training to a randomly sampled subset of clients,
> * Avoiding server-side GNN training,
> * Offloading aggregation to the server and remote clients,
> * No need for transferring raw node features,
> * Achieving superior communication efficiency, such as reducing total communication cost by at least **41%** on the large ogbn-products dataset and at least **32%** on the smaller and sparser ogbn-arxiv dataset, all while enjoying comparable accuracy.
>
> In summary, Swift-FedGNN combines algorithmic, theoretical, and system-level innovations tailored to the FGL setting. While some components share similarities with known ideas, its judicious algorithmic design, rigorous theoretical analysis, and empirical results represent a *novel* and *impactful* contribution to the field.
>
>
> [1] Ramezani, M., Cong, W., Mahdavi, M., Kandemir, M. T., & Sivasubramaniam, A. (2022). Learn locally, correct globally: A distributed algorithm for training graph neural networks. In International Conference on Learning Representations.
>
> [2] Du, B., & Wu, C. (2022, June). Federated graph learning with periodic neighbour sampling. In 2022 IEEE/ACM 30th International Symposium on Quality of Service (IWQoS) (pp. 1-10). IEEE.

---

> ### Author Response · Authors · 2025-11-22
>
> > **Weakness 2:** Moreover, the experimental evaluation, though conducted on two standard benchmarks, is not sufficiently comprehensive to support the claimed generality or superiority.
>
> **Response:** Thanks for your comment. We agree that broad experimental coverage is important for assessing the generality and superiority of our proposed method.
>
> In our original submission, we evaluated Swift-FedGNN across **five real-world graph datasets** (ogbn-products, Reddit, ogbn-arxiv, flickr, and Citeseer), which span diverse scales and densities. Notably, ogbn-products is among the largest datasets used in the federated graph learning literature, and Reddit is known for its high density, both posing significant challenges for scalable federated graph learning.
>
> To assess robustness under different graph heterogeneity scenarios, in our original submission, we additionally experiment with **two partition strategies**: i) METIS-based partitioning, which generates connectivity-aware and balanced subgraphs; and ii) Random partitioning, which introduces heterogeneity by randomly assigning nodes to different subgraphs, thereby implicitly inducing non-identical and structurally unbalanced local subgraphs. We also test **two GNN models**, GraphSAGE and GIN, to validate model-agnostic effectiveness.
>
> We compare Swift-FedGNN against **three baselines** (FedGNN-G, FedGNN-PNS, and LLCG) that aim to mitigate information loss from cross-client neighbors, as well as **an additional baseline**, FedSage, included in the appendix. FedSage entirely ignores cross-client neighbor information and, as expected, results in degraded performance. We would like to clarify that the selected baselines were chosen because they are the **most closely related** to our proposed algorithm. These baselines specifically address the core challenges of reducing communication costs in federated graph learning and mitigating information loss from cross-client neighbors through periodic (sampling-based) full-neighbor training.
>
> Our experimental results consistently demonstrate that Swift-FedGNN achieves comparable or even better validation accuracy while significantly reducing communication cost **across all datasets, GNN models, and partitioning strategies**. For example, under METIS-based partitioning with the GraphSAGE model, Swift-FedGNN reduces the total communication cost by at least **41%** on the large ogbn-products dataset and by at least **32%** on the smaller and sparser ogbn-arxiv dataset when achieving the same validation accuracy as the baselines. These results highlight the effectiveness of Swift-FedGNN in achieving communication efficiency without sacrificing model performance, thereby demonstrating its robustness, generality, superiority, and scalability in diverse federated graph learning scenarios.
>
> To conclude, we summarize all the above discussions on our comprehensive experiment coverage in the following table.
>
> |  | Experiment Summary |
> | -------- | -------- |
> | **Datasets**     | ogbn-products, Reddit, ogbn-arxiv, flickr,  Citeseer     |
> | **Partition Strategies**     | METIS-based partitioning, Random partitioning     |
> | **GNN Models**     | GraphSAGE, GIN     |
> | **Baselines**     | FedGNN-G, FedGNN-PNS, LLCG, FedSage     |

---

> ### Author Response · Authors · 2025-11-22
>
> > **Weakness 3:** The set of baselines is limited, omitting comparisons with stronger or more recent federated GNN approaches that also target efficiency and scalability.
>
>
> **Response:** Thanks for your comment. We would like to clarify that the baselines included in our paper are the most closely related to our proposed method, as discussed in our response to your *Weakness 2*. Regarding stronger or more recent federated GNN approaches targeting communication efficiency and scalability, we have carefully surveyed the literature in this rebuttal period and found very few methods that are directly comparable to Swift-FedGNN, specifically those that (i) support mini-batch training on large-scale graphs, and (ii) explicitly target communication efficiency.
>
> To our knowledge, GraphProxy [1] (2024) is the only related approach, which does aim to reduce communication in federated graph learning. However, Swift-FedGNN and GraphProxy are designed to tackle fundamentally different challenges:
> * GraphProxy focuses on reducing the cost of full-neighbor GNN training by replacing remote neighbors with locally constructed proxy nodes based on embedding similarity. This approach works well on small or moderately sized graphs and assumes a full-neighbor message passing model. However, it is not suitable for large-scale graphs due to the prohibitive size of full neighbor sets.
> * In contrast, Swift-FedGNN targets the high communication cost of neighbor sampling and data exchange in mini-batch training on large-scale graphs. Our method reduces communication overhead by controlling when and which clients engage in cross-client training, using a dual aggregation scheme that enables both scalability and accuracy.
>
> Moreover, Swift-FedGNN offers theoretical convergence guarantees under partial client participation and stochastic updates, while GraphProxy does **not** consider such regimes. As a result, we believe that a direct empirical comparison would not yield meaningful insights due to these fundamental differences in scope and assumptions.
>
> That said, we remain open to including newer baselines as they emerge and appreciate your suggestion as a valuable direction for future benchmarking.
>
>
> [1] Wang, J., Zhang, L., Wang, J., Yuan, M., Cheng, Y., Xu, Q., & Yu, B. (2024, May). GraphProxy: Communication-efficient federated graph learning with adaptive proxy. In IEEE INFOCOM 2024-IEEE Conference on Computer Communications (pp. 2179-2188). IEEE.

---

> ### Author Response · Authors · 2025-11-22
>
> > **Weakness 4:** The ablation and sensitivity analyses are relatively shallow, leaving unclear how different design choices impact performance in heterogeneous settings.
>
> **Response:** Thanks for your comment. To illustrate the impact of different choices for key parameters in Swift-FedGNN (i.e., the correction frequency $I$ and the client sampling size $K$), in this rebuttal period, we have conducted additional experiments to report the final validation accuracy of Swift-FedGNN on the ogbn-products dataset under different settings of $I$ and $K$ in Tables 1 and 2, respectively. The results demonstrate that:
> * Increasing the correction interval ($I$) from 5 to 40 (Table 1) leads to only a minor accuracy degradation (from 88.91% to 88.44%), demonstrating that **less frequent cross-client training still preserves model quality**.
> * Decreasing the number of sampled clients ($K$) from 15 to 1 (Table 2) also results in a minor accuracy drop (from 89.22% to 88.47%), indicating that **a small number of sampled clients is sufficient to maintain strong performance**.
>
> These findings are consistent with our theoretical conclusion in Remark (3) in Line 406 on Page 8.
>
> Complementary to the above findings, Tables 3 and 4 present the total communication cost needed to achieve a target accuracy of **87%** under the same parameter variations. These results show that:
> * Increasing $I$ and decreasing $K$ substantially reduce communication cost. For example, increasing $I$ from 5 to 40 saves approximately **80%** of the communication overhead, while reducing $K$ from 10 to 1 saves approximately **94%**.
>
> Collectively, these results validate Swift-FedGNN’s ability to reduce communication without incurring major information loss and demonstrate that Swift-FedGNN provides a tunable balance between communication efficiency and accuracy, and the trade-off can be controlled via $I$ and $K$.
>
> In practice, we observe that a wide range of $I$ and $K$ settings yield strong accuracy while substantially reducing communication overhead. This flexibility makes Swift-FedGNN well-suited for practical deployment under varying resource constraints. Practitioners can choose $I$ and $K$ based on their specific system budgets and application requirements. Specifically, for resource-constrained clients, using larger $I$ (less frequent cross-client training) and smaller $K$ (fewer sampled clients) can significantly reduce communication without major accuracy loss. Conversely, for accuracy-critical applications with more communication budget, smaller $I$ and larger $K$ offer the best accuracy.
>
>
> Table 1: Final validation accuracy of Swift-FedGNN with different correction frequencies ($I$) on ogbn-products dataset, with the number of sampled clients fixed at $K = 10$.
> | Correction Frequencies ($I$) | 5| 10 | 20 | 40 |
> | -------- | -------- | -------- |-------- |-------- |
> | Validation Accuracy (%)     | 88.91 | 88.88     | 88.60     | 88.44 |
>
>
> Table 2: Final validation accuracy of Swift-FedGNN with different numbers of sampled clients ($K$) on ogbn-products dataset, with the correction frequency fixed at $I = 10$.
> | # of Sampled Clients ($K$) | 1 | 5 | 10| 15|
> | -------- | -------- | -------- |-------- |-------- |
> | Validation Accuracy (%)     | 88.47     | 88.72     | 88.88 |89.22|
>
>
> Table 3: Total communication cost of Swift-FedGNN with different correction frequencies ($I$) on ogbn-products dataset when achieving a target validation accuracy of 87%.
> | Correction Frequencies ($I$) | 5| 10 | 20 | 40 |
> | -------- | -------- | -------- |-------- |-------- |
> |Total Communication Cost (MB)|1344.0|675.5|324.5|275.0|
>
>
> Table 4: Total communication cost of Swift-FedGNN with different numbers of sampled clients ($K$) on ogbn-products dataset when achieving a target validation accuracy of 87%.
> | # of Sampled Clients ($K$) | 1 | 5 | 10|15|
> | -------- | -------- | -------- |-------- |-------- |
> |Total Communication Cost (MB)|57.8|342.2|675.5|1027.4|
>
>
>
>
> > **Weakness 5:** Finally, the paper’s writing and presentation could be improved for clarity and precision, as some methodological descriptions are verbose yet lack formal rigor or intuitive explanations, which diminishes the overall readability and perceived contribution.
>
> **Response:** Thanks for your suggestion. We will refine and streamline the methodological descriptions and enhance both the formal rigor and the intuitive explanations in the revised version of our paper.

---

> ### Author Response · Authors · 2025-11-22
>
> > **Question 1:** The paper claims to achieve low communication and sampling complexity, but the analysis remains largely empirical. A clearer quantitative or theoretical comparison of communication cost with existing methods such as FedSage+ or FedGNN-PNS would strengthen the claim.
>
> **Response:** Thanks for your comment. We agree that presenting a more explicit quantitative or theoretical comparison of communication cost with existing methods would strengthen our communication-efficiency claims.
>
> Due to the complexity of precisely analyzing communication cost across different methods in different graphs, we provide a high-level asymptotic analysis based on key system parameters. Below, we compare the communication overhead incurred during rounds that involve the transmission of cross-client neighbor features or embeddings.
>
> Throughout the analysis, we assume an $L$-layer GNN and the following parameters:
> * $M$: The total number of clients;
> * $F$: The same number of neighbor sampling fan-out used at each layer;
> * $F^{l}$: The worst-case number of neighbors each training node at each GNN layer $l\in[1,L]$ using $F$-fan-out;
> * $p_{(l)} \in (0, 1)$: The fraction of the neighbors that are located on other clients.
>
> **1. Swift-FedGNN:** Every $I$ iterations, each of the $K$ sampled clients performs cross-client training and exchanges **aggregated embeddings** for its cross-client neighbors. The total communication cost per cross-client training round for exchanging these embeddings is on the order of:
> $$\mathcal{O}\left(KB\sum_{l=1}^{L}p_{(l)}F^ld_{(l-1)}^{emb}\right),$$ where $B$ is the batch size per client, $d_{(l)}^{\text{emb}}$ is the embedding (hidden) dimension at layer $l$, and $F^l$ reflects the exponential expansion in sampled neighborhoods as the layer depth increases.
>
> Note that this estimate does **not** account for the two-stage aggregation in Swift-FedGNN, which will significantly reduce the size of transferred embeddings. Therefore, this expression only represents a **conservative (worst-case) upper bound**, and the actual communication overhead is likely to be much lower.
>
> If the $p_{(l)}F^l$ cross-client neighbors at layer $l$ are distributed across $C_{(l)} < M$ remote clients, then after aggregation, the communication cost becomes:
> $$\mathcal{O}\left(KB\sum_{l=1}^{L}C_{(l)}d_{(l-1)}^{emb}\right),$$ where $C_{(l)}\ll p_{(l)}F^l$ due to the aggregation mechanism in Swift-FedGNN.
>
> **2. FedGNN-PNS:** FedGNN-PNS reduces communication by reusing the same sampled training nodes and their sampled neighbors across multiple training rounds, but it directly transmits **raw input features** for those cross-client neighbors. The total communication cost per cross-client neighbor update is approximately:
> $$\mathcal{O}\left(MB\sum_{l=1}^{L}p_{(l)}F^ld_{(0)}^{emb}\right),$$ where $d_{(0)}^{\text{emb}}$ is the input feature dimension, typically larger than hidden dimensions in deeper layers.
>
> **3. Comparison:** When communication of cross-client neighbors occurs, Swift-FedGNN is more efficient than FedGNN-PNS due to three key reasons:
> * It involves only $K < M$ clients per round;
> * It transmits lower-dimensional hidden embeddings (i.e., $d_{(l-1)}^{\text{emb}} < d_{(0)}^{\text{emb}}$ for $l \geq 2$);
> * It leverages two-stage aggregation to compress information prior to transmission (i.e., $C_{(l)}\ll p_{(l)}F^l$).
>
> Moreover, as training progresses, FedGNN-PNS increases the frequency of graph data communication, which can lead to significant cumulative overhead. In contrast, Swift-FedGNN maintains a **fixed periodic communication schedule** and reduces transferred data per round, resulting in substantially lower overall communication cost.
>
> We omit the quantitative communication-cost comparison between FedSage+ and Swift-FedGNN because FedSage+ is **not** designed to reduce communication overhead in federated graph learning. Instead, its primary goal is to repair incomplete local subgraphs through a missing-neighbor generator, which is trained separately and does **not** aim to minimize cross-client communication. As a result, FedSage+ operates under a fundamentally different design objective and is not directly comparable to communication-efficient FGL approaches such as FedGNN-PNS and our Swift-FedGNN method.

---

> ### Author Response · Authors · 2025-11-22
>
> > **Question 2:** The periodic cross-client training strategy relies on fixed intervals and random client subsets, which appear heuristic. The sensitivity of performance to these design choices and whether adaptive or data-driven scheduling could offer improvements remain unclear.
>
> **Response:** Thanks for your suggestion. We acknowledge that the choice of intervals ($I$) and random client subsets ($K$) plays a critical role in the performance of Swift-FedGNN. To address this, we have conducted a detailed analysis of the sensitivity of model performance to these hyperparameters. Please refer to our response to your *Weakness 4* for additional experimental results and discussions on how varying $I$ and $K$ affects both accuracy and communication cost.
>
> We would like to clarify that although the use of fixed intervals and random client subsets may appear heuristic, it already suffices to offer strong **theoretical convergence guarantees** and **significant communication cost reductions**, which constitute the primary goals of this work. Moreover, our paper introduces several key novelties, as outlined in our response to your *Weakness 1*.
>
> We agree with the reviewer that incorporating adaptive or data-driven scheduling mechanisms (e.g., based on gradient variance, local loss, or communication budget) could offer further improvements by dynamically selecting participating clients or adjusting the timing of cross-client training. However, designing such mechanisms involves substantial additional algorithmic complexity, which is also somewhat orthogonal to our main focus on communication complexity in this paper. We believe that adaptive and/or data-driven scheduling design will be better suited as an independent study in our future work. Last but not least, it is worth pointing out that the **modular nature** of Swift-FedGNN naturally supports such adaptive/data-driven extensions, and the contributions of our work remain significant even under the current design.
>
>
>
> > **Question 3:** The experimental evaluation is limited in scope. Additional benchmarks and comparisons with more recent or stronger federated GNN baselines would help demonstrate the robustness and generality of the proposed approach.
>
> **Response:** Thanks for your suggestion. To avoid repetition, please refer to our responses to your *Weaknesses 2* and *3*.

---

> > ### Comment · Reviewer_uxRE · 2025-11-23
> >
> > Thanks for the response, my concerns have been addressed, I will increase the score.

---

> ### Author Response · Authors · 2025-11-24
>
> Dear Reviewer uxRE,
>
> We truly appreciate your thoughtful follow-up and are grateful for the increased score. We’re very glad to hear that our responses addressed your concerns. Thank you again for your time and constructive feedback, which helped us improve the quality and clarity of our work.
>
> Sincerely,\
> The Authors

---

### Official Review · Reviewer_MwMc · 2025-11-01

**Soundness:** 3
**Presentation:** 3
**Contribution:** 3
**Rating:** 6
**Confidence:** 4

**Summary:**

The paper proposes Swift-FedGNN, a federated learning framework for graph neural networks that minimizes sampling and communication overhead in geo-distributed data. It performs local parallel training on each client and periodically conducts cross-client aggregation among a random subset of clients. Remote clients and the server collaboratively pre-aggregate neighbor embeddings so that only aggregated features, not raw data, are exchanged—reducing bandwidth and preserving privacy. The authors prove an $O(T^{-1/2})$ convergence rate matching state-of-the-art GNN methods.

**Strengths:**

S1. This paper targets a relevant and challenging setting where graph data are geo‑distributed with cross‑client dependencies.

S2. The proposed Swift-FedGNN has shown good convergence rates comparable to state-of-the-art methods theoretically and empirically.

S3. Experiments on five benchmarks show up to 4× faster convergence and 21× lower communication cost than existing FGL baselines while maintaining comparable accuracy.

**Weaknesses:**

W1. This paper’s privacy story is incomplete. The approach assumes a trusted server and, for non-element-wise operators (e.g., GAT), raw features or intermediate activations may need to be sent to the server, which undermines the “aggregated-only” guarantee and weakens privacy claims.

W2. The convergence analysis is proved for GCN, whereas the experiments emphasize GraphSAGE/GIN. The paper does not establish conditions under which the analysis extends to these architectures. Moreover, the error neighborhood explicitly grows with depth $L$, but the paper offers no practical prescription for training deeper models without degrading guarantees.

W3. The empirical comparison is limited to three or four methods. Important FGL variants—such as SpreadGNN and FedGCN—are not included in the core efficiency comparisons, despite being cited.

W4. The method section largely describes a clean but straightforward system design. Without a stronger learning-algorithm contribution (e.g., new estimators or privacy mechanisms), the work can read as primarily an engineering improvement.



Minor typos: "Communicaiton" -> "Communication" in Figs 6 and 7

**Questions:**

Q1. The background section cites the Microsoft Academic Graph (MAG, 100 M nodes) as an example of large-scale, geo-distributed graphs, yet experiments stop at ogbn-products (2.4 M nodes). Could the authors clarify why MAG or an equivalently large dataset was not used? Was this due to memory limits, partitioning complexity, or cross-client sampling constraints?

Q2. The main theoretical analysis explicitly targets GCNs. Can the authors discuss whether and how this framework extends to GraphSAGE, GIN, or attention-based models (e.g., GAT)? In particular, how would the error neighborhood term scale with network depth $L$ and non-linear aggregation functions?

Q3. Since the theory demonstrates that gradient bias increases with depth $L$, have the authors observed this empirically? For example, does validation accuracy degrade more rapidly for deeper architectures under the same $I,K$ settings?

---

> ### Author Response · Authors · 2025-11-22
>
> > **Weakness 1:** This paper’s privacy story is incomplete. The approach assumes a trusted server and, for non-element-wise operators (e.g., GAT), raw features or intermediate activations may need to be sent to the server, which undermines the “aggregated-only” guarantee and weakens privacy claims.
>
> **Response:** Thanks for your comment. We would like to clarify that we do not consider non-element-wise operators (e.g., GAT) in the design of our communication-efficient federated GNN algorithm. We are fully aware that non-element-wise operators are highly popular GNN models, both in the literature and in practice. However, non-element-wise operators are **fundamentally not a good fit** in any communication-efficient federated GNN algorithm design.
>
> Taking GAT as an example, GAT requires direct access to **raw neighbor features/embeddings** to compute attention weights based on **nonlinear** pairwise interactions (see Eq. (1) in [1]). This requirement necessitates transmitting raw neighbor features/embeddings across clients, which leads to significantly high communication overhead. In other words, it is **impossible** for GAT to leverage the same communication-efficient aggregated transmissions as in those GNN models based on linear weighted aggregations (e.g., GCN-type models). As a result, GAT is not an ideal GNN model choice in those federated GNN algorithm design settings, where communication efficiency is of utmost importance.
>
> To further quantitatively understand GAT's communication efficiency limitation in federated GNN algorithm design, we analyze the communication cost of incorporating GAT into Swift-FedGNN (denoted as GAT-Swift-FedGNN) in here and compare it with our original Swift-FedGNN design. Throughout the analysis, we assume an $L$-layer GNN and the following parameters:
> * $M$: The total number of clients;
> * $F$: The same number of neighbor sampling fan-out used at each layer;
> * $F^{l}$: The worst-case number of neighbors at each training node at each GNN layer $l\in[1,L]$ using $F$-fan-out;
> * $p_{(l)} \in (0, 1)$: The fraction of the neighbors that are located on other clients.
>
> **1. GAT-Swift-FedGNN:** Every $I$ iterations, each of the $K$ sampled clients performs cross-client training and exchanges **raw features/embeddings** for its cross-client neighbors. The total communication cost per cross-client training round for exchanging these embeddings is on the order of:
> $$\mathcal{O}\left(KB\sum_{l=1}^{L}p_{(l)}F^ld_{(l-1)}^{emb}\right),$$ where $B$ is the batch size per client, $d_{(l)}^{\text{emb}}$ is the embedding (hidden) dimension at layer $l$, and $F^l$ reflects the exponential expansion in sampled neighborhoods as the layer depth increases.
>
> **2. Swift-FedGNN:** In contrast, Swift-FedGNN avoids transferring raw features/embeddings by sharing **aggregated neighbor features/embeddings**. If the $p_{(l)}F^l$ cross-client neighbors at layer $l$ are distributed across $C_{(l)} < M$ remote clients, then after aggregation, the communication cost is on the order of:
> $$\mathcal{O}\left(KB\sum_{l=1}^{L}C_{(l)}d_{(l-1)}^{emb}\right).$$
>
> Since $C_{(l)} \ll p_{(l)}F^l$, Swift-FedGNN achieves significantly lower communication overhead than the GAT variant.
>
>
> From the above analysis, we can see that GAT-Swift-FedGNN incurs a communication cost that is not only $F^l$ times higher than Swift-FedGNN, but the gap between them also grows **exponentially** as the number of layers increases.
>
> Again, we acknowledge that non-element-wise operators (e.g., GAT) are important GNN models. However, how to **reduce the communication cost and avoid raw neighbor feature transmission** for non-element-wise operators in the federated setting is a fundamentally hard problem, which would require major architectural design changes in non-element-wise operators rather than straightforward adaptation. Clearly, this is far beyond the scope of this work, and even the feasibility of modifying non-element-wise operators for the federated setting itself is substantial enough to merit a separate study.
>
>
> [1] Veličković, P., Cucurull, G., Casanova, A., Romero, A., Liò, P., & Bengio, Y. (2018, February). Graph Attention Networks. In International Conference on Learning Representations.

---

> ### Author Response · Authors · 2025-11-22
>
> > **Weakness 2:** The convergence analysis is proved for GCN, whereas the experiments emphasize GraphSAGE/GIN. The paper does not establish conditions under which the analysis extends to these architectures. Moreover, the error neighborhood explicitly grows with depth $L$, but the paper offers no practical prescription for training deeper models without degrading guarantees.
>
> **Response:** Thanks for your comment. We chose GCN for the theoretical analysis due to its mathematical simplicity and tractability. However, the key insights and bounds (e.g., the bias introduced by neighbor sampling and missing cross-client neighbors) **extend** to other GNN architectures. In particular, our convergence analysis is also applicable to GraphSAGE and GIN under similar assumptions. The differences would only affect the proof of Lemmas 5.4–5.5, thereby *only influencing the exact form of the bias term in the convergence result.*
>
> The main challenge in extending the theoretical analysis to GraphSAGE/GIN lies in handling non-linear and heterogeneous aggregation functions, which are more prominent in GraphSAGE (e.g., max-pooling, LSTM) and GIN (e.g., MLP-based injective updates). These functions introduce additional sources of nonlinearity and variance in the layer-wise error propagation, making it harder to bound the bias and variance of the stochastic gradients.
>
> Even with the relatively simple GCN architecture, the theoretical analysis remains **highly non-trivial** due to the presence of **biased stochastic gradients** and the **structural entanglement** inherent to graph convolutions. Unlike prior works that rely on strong and often unrealistic assumptions (e.g., unbiased or consistent gradient estimators), our analysis is, to our knowledge, the first in the FGL literature to rigorously bound the approximation error of biased stochastic gradients. This advance provides theoretical insights *of independent interest,* beyond the specific scope of Swift-FedGNN.
>
> Regarding the error neighborhood's dependency on GNN depth $L$, we agree this is an important practical concern. However, this phenomenon stems from a **fundamental limitation of GNN architectures**, rather than a limitation of our Swift-FedGNN design. Mitigating this issue typically requires **architecture-specific enhancements**, which are orthogonal to the scope of this work. That said, we note that many of the existing GNN architectural solutions (e.g., [1]) are compatible with Swift-FedGNN and can be integrated to address depth-related degradation in practice.
>
> [1] Chen, T., Zhou, K., Duan, K., Zheng, W., Wang, P., Hu, X., & Wang, Z. (2022). Bag of tricks for training deeper graph neural networks: A comprehensive benchmark study. IEEE Transactions on Pattern Analysis and Machine Intelligence, 45(3), 2769-2781.
>
>
> > **Weakness 3:** The empirical comparison is limited to three or four methods. Important FGL variants—such as SpreadGNN and FedGCN—are not included in the core efficiency comparisons, despite being cited.
>
> **Response:** Thanks for your comment. We would like to clarify that the baselines used in our experiments were selected because they are the **most closely aligned with the goals** of our proposed algorithm. Specifically, these baselines focus on reducing communication costs in federated graph learning and mitigating information loss from cross-client neighbors through periodic (sampling-based) full-neighbor training, which makes them directly comparable to the design of Swift-FedGNN.
>
> We omitted FedGCN from our core comparisons because it performs a one-time exchange of full cross-client neighbor information prior to training, which restricts it to **full-graph training** with centralized preprocessing. This strategy incurs significant per-client memory overhead and is **impractical for large-scale graphs**, which are the primary focus of our study. Moreover, directly applying FedGCN’s techniques to our sampling-based training setting would require frequent, large-scale transmission of encrypted graph features, incurring substantial communication overhead and rendering the method infeasible in practice.
>
> As for SpreadGNN, it is not directly comparable to Swift-FedGNN for two key reasons: i) It is designed for a **fully decentralized setting** without a central server, which is a fundamentally different system model from ours; and ii) It is specifically developed for **molecular graph datasets**, which are *graph-level* tasks where cross-client neighbors do not exist. As a result, it does not address the same communication challenges that are central to our framework.
>
> We appreciate the opportunity to clarify this and will revise our paper to remove the citation of SpreadGNN, as it does not align well with our problem setting.

---

> ### Author Response · Authors · 2025-11-22
>
> > **Weakness 4:** The method section largely describes a clean but straightforward system design. Without a stronger learning-algorithm contribution (e.g., new estimators or privacy mechanisms), the work can read as primarily an engineering improvement.
>
> **Response:** Thanks for your comment. While we agree that the system design of Swift-FedGNN emphasizes simplicity and modularity, we would like to emphasize that our work goes **far beyond** an engineering improvement and offers significant **algorithmic contribution**.
>
> Specifically, Swift-FedGNN introduces **nontrivial algorithmic and theoretical innovations** tailored to the unique challenges of federated graph learning (FGL). We summarize our theoretical and algorithmic contributions as follows:
>
> 1. **Novel FGL Framework with Two-Stage Aggregation:** Our proposed “aggregate-then-transfer” mechanism is not merely an engineering tweak, but a design tailored to the FGL setting, balancing communication efficiency, privacy preservation (by avoiding raw feature transmission), and scalability.
> 2. **Federated Scheduling Design for Periodic Cross-Client Training:** We introduce a lightweight yet effective scheduling strategy for periodic cross-client training, enabling significant communication cost reductions without sacrificing model accuracy. Importantly, this algorithmic design is **not a heuristic** since it offers **strong theoretical finite-time convergence rate guarantee** and is validated across five real-world datasets.
> 3. **New Theoretical Analysis of Biased Stochastic Gradients in GNNs:** To our knowledge, our work is the first to rigorously analyze the convergence of FGL with **biased stochastic gradients** arising from both neighbor sampling and missing cross-client neighbors, rather than relying on strong assumptions such as unbiased or consistent gradients as in prior works. We explicitly bound the gradient approximation errors (Lemmas 5.4–5.5) and discover their positive correlation with GNN depth, a structural property unique to graph models. This leads to a new convergence guarantee (Theorem 5.6) that matches the state-of-the-art convergence rate for sampling-based GNN training, despite operating under the more restrictive federated setting.
>
>
> Collectively, these elements go beyond a pure system engineering effort. Swift-FedGNN delivers a principled algorithmic framework for federated GNN training under practical constraints, with theoretical rigor and strong empirical results across five real-world datasets.
>
>
>
> > **Weakness 5:** Minor typos: "Communicaiton" -> "Communication" in Figs 6 and 7
>
> **Response:** Thanks for pointing this out. We have corrected these typos in the revised version of our paper.

---

> ### Author Response · Authors · 2025-11-22
>
> > **Question 1:** The background section cites the Microsoft Academic Graph (MAG, 100 M nodes) as an example of large-scale, geo-distributed graphs, yet experiments stop at ogbn-products (2.4 M nodes). Could the authors clarify why MAG or an equivalently large dataset was not used? Was this due to memory limits, partitioning complexity, or cross-client sampling constraints?
>
> **Response:** Thanks for your question. We agree that evaluating on large-scale datasets such as MAG (100M nodes) would further strengthen the empirical support for Swift-FedGNN. While we did consider including MAG in our evaluation, several practical factors prevented us from doing so:
>
> * **Resource Limitations:** Although Swift-FedGNN is communication-efficient, training on a dataset of MAG’s scale requires substantial GPU memory for storing graph features and training, CPU memory for graph partitioning, and other computational resources. Our current server infrastructure could not support tasks at this scale.
> * **Preprocessing and Partitioning Complexity:** MAG is a heterogeneous graph with multiple node and edge types, which introduces significant complexity in preprocessing and partitioning for federated settings.
>
> Instead, we chose ogbn-products (2.4M nodes) as the largest dataset in our experiments because it already presents substantial challenges for federated graph learning, including a high node count, a dense edge structure (25.8 edges per node), and significant communication overhead during cross-client neighbor aggregation. Our current results on ogbn-products already demonstrate the scalability and communication efficiency of Swift-FedGNN in challenging settings.
>
>
>
> > **Question 2:** The main theoretical analysis explicitly targets GCNs. Can the authors discuss whether and how this framework extends to GraphSAGE, GIN, or attention-based models (e.g., GAT)? In particular, how would the error neighborhood term scale with network depth $L$ and non-linear aggregation functions?
>
> **Response:** Thanks for your questions. We address multiple distinct questions in your comment as follows:
>
> * **1) Extension to GraphSAGE and GIN:** Regarding the extension of our theoretical analysis to GraphSAGE and GIN, to avoid repetition, please refer to our detailed response to your *Weakness 2*. For attention-based models (e.g., GAT) not considered in the design of Swift-FedGNN, to avoid repetition again, please see our response to your *Weakness 1* for a more thorough discussion.
>
> * **2) Error Neighborhood Term Scaling:** As for the scaling behavior of the error neighborhood term with respect to network depth $L$ and aggregation functions: our analysis (Lemmas 5.4–5.5 and Theorem 5.6) shows that both the sampling-induced gradient bias and the error from missing cross-client neighbors grow with $L$. This is attributed to the recursive nature of neighbor aggregation and the stacking of nonlinear transformations across layers. While this result is derived under the GCN architecture, we expect an **order-wise similar trend** for GraphSAGE and GIN, where neighbor aggregation is still layer-wise and interleaved with nonlinear transformations. The precise scaling behavior may vary depending on the Lipschitz continuity, smoothness constants, or other properties of the respective aggregation functions and MLPs used, which could be incorporated with additional assumptions.
>
>
> > **Question 3:** Since the theory demonstrates that gradient bias increases with depth $L$, have the authors observed this empirically? For example, does validation accuracy degrade more rapidly for deeper architectures under the same $I,K$ settings?
>
> **Response:** Thanks for your questions. To empirically support our theoretical claim that the stochastic gradient bias increases with GNN depth, in this rebuttal period, we directly measured the gradient bias between the full gradient and the stochastic gradient on the ogbn-arxiv dataset under varying GNN depths. This bias metric provides a more direct and interpretable performance evaluation than validation accuracy for this analysis. The results, summarized in the table below, clearly show that the gradient bias grows as the GNN becomes deeper, aligning well with our theoretical analysis.
>
> Table: Gradient bias under varying GNN depths on the ogbn-arxiv dataset.
> | GNN Depth | 2 | 3 | 4 | 5 |
> | -------- | -------- | -------- | -------- |-------- |
> | Gradient Bias     | 0.46     | 0.78     | 1.12     |1.59|

---

> > ### Comment · Reviewer_MwMc · 2025-11-27
> >
> > Thank you for the detailed response and the revisions. Most of my concerns have been addressed. The remaining concerns are: (i) the formal analysis still only covers GCN, while the experiments cover a broader set of models; and (ii) the method is restricted to element-wise aggregation GNNs, with attention-based models left for future work. Given the new clarifications, the extra experiment on depth-dependent bias, and the overall empirical gains, I will keep my positive score at 6.

---

> > > ### Author Response · Authors · 2025-11-28
> > > **Response to the Two Remaining Concerns (Part 1)**
> > >
> > > > **Comment 1:** Thank you for the detailed response and the revisions. Most of my concerns have been addressed. The remaining concerns are: (i) the formal analysis still only covers GCN, while the experiments cover a broader set of models; and (ii) the method is restricted to element-wise aggregation GNNs, with attention-based models left for future work. Given the new clarifications, the extra experiment on depth-dependent bias, and the overall empirical gains, I will keep my positive score at 6.
> > >
> > >
> > > **Response:** Thank you again for your thoughtful feedback and for maintaining your positive score. We sincerely appreciate your acknowledgment of the clarifications, the added depth-dependent bias experiments, and the empirical strength of our results.
> > >
> > > Regarding the two remaining concerns, we would like to further clarify as follows:
> > >
> > > **1) On GCN-only theory vs. broader experiments:** Our choice of GCN for the theoretical analysis stems from its tractability under biased stochastic gradients, which allows us to rigorously characterize the impact of neighbor sampling and missing cross-client neighbors on convergence. To our knowledge, this is **the first convergence guarantees** for federated GNN training **without assuming unbiased or constant-bias gradient estimators**. The analysis remains highly non-trivial due to the presence of biased stochastic gradients and the structural entanglement inherent to GNNs.
> > >
> > > We fully agree that extending the formal analysis to a broader set of models, such as GraphSAGE or GIN, would be valuable. However, the GCN-style message-passing framework we have analyzed captures the **core architecture** of many **element-wise aggregation-based GNNs**. Importantly, the key theoretical insights *(e.g., the residual error scaling with the correction frequency $I$ and the client sampling size $K$)* **persists beyond GCN**. Specifically, our convergence analysis also applies to GraphSAGE and GIN under similar assumptions. The primary differences would lie in the proofs of Lemmas 5.4–5.5, thereby affecting only the exact form of the bias term in the convergence result. The applicability of our results in GraphSAGE and GIN is also supported by our empirical results, where similar trends hold across multiple GNN architectures.
> > >
> > > GraphSAGE and GIN introduce additional sources of nonlinearity and variance through pooling or MLP-based injective updates. These added complexities make the analysis more challenging, but they do not invalidate the overall bounding strategy and theoretical trends.
> > >
> > > To understand why our theoretical results continue to hold in GraphSAGE and GIN, we provide a detailed explanation in *Appendix I* of the revised paper, outlining how the analysis can be extended to these architectures. By replacing the GCN updates with the corresponding update rules for GraphSAGE or GIN, we can derive similar upper bounds on the gradients. The difference only lies in the mathematical form of these bounds, which depend on the respective GNN architectures (e.g., GraphSAGE or GIN), but not their asymptotic behavior. Substituting these bounds into the proof of Theorem 5.6 in *Appendix H.4* yields similar convergence trends for both GraphSAGE and GIN:
> > > $$\frac{1}{T} \sum_{t=0}^{T-1} \left\|\nabla \mathcal{L}\left(\theta_{t}\right)\right\|^2  \le \frac{2\left( \mathcal{L}\left( \theta_{0} \right)-\mathcal{L}\left( \theta^*\right) \right)}{\sqrt{MT}}+\left(C_{\Delta G}^l+C_{\Delta G}^r\right)^2+\frac{K}{IM} \left( \left(C_{\Delta G}^{f}\right)^2-\left(C_{\Delta G}^l+C_{\Delta G}^r\right)^2 \right),$$ where $C_{\Delta G}^{l}$, $C_{\Delta G}^{f}$, and $C_{\Delta G}^{r}$ are constants that depend on the respective GNN architectures (e.g., GraphSAGE or GIN).
> > >
> > >
> > > While a complete formal extension to GraphSAGE/GIN would be valuable, we believe our current analysis, along with the guidance provided for extending it, offers meaningful insights that are already indicative of broader behaviors. The theoretical convergence analysis provides formal performance guarantees for Swift-FedGNN, and the extensive experiments further demonstrate its effectiveness across a range of GNN architectures.
> > >
> > > We continue our response to the two remaining concerns in the next block.

---

> > > ### Author Response · Authors · 2025-11-28
> > > **Response to the Two Remaining Concerns (Part 2)**
> > >
> > > Continuing our response to the two remaining concerns:
> > >
> > > **2) On restriction to element-wise aggregation GNNs:** You are correct that our current framework is designed for element-wise aggregation GNNs. However, such a restriction is **not due to** the limitation of our Swift-FedGNN framework. Rather, we make a deliberate choice to avoid adapting GAT due to its high communication costs in raw feature exchanges in federated graph learning. As a result, we **never** intend to incorporate non-element-wise aggregation GNNs such as GAT into our current framework. The evidence can be seen in the **Main Contributions** part in the **Introduction** section (cf. Lines 95 -- 99):
> > >
> > > *"The cross-client neighbor information is aggregated at remote clients before communicating to the server and accumulated one more time before transferring to the training client, further minimizing data transfer cost and enhancing privacy by ensuring only aggregated neighborhood features - **never raw node features** - are exchanged"*
> > >
> > > The paragraph above makes our intention **unmistakably clear**: In order to achieve high communication efficiency in graph learning, we want to **avoid** raw node feature exchanges. This already **preclude the use of non-element-wise aggregation GNNs** in our Swift-FedGNN framework.
> > >
> > > Also, in our response to your comments in the previous round, we had provided a detailed quantitative analysis of the communication cost of integrating GAT into our framework (denoted as GAT‑Swift‑FedGNN). This analysis **quantitatively and precisely explains why GAT extension was not considered**: GAT requires direct access to raw neighbor features/embeddings to compute attention scores, which leads to **orders-of-magnitude higher communication cost** under mini-batch training on large-scale graphs. This limitation is **not unique to Swift-FedGNN but is inherent to any communication-efficient federated graph learning design**.
> > >
> > >
> > > To concretely illustrate this, we again explain why we never intend to consider the GAT-adaptation of Swift-FedGNN (i.e., the hypothetical GAT-Swift-FedGNN extension) using the following to summarize and contrast Swift-FedGNN and GAT-Swift-FedGNN:
> > > |  | Swift-FedGNN | GAT‑Swift‑FedGNN (Hypothetical Extension) |
> > > | -------- | -------- | -------- |
> > > | **Cross-client information transmitted**     | Aggregated neighbor features/embeddings     | Raw neighbor features/embeddings     |
> > > |**Communication cost per cross-client training round for exchanging embeddings**|$\mathcal{O}\left(KB\sum_{l=1}^{L}p_{(l)}F^ld_{(l-1)}^{emb}\right)$|$\mathcal{O}\left(KB\sum_{l=1}^{L}C_{(l)}d_{(l-1)}^{emb}\right)$|
> > > |**Efficiency implication**|Communication‑efficient (since $C_{(l)} \ll p_{(l)}F^l$)|Communication‑inefficient; cost grows exponentially with GNN depth|
> > > |**Intended use case**|Federated settings with limited/realistic bandwidth constraints |Federated settings with unconstrained or very large bandwidth (not the focus of our work)|
> > >
> > > Supporting GAT in a **communication‑efficient** manner in federated graph learning settings would require **substantial additional architectural innovations**, such as compressed attention or proxy‑based attention, each of which constitutes its own research problem. We believe such developments merit a separate line of work and are valuable for future exploration.
> > >
> > > In summary, the theoretical and empirical generality across multiple element-wise-aggregation-based GNN models provide strong evidence that our insights are broadly applicable in federated graph learning. We sincerely thank you again for your thoughtful and constructive feedback.

---

### Official Review · Reviewer_Aug9 · 2025-11-01

**Soundness:** 2
**Presentation:** 2
**Contribution:** 2
**Rating:** 4
**Confidence:** 4

**Summary:**

This paper presents Swift-FedGNN, a federated graph learning framework addressing cross-client neighbor dependencies and high communication costs in GNN training. Key contributions: periodic cross-client training for sampled clients (most iterations use efficient local training), aggregated neighbor feature exchange, and rigorous theoretical analysis. Experiments on real-world datasets show Swift-FedGNN achieves competitive accuracy while reducing communication overhead significantly compared to baselines, offering an efficient solution for privacy-preserving federated graph learning.

**Strengths:**

1. Innovative Dynamic Training Framework Balances Efficiency and Accuracy
The paper proposes a "local training-dominated, periodic cross-client training-supplemented" mechanism: clients mainly perform parallel local training to reduce overhead, and only randomly sample a subset of clients for cross-client training at intervals to avoid information loss. This design addresses the inefficiency of traditional FedGNN methods and the accuracy degradation from ignoring cross-client neighbors, achieving a rational trade-off between efficiency and performance.

2. Rigorous Theoretical Analysis Breaks Traditional Assumptions
Unlike existing FedGNN works that rely on strong assumptions for convergence analysis, this paper directly bounds the approximation error of biased stochastic gradients in GNNs, a unique challenge in federated graph learning, and reveals the positive correlation between gradient bias and network depth. It further proves Swift-FedGNN achieves an $O(T^{-1/2})$ convergence rate, matching the SOTA of centralized sampling-based GNNs in complex federated scenarios, providing solid theoretical support.

3. Notable Experimental Efficiency Advantages
Extensive experiments on real-world datasets show Swift-FedGNN outperforms baselines in efficiency: it reduces communication overhead by 7-21x compared to FedGNN-G on 4/5 datasets, and completes cross-client training in <200ms on ogbn-products. Meanwhile, its validation accuracy is comparable to FedGNN-G, verifying the method’s practical value.

**Weaknesses:**

1. Lack of Validation for GAT, a Widely Used GNN
The paper only validates performance on GraphSAGE and GIN. For GAT, it merely notes "remote clients need to transmit raw features for aggregation" but provides no experimental data. It fails to discuss whether raw feature transmission undermines the "aggregation-driven overhead reduction" advantage or how accuracy is affected, limiting generality for complex GNN architectures.

2. Insufficient Discussion on Key Hyperparameters
While the paper shows the trend of hyperparameters on performance, it lacks in-depth analysis of critical issues: e.g., optimal I/K criteria across datasets, the adaptive ratio of K to total client count M, and robustness to small parameter fluctuations. This hinders practical parameter tuning and weakens reproducibility.

3. Narrow Experimental Scenarios
Experiments only use 10-20 clients and do not test large-scale scenarios, leaving uncertainty about scheduling overhead and server latency at scale. Additionally, data partitioning is limited to METIS-balanced and random modes, ignoring typical non-IID scenarios in federated learning. No analysis of data imbalance’s impact on training limits conclusion generalizability.

**Questions:**

1. For GAT, do you have specific designs to adapt the "two-layer aggregation" mechanism? If raw features must be transmitted, what are the expected communication overhead and accuracy loss, and are there plans for supplementary experiments?

2. Non-IID data is a core federated learning challenge. Do you have preliminary ideas to optimize Swift-FedGNN for non-IID scenarios? Are there plans to supplement related experiments?

---

> ### Author Response · Authors · 2025-11-22
>
> > **Weakness 1:** Lack of Validation for GAT, a Widely Used GNN The paper only validates performance on GraphSAGE and GIN. For GAT, it merely notes "remote clients need to transmit raw features for aggregation" but provides no experimental data. It fails to discuss whether raw feature transmission undermines the "aggregation-driven overhead reduction" advantage or how accuracy is affected, limiting generality for complex GNN architectures.
>
> **Response:** Thank you for your comment. We would like to clarify why we do not consider GAT in the design and experiments of our communication-efficient federated GNN algorithm. We are fully aware that GAT is a highly popular GNN model, both in the literature and in practice. However, GAT is **fundamentally not a good fit** in any communication-efficient federated GNN algorithm design. The reason is that GAT requires direct access to **raw neighbor features/embeddings** to compute attention weights based on **nonlinear** pairwise interactions (see Eq. (1) in [1]). This requirement necessitates transmitting raw neighbor features/embeddings across clients, which leads to significantly high communication overhead. In other words, it is **impossible** for GAT to leverage the same communication-efficient aggregated transmissions as in those GNN models based on linear weighted aggregations (e.g., GCN-type models). As a result, GAT is not an ideal GNN model choice in those federated GNN algorithm design settings, where communication efficiency is of utmost importance.
>
> To further quantitatively understand GAT's communication efficiency limitation in federated GNN algorithm design, we analyze the communication cost of incorporating GAT into Swift-FedGNN (denoted as GAT-Swift-FedGNN) in here and compare it with our original Swift-FedGNN design. Throughout the analysis, we assume an $L$-layer GNN and the following parameters:
> * $M$: The total number of clients;
> * $F$: The same number of neighbor sampling fan-out used at each layer;
> * $F^{l}$: The worst-case number of neighbors at each training node at each GNN layer $l\in[1,L]$ using $F$-fan-out;
> * $p_{(l)} \in (0, 1)$: The fraction of the neighbors that are located on other clients.
>
> **1. GAT-Swift-FedGNN:** Every $I$ iterations, each of the $K$ sampled clients performs cross-client training and exchanges **raw features/embeddings** for its cross-client neighbors. The total communication cost per cross-client training round for exchanging these embeddings is on the order of:
> $$\mathcal{O}\left(KB\sum_{l=1}^{L}p_{(l)}F^ld_{(l-1)}^{emb}\right),$$ where $B$ is the batch size per client, $d_{(l)}^{\text{emb}}$ is the embedding (hidden) dimension at layer $l$, and $F^l$ reflects the exponential expansion in sampled neighborhoods as the layer depth increases.
>
> **2. Swift-FedGNN:** In contrast, Swift-FedGNN avoids transferring raw features/embeddings by sharing **aggregated neighbor features/embeddings**. If the $p_{(l)}F^l$ cross-client neighbors at layer $l$ are distributed across $C_{(l)} < M$ remote clients, then after aggregation, the communication cost is on the order of:
> $$\mathcal{O}\left(KB\sum_{l=1}^{L}C_{(l)}d_{(l-1)}^{emb}\right).$$
>
> Since $C_{(l)} \ll p_{(l)}F^l$, Swift-FedGNN achieves significantly lower communication overhead than the GAT variant.
>
>
> From the above analysis, we can see that GAT-Swift-FedGNN incurs a communication cost that is not only $F^l$ times higher than Swift-FedGNN, but the gap between them also grows **exponentially** as the number of layers increases.
>
> Again, we acknowledge that GAT is an important GNN model. However, how to reduce the communication cost and avoid raw neighbor feature transmission for GAT in the federated setting is a fundamentally hard problem, which would require major architectural design changes in GAT rather than straightforward adaptation. Clearly, this is far beyond the scope of this work, and even the feasibility of modifying GAT for the federated setting itself is substantial enough to merit a separate study.
>
>
> [1] Veličković, P., Cucurull, G., Casanova, A., Romero, A., Liò, P., & Bengio, Y. (2018, February). Graph Attention Networks. In International Conference on Learning Representations.

---

> ### Author Response · Authors · 2025-11-22
>
> > **Weakness 2:** Insufficient Discussion on Key Hyperparameters.
>
> **Response:** Thanks for your comment. We agree that a more in-depth analysis of key hyperparameters would enhance both the practical utility and reproducibility of our work. To address this, in this rebuttal period, we have conducted a detailed investigation on how different correction frequencies ($I$) and client sampling sizes ($K$) affect model performance on the ogbn-products dataset (see Tables 1-4).
>
> **1. Impact on Model Accuracy:** Tables 1 and 2 report the final validation accuracy of Swift-FedGNN under varying $I$ and $K$ settings, respectively. The results demonstrate that:
> * Increasing $I$ from 5 to 40 (Table 1) leads to only a minor accuracy degradation (from 88.91% to 88.44%), demonstrating that less frequent cross-client training still preserves model quality.
> * Decreasing $K$ from 15 to 1 (Table 2) also results in a minor accuracy drop (from 89.22% to 88.47%), indicating that a small number of sampled clients is sufficient to maintain strong performance.
>
> These findings are consistent with our theoretical conclusion in Remark (3) in Line 406 on Page 8.
>
> **2. Impact on Communication Cost:** Tables 3 and 4 present the total communication cost needed to achieve a target accuracy of **87%** under the same parameter variations. The results show that:
> * Increasing $I$ and decreasing $K$ substantially reduce communication cost. For example, raising $I$ from 5 to 40 saves approximately **80%** of the communication overhead, while reducing $K$ from 15 to 1 saves approximately **94%**.
>
> Collectively, these results validate Swift-FedGNN’s ability to reduce communication without incurring major information loss and demonstrate that Swift-FedGNN provides a tunable balance between communication efficiency and accuracy, and the trade-off can be controlled via $I$ and $K$.
>
>
> **3. Practical Guidance on Parameter Choices and Robustness:** The flexibility of Swift-FedGNN having two "control knobs" in $I$ and $K$ makes it well-suited for practical deployment under varying resource constraints. Practitioners can choose $I$ and $K$ based on their specific system budgets and application requirements. Specifically, for resource-constrained clients, using a larger $I$-value (less frequent cross-client training) and a smaller $K$-value (fewer sampled clients) can significantly reduce communication without major accuracy loss. Conversely, for accuracy-critical applications with more communication budget, a smaller $I$-value and a larger $K$-value offer the best accuracy.
>
> We emphasize that there is **no universal "optimal" setting** for $I$ and $K$, and the ideal configuration depends on the specific deployment context. Moreover, our results demonstrate that Swift-FedGNN is **robust to a wide range of $I$- and $K$-values**, with minimal degradation in validation accuracy. This robustness supports practical parameter tuning and facilitates reproducible deployment across heterogeneous systems.
>
>
> **4. Fixed vs. Adaptive Ratio of $K$ to $M$:** In Swift-FedGNN, we adopt a fixed ratio of $K$ to total client count $M$, which is a standard practice in both general federated learning and federated graph learning literature. Despite this simplicity, it already suffices to offer strong **theoretical convergence guarantees** and **significant communication cost reductions**, which constitute the primary goals of this work. Moreover, Swift-FedGNN consistently demonstrates strong performance across datasets.
>
> On the other hand, while adapting the $K/M$ ratio dynamically may further improve efficiency or performance, such mechanisms would require additional algorithmic complexity and design (e.g., client utility scoring, adaptive scheduling), which is orthogonal to our main focus on communication complexity in this paper and better suited as an independent line of future work. Last but not least, it is worth pointing out that the **modular nature** of Swift-FedGNN naturally supports such adaptive extensions, and the contributions of our work remain significant even under the current design.
>
>
> Table 1: Validation accuracy of Swift-FedGNN on ogbn-products with different $I$ and fixed $K = 10$.
> |Correction Frequencies ($I$)|5|10|20|40|
> |-|-|-|-|-|
> |Validation Accuracy (%)|88.91|88.88|88.60|88.44|
>
> Table 2: Validation accuracy of Swift-FedGNN on ogbn-products with different $K$ and fixed $I = 10$.
> |# of Sampled Clients ($K$)|1|5|10|15|
> |-|-|-|-|-|
> |Validation Accuracy (%)|88.47|88.72|88.88|89.22|
>
> Table 3: Communication cost of Swift-FedGNN with different $I$ on ogbn-products when achieving a target validation accuracy of 87%.
> |Correction Frequencies ($I$)|5|10|20|40|
> |-|-|-|-|-|
> |Communication Cost (MB)|1344.0|675.5|324.5|275.0|
>
> Table 4: Communication cost of Swift-FedGNN with different $K$ on ogbn-products when achieving a target validation accuracy of 87%.
> |# of Sampled Clients ($K$)|1|5|10|15|
> |-|-|-|-|-|
> |Communication Cost (MB)|57.8|342.2|675.5|1027.4|

---

> ### Author Response · Authors · 2025-11-22
>
> > **Weakness 3:** Narrow Experimental Scenarios Experiments only use 10-20 clients and do not test large-scale scenarios, leaving uncertainty about scheduling overhead and server latency at scale. Additionally, data partitioning is limited to METIS-balanced and random modes, ignoring typical non-IID scenarios in federated learning. No analysis of data imbalance’s impact on training limits conclusion generalizability.
>
> **Response:** Thanks for your comment. We would like to respond to two distinct aspects in this comment point-by-point as follows:
>
> **1) Large-scale Scenarios:** In this rebuttal period, we have extended our experiments to two large-scale settings with **80 clients** and **100 clients** to more thoroughly evaluate scalability. Tables 1 and 3 below report the total communication cost on ogbn-products dataset when achieving a target validation accuracy, and Tables 2 and 4 present the final validation accuracy. These results show that:
> * With $K=10$, Swift-FedGNN reduces total communication cost by **at least 82%** in the 80-client setting and **at least 83%** in the 100-client setting compared to all baselines.
> * With $50\\%$ client sampling, Swift-FedGNN still achieves **at least 38%** communication savings in the 80-client setting and **at least 23%** communication savings in the 100-client setting over all baselines.
> * In both settings, Swift-FedGNN maintains comparable or better validation accuracy, with larger $K$ yielding slightly improved performance.
>
> The above findings highlight Swift-FedGNN’s effectiveness and scalability, validating its communication efficiency even in large-scale federated graph learning settings. We also emphasize that Swift-FedGNN is designed to reduce communication overhead while maintaining competitive accuracy, not to surpass baselines in absolute accuracy.
>
> **2) Non-IID Scenarios:** Regarding the comment on typical non-IID scenarios in federated learning (FL), we would like to clarify that graph data are **fundamentally different** from traditional feature-label-type datasets (e.g., image or text datasets), and the conventional notion of non-IID data distributions in conventional FL is **not applicable** to federated GNN. The **fundamental reason** is that, in traditional FL, each client has a *full copy* of the model, while the dataset is partitioned and distributed at each client. Hence, traditional FL falls into the "**data parallelism**" paradigm. In stark contrast, both the graph neural network model and the graph dataset in federated GNN are **partitioned** and distributed at each client. That is, **none of the clients has a full copy of the graph neural network model**. Hence, federated GNN falls into the "**model parallelism**". Thus, the notion of "non-IID" dataset is no longer applicable in federated GNN. In fact, in federated GNN, the graph data at each client are inherently **non-identical**, since the subgraph data at each client are completely different in general.
>
>   Moreover, graph data also follow **structural properties**, such as power-law or non-power-law degree distributions, and our experimental datasets reflect a range of these characteristics by considering two graph partitioning strategies: 1) The METIS-based partitioning generates connectivity-aware and balanced subgraphs; and 2) the **random partitioning** strategy introduces heterogeneity by randomly assigning nodes to different subgraphs. This second partitioning approach **implicitly induces non-identical and structurally unbalanced local graphs**, thereby simulating **"non-IID" client data** in the context of federated graph learning.
>
>
> Table 1: Total communication cost on ogbn-products dataset with **80 clients** when achieving a target validation accuracy of 83%.
>
> ||Swift-FedGNN (K/M=10/80)|Swift-FedGNN (K/M=40/80)|FedGNN-G|LLCG|FedGNN-PNS|
> |-|-|-|-|-|-|
> |Total Communication Cost (MB)|711.5|2476.8|38154.2|3988.8|4601.8|
>
>
> Table 2: Final validation accuracy (%) on ogbn-products dataset with **80 clients**.
>
> ||Swift-FedGNN (K/M=10/80)|Swift-FedGNN (K/M=40/80)|FedGNN-G|LLCG|FedGNN-PNS|
> |-|-|-|-|-|-|
> |Final Validation Accuracy (%)|85.74|86.12|86.52|83.54|85.67|
>
>
> Table 3: Total communication cost on ogbn-products dataset with **100 clients** when achieving a target validation accuracy of 84.3%.
>
> ||Swift-FedGNN (K/M=10/100)|Swift-FedGNN (K/M=50/100)|FedGNN-G|LLCG|FedGNN-PNS|
> |-|-|-|-|-|-|
> |Total Communication Cost (MB)|1196.0|5367.7|60419.5|6987.4|8880.5|
>
>
> Table 4: Final validation accuracy (%) on ogbn-products dataset with **100 clients**.
>
> ||Swift-FedGNN (K/M=10/100)|Swift-FedGNN (K/M=50/100)|FedGNN-G|LLCG|FedGNN-PNS|
> |-|-|-|-|-|-|
> |Final Validation Accuracy (%)|85.07|85.53|85.63|84.35|85.15|

---

> ### Author Response · Authors · 2025-11-22
>
> > **Question 1:** For GAT, do you have specific designs to adapt the "two-layer aggregation" mechanism? If raw features must be transmitted, what are the expected communication overhead and accuracy loss, and are there plans for supplementary experiments?
>
> **Response:** Thanks for your questions. To avoid repetition, please refer to our response to your *Weakness 1* for a detailed discussion on GAT in federated settings and adapting Swift-FedGNN to GAT.
>
> > **Question 2:** Non-IID data is a core federated learning challenge. Do you have preliminary ideas to optimize Swift-FedGNN for non-IID scenarios? Are there plans to supplement related experiments?
>
> **Response:** Thanks for your question. We would like to emphasize that non-IID scenarios are already simulated inherently through our random partitioning strategy. To avoid repetition, for more detailed explanations, please refer to our response to your *Weakness 3*.

---

> > ### Comment · Reviewer_Aug9 · 2025-11-25
> > **Reply for Rebuttal**
> >
> > Thank you for your rebuttal.
> >
> > Your defense for excluding GAT is unconvincing and self-contradictory. Your paper’s Footnote 1 (Lines 268-269) explicitly proposes an adaptation for GAT, yet your rebuttal analyzes this exact method only to dismiss it as inefficient. This appears to be a post-hoc justification for an experimental omission, suggesting the framework's efficiency is brittle and confined to a narrow class of GNNs. A top-tier paper must transparently address such limitations, not define them away as "out of scope." Why was this critical experiment, which would have quantified the trade-off for attention-based models, not conducted and reported?
> >
> > The new hyperparameter analysis is superficial and reveals a critical disconnect between your theory and empirical results. Claiming qualitative "consistency" with your theory is insufficient. A rigorous paper must demonstrate that its theoretical bounds are not vacuous by quantitatively linking them to practical outcomes. Without this, the "rigorous theoretical analysis" appears detached from the algorithm's actual behavior. Can you provide a concrete analysis mapping the empirical accuracy degradation from varying I and K to the quantitative change in the residual error term in Theorem 5.6?
> >
> > Most critically, your rebuttal demonstrates a fundamental misunderstanding of federated learning. Your claim that the concept of non-IID is "not applicable" is factually incorrect; FL is a data-parallel paradigm where statistical heterogeneity is a central challenge. Furthermore, your assertion that random partitioning simulates non-IID is the opposite of established practice, as it is the standard method for approximating an IID setting. Consequently, your experiments were performed only under the most benign, unrealistic FL conditions, severely undermining the paper's claimed relevance. How can the IID-centric convergence guarantees and reported results be considered meaningful for real-world federated applications, which are inherently non-IID?

---

> > > ### Author Response · Authors · 2025-11-27
> > > **Response to Follow-Up Comment on Exploring GAT Extensions**
> > >
> > > > **Comment 1:** Your defense for excluding GAT is unconvincing and self-contradictory ...
> > >
> > > **Response:** Thanks for your follow-up comments. We respectfully disagree that our explanation is self-contradictory and a "post-hoc justification." In the footnote in our original submission, we wrote:
> > >
> > > *"To support non-element-wise operation, e.g.,, GAT, each remote client can transfer the raw graph features or activations to the server for aggregation, instead of performing the locally partial aggregation first."*
> > >
> > > We acknowledge that this statement may be misleading. But its intent is merely to suggest **a possible way to accommodate the exchange of raw graph features**; it **was not meant to** imply that we planned to adapt Swift-FedGNN to support GAT. Indeed, as seen in our Algorithms 1-3, **no such "GAT adaptation" exists** in our Swift-FedGNN framework. Moreover, the original footnote **never claims** that this "GAT adaptation" is communication-efficient, which is the goal of our work. In summary, we **never** intend to incorporate GAT into our current framework.
> > >
> > > **Further evidence** that we never intended to incorporate GAT into our framework can be seen in the **Main Contributions** part in the **Introduction** section (cf. Lines 95 -- 99):
> > >
> > > *"The cross-client neighbor information is aggregated at remote clients before communicating to the server and accumulated one more time before transferring to the training client, further minimizing data transfer cost and enhancing privacy by ensuring only aggregated neighborhood features - **never raw node features** - are exchanged"*
> > >
> > > The paragraph above makes our intention **unmistakably clear**: In order to achieve high communication efficiency in graph learning, we want to **avoid** raw node feature exchanges. This already **preclude the use of GAT** in our Swift-FedGNN framework.
> > >
> > > To further clarify: the footnote describes a mechanism that is only feasible **if one were willing to transmit raw neighbor features or embeddings**, in which GAT-based message passing could be implemented in a federated setting. However, this prerequisite directly contradicts the design objective of any **communication-efficient** federated graph learning algorithms, not just our Swift-FedGNN. Therefore, the footnote merely acknowledges conceptual feasibility, not practical viability, and does not suggest that a GAT-based extension would maintain the communication efficiency central to our approach.
> > >
> > > Also, in the rebuttal, we provided a detailed quantitative analysis of the communication cost of integrating GAT into our framework (denoted as GAT‑Swift‑FedGNN). This analysis was **not intended as a post‑hoc dismissal**, but rather to further elaborate our original footnote. It **quantitatively and precisely explains why GAT-based experiments were not considered**: GAT requires direct access to raw neighbor features/embeddings to compute attention scores, which leads to orders-of-magnitude higher communication cost under mini-batch training on large-scale graphs.
> > >
> > > To concretely illustrate this, we again explain why we never intend to consider the GAT-adaptation of Swift-FedGNN (i.e., the hypothetical GAT-Swift-FedGNN extension) using the following to summarize and contrast Swift-FedGNN and GAT-Swift-FedGNN:
> > > || Swift-FedGNN | GAT‑Swift‑FedGNN (Hypothetical Extension) |
> > > |-|-|-|
> > > | **Cross-client information transmitted**| Aggregated neighbor features/embeddings| Raw neighbor features/embeddings     |
> > > |**Communication cost per cross-client training round for exchanging embeddings**|$\mathcal{O}\left(KB\sum_{l=1}^{L}p_{(l)}F^ld_{(l-1)}^{emb}\right)$|$\mathcal{O}\left(KB\sum_{l=1}^{L}C_{(l)}d_{(l-1)}^{emb}\right)$|
> > > |**Efficiency implication**|Communication‑efficient (since $C_{(l)} \ll p_{(l)}F^l$)|Communication‑inefficient; cost grows exponentially with GNN depth|
> > > |**Intended use case**|Federated settings with limited/realistic bandwidth constraints |Federated settings with unconstrained or very large bandwidth (not the focus of our work)|
> > >
> > > Because our paper’s central contribution is a communication‑efficient framework for federated graph learning under realistic bandwidth constraints, we felt it would be misleading to include a **communication-inefficient** variant (GAT-Swift-FedGNN) as an experimental algorithm. Including such an extension would not reflect realistic constraints and would distract from our main contribution.
> > >
> > > Supporting GAT in a communication‑efficient manner would require **substantial additional architectural innovations**, such as compressed attention or proxy‑based attention, each of which constitutes its own research problem. We believe such developments merit a separate line of work and are valuable for future exploration.
> > >
> > > That said, we appreciate the reviewer’s perspective, and we agree that exploring attention-based extensions is a valuable direction for future research. We have added this point explicitly to the paper as a promising avenue for future work.

---

> > > ### Author Response · Authors · 2025-11-27
> > > **Response to Follow-Up on Empirical Alignment with the Residual Error Term in Theorem 5.6**
> > >
> > > > **Comment 2:** The new hyperparameter analysis is superficial and reveals a critical disconnect between your theory and empirical results. Claiming qualitative "consistency" with your theory is insufficient. A rigorous paper must demonstrate that its theoretical bounds are not vacuous by quantitatively linking them to practical outcomes. Without this, the "rigorous theoretical analysis" appears detached from the algorithm's actual behavior. Can you provide a concrete analysis mapping the empirical accuracy degradation from varying I and K to the quantitative change in the residual error term in Theorem 5.6?
> > >
> > >
> > > **Response:** Thanks for your follow-up comments. We agree that theoretical analysis should ideally offer insights into practical performance and not appear detached from empirical outcomes. Our theoretical result in Theorem 5.6 bounds the residual error (gradient bias) in terms of the communication interval $I$ and the number of sampled clients $K$, showing that increasing $I$ or decreasing $K$ would increase the residual error (Remark (3) of Theorem 5.6). This trend aligns with our empirical observations, as discussed in our response to your *Weakness 2*.
> > >
> > > To provide a more concrete mapping between empirical accuracy degradation and the residual error term in Theorem 5.6, we conduct additional experiments to estimate the realistic magnitudes of the constants involved (see Table 1 below) and report validation accuracy under varying $I$ and $K$ (see Tables 2 and 3 below). The results show that:
> > > * Table 1 indicates that the constants in Theorem 5.6 lead to an estimated residual error term of the form $\mathcal{O}(7.0225-\frac{K}{I}\times 0.68289)$.
> > > * Table 2 fixes $K=5$ and varies $I$, demonstrating that increasing $I$ results in higher residual error and, correspondingly, lower validation accuracy.
> > > * Table 3 fixes $I=10$ and varies $K$, showing that increasing $K$ reduces the residual error and improves validation accuracy.
> > >
> > > These results collectively demonstrate a clear quantitative relationship between the empirical accuracy degradation and the theoretical residual error term in Theorem 5.6, thereby supporting the practical relevance of our theoretical analysis.
> > >
> > >
> > >
> > >
> > > Table 1: Realistic magnitudes of constants in Theorem 5.6 on ogbn-arxiv dataset (with $M=10$ and $L=2$).
> > > | Constant | $LB_{\Delta G}^{l}$ | $LB_{\Delta G}^{r}$ | $LB_{\Delta G}^{f}$ |
> > > | -------- | -------- | -------- |-------- |
> > > | Realistic Magnitude     | 0.46     | 2.19 |0.44     |
> > >
> > >
> > > Table 2: Validation accuracy of Swift-FedGNN on ogbn-arxiv dataset with different $I$ and fixed $K = 5$.
> > > |Correction Frequencies ($I$)|5|10|15|20|40|
> > > |-|-|-|-|-|-|
> > > |Validation Accuracy (%)|57.97| 57.17|56.77|56.62|56.43|
> > >
> > >
> > > Table 3: Validation accuracy of Swift-FedGNN on ogbn-arxiv dataset with different $K$ and fixed $I = 10$.
> > > |# of Sampled Clients ($K$)|1|2|5|8|10|
> > > |-|-|-|-|-|-|
> > > |Validation Accuracy (%)|56.05|56.50| 57.17|57.27|58.05|

---

> ### Author Response · Authors · 2025-11-27
> **Response to Follow-Up Comment on the Misunderstanding of Federated Learning (Part 1)**
>
> > **Comment 3:** Most critically, your rebuttal demonstrates a fundamental misunderstanding of federated learning. Your claim that the concept of non-IID is "not applicable" is factually incorrect; FL is a data-parallel paradigm where statistical heterogeneity is a central challenge. Furthermore, your assertion that random partitioning simulates non-IID is the opposite of established practice, as it is the standard method for approximating an IID setting. Consequently, your experiments were performed only under the most benign, unrealistic FL conditions, severely undermining the paper's claimed relevance. How can the IID-centric convergence guarantees and reported results be considered meaningful for real-world federated applications, which are inherently non-IID?
>
>
> **Response:** Thanks for your follow-up comments. Due to the length of this comment, we would like to respond point-by-point as follows:
>
> **1) Misconception of federated learning:** We respectfully disagree with the assertion that our rebuttal reflects a misunderstanding of federated learning (FL). First of all, the statement "FL is a data-parallel paradigm" is not accurate and it is **only true** for the conventional **horizontal FL** paradigm (i.e., a set of clients collaboratively learn a common model through the coordination from a server, and each client holds a full copy of the model and a subset of the training data).
>
> However, besides horizontal FL, FL could also be **vertical** (i.e., vertical FL) in the sense that different clients hold **different features** of the same data samples. Clearly, since features of the same data sample spread across all clients, vertical FL is **model parallel**, i.e., none of the clients holds a copy of the full model. Vertical FL has found wide range applications where data security is critical (e.g., in banking systems, different fields of a user's data record are located in different servers at different geographical locations).
>
> In what follows, for convenience, we will also refer to the horizontal FL as "traditional FL" or "conventional FL."
>
> **2) Our federated graph learning is model-parallel:** Next, with the understanding that FL could either be data-parallel or model-parallel, we would like to re-emphasize the setting of our federated graph learning systems and clarify why it is **model-parallel**. As mentioned in Section 3 (preliminaries of federated graph learning) in our paper (cf. Lines 166-167):
>
> "*The graph $\mathcal{G}$ is geographically distributed over these clients, and each client $m$ contains a **subgraph** represented by $\mathcal{G}^{m}\left( \mathcal{V}^{m},\mathcal{E}^{m} \right)$.*"
>
> Clearly, since a sample of the graph data (i.e., each data sample is a graph) is partitioned into subgraphs that are stored at different clients (such a graph partition can also be seen in **Fig. 1**), **different graph features (i.e., nodes) could be located at different clients**. This is similar to vertical FL and belongs to the **model parallelism** paradigm. In other words, in federated graph learning, **none of the clients holds a copy of the full GNN model**.
>
> Meanwhile, we suspect the reviewer's question regarding "IID vs. non-IID" may come from misunderstanding our federate graph learning as "graph-level" horizontal federated learning [1], where each client holds a subset of the full graph data samples. Such a "graph-level" FL is indeed data-parallel. However, such a graph-level FL system is **not** what we consider in this work.
>
> We continue our response to the follow-up comment regarding the misunderstanding of federated learning in the next block.

---

> ### Author Response · Authors · 2025-11-27
> **Response to Follow-Up Comment on the Misunderstanding of Federated Learning (Part 2)**
>
> Continuing our response to the follow-up comment concerning the misunderstanding of federated learning:
>
> **3) Clarification on our statement "conventional notion of non-IID data distribution in conventional FL is not applicable to federated GNN":** Our statement that *"the conventional notion of non-IID data distributions in conventional FL is not applicable to federated GNN"* was **not intended to dismiss the importance of data heterogeneity in FL**. Rather, it highlights the **structural and modeling differences** between **traditional FL** (operating on i.i.d./non-i.i.d. feature-label data) and **federated graph learning** (operating on graph-structured data).
>
>
> To clarify, we do not claim that the concept of non-IID is “not applicable” to FL in general. Our point is that **the conventional notion of non-IID data partitions**, which is typically defined by label distribution skew or feature distribution heterogeneity across clients, is **less directly applicable to graph-structured data**. In **traditional FL tasks** (e.g., image or text classification), non-IID data occur due to the heterogeneity of data generation/collection at different geo-locations. However, in **federated graph learning** (FGL), the graph-structured data is inherently connected across different clients, and the dataset splitting in traditional FL is not relevant to graph-structured data.
>
> For ease of contrasting and understanding, below is a summary of the key differences between FGL and traditional FL:
>
> |  | Federated Graph Learning | Traditional Federated Learning |
> | -------- | -------- | -------- |
> |**Data type**| Graph-structured data | Feature-label data|
> | **Data per client** | **Non-identical subgraphs** with inter-client dependencies (cross-client edges)| Independent local datasets (i.i.d. or non-i.i.d.)|
> | **Source of heterogeneity**| Structural (arising from graph topology, degree distribution, and neighbor dependencies) | Statistical (arising from variations in feature or label distributions) |
> |**Model architecture**| Graph Neural Network (GNN); model and data jointly partitioned (**model-parallel**) | Traditional neural networks; model replicated across clients (**data-parallel**) |
> |**Note**| Each client only holds a subgraph and lacks access to the full GNN model (e.g., message-passing over missing cross-client neighbors). This makes federated GNNs inherently model-parallel. | Each client operates independently on a local dataset (e.g., images or text), which typically conforms to the data-parallel paradigm. |
>
>
>
>
> While we acknowledge that our use of the term “non-IID” may have caused confusion, our intention was to emphasize that **conventional statistical notions of label/feature non-IID do not directly translate to graph data, which is structurally non-IID by default**.
>
> To simulate realistic and diverse FGL settings, we employ two widely used graph partitioning strategies:
> * *METIS Partitioning:* Connectivity-aware and load-balanced; produces realistic spatial locality in subgraphs.
> * *Random Partitioning:* Contrary to traditional FL (where it induces IID), **random partitioning in FGL leads to more fragmented neighborhoods, creating structurally unbalanced and topologically diverse subgraphs, which increases the cross-client dependency and graph-level structural heterogeneity**.
>
> This setup ensures our evaluations cover both **balanced** and **structurally heterogeneous** subgraph settings, aligned with standard practice in FGL literature.
>
> Finally, our convergence analysis accommodates stochastic sampling and partial client participation, both of which are key characteristics of real-world FGL deployments. Notably, we **do not make restrictive assumptions on subgraph structures** (e.g., homogeneity), making our analysis broadly applicable to a wide range of subgraphs with diverse structural heterogeneity.
>
> [1] He, C., Ceyani, E., Balasubramanian, K., Annavaram, M., & Avestimehr, S. (2021). Spreadgnn: Serverless multi-task federated learning for graph neural networks. arXiv preprint arXiv:2106.02743.

---

### Official Review · Reviewer_8oJB · 2025-11-01

**Soundness:** 3
**Presentation:** 3
**Contribution:** 3
**Rating:** 6
**Confidence:** 3

**Summary:**

This paper tackles federated graph learning (FGL) on geo-distributed graphs where nodes and edges are partitioned across clients, causing severe cross-client neighbor dependence. Existing methods either ignore these dependencies (hurting accuracy) or fully synchronize them (hurting efficiency and privacy).
The authors propose Swift-FedGNN, a mini-batch-based federated GNN algorithm that performs mostly local training and occasionally triggers cross-client training among a sampled subset of clients. In those rounds, each remote client aggregates its own neighbor embeddings and sends only aggregated representations to the server, which re-aggregates and forwards them to the requesting client—avoiding raw data exchange.

They provide a non-trivial convergence analysis without assuming unbiased gradients, bounding the bias introduced by neighbor sampling and missing cross-client neighbors. The algorithm achieves an $O(T^{-0.5})$ convergence rate matching single-machine sampling-based GNNs. Extensive experiments on five benchmarks (ogbn-products, Reddit, arxiv, flickr, citeseer) show that Swift-FedGNN attains comparable accuracy with 4× speed-up and 7–21× lower communication cost compared to strong baselines.

**Strengths:**

Clear motivation and relevance. The paper addresses a real bottleneck in FGL—communication-heavy cross-client neighbor access—through an elegant and realistic training schedule.

Algorithmic novelty. The two-stage neighbor aggregation and selective cross-client correction provide a principled and privacy-aware way to integrate distributed neighborhood information.

Strong theoretical grounding. The convergence proof relaxes the common unbiased-gradient assumption, offering bounds that explicitly model bias growth with GNN depth—a fresh theoretical contribution.

Empirical thoroughness. Evaluations cover multiple datasets, two architectures (GraphSAGE, GIN), partition types, and hyperparameter sensitivity. The improvements in both communication efficiency and convergence speed are significant.

Well-balanced trade-off analysis. The theory and experiments together demonstrate a tunable trade-off between communication frequency $(I, K)$  and residual information loss.

**Weaknesses:**

Limited novelty in aggregation design. While efficient, the idea of “aggregate-then-transfer” resembles known hierarchical aggregation patterns; the real novelty lies more in the federated scheduling and analysis than in the aggregation itself.

Evaluation scope. Although the method is shown on standard benchmarks, all datasets are homophilic. Performance on heterophilic or highly dynamic graphs (e.g., temporal, relational) is unknown.

Privacy and security claims. The paper asserts improved privacy due to aggregated communication, but lacks formal guarantees (e.g., differential privacy or leakage analysis).

Hyperparameter sensitivity. The method depends on correction frequency $I$ and sampling size $K$; although tested, guidance for tuning them across heterogeneous clients is limited.

**Questions:**

Gradient bias quantification: Can the authors provide empirical evidence (e.g., gradient variance plots) to support the theoretical claim that stochastic-gradient bias grows with GNN depth?

Generalization to heterophilic graphs: How would Swift-FedGNN behave on datasets where cross-client neighbors convey opposing label information?

Scalability in extreme settings: What happens when the number of clients $M$ is very large (e.g., hundreds)? Does the sampling of $K$ clients still ensure stable convergence?

Privacy leakage analysis: Even though raw features are not shared, could aggregated embeddings leak information about rare node attributes?

Applicability beyond node classification: Can the same framework handle link prediction or graph-level tasks where labels are not node-based?

---

> ### Author Response · Authors · 2025-11-22
>
> > **Weakness 1:** Limited novelty in aggregation design. While efficient, the idea of “aggregate-then-transfer” resembles known hierarchical aggregation patterns; the real novelty lies more in the federated scheduling and analysis than in the aggregation itself.
>
> **Response:** Thanks for your comment. We agree with the reviewer that designing new aggregation techniques is not the primary novelty of this paper, since the main focus of this work is on communication-efficient federated GNN learning algorithm design. However, our "aggregate-then-transfer" design is carefully tailored to address a key challenge in federated graph learning (FGL): how to efficiently exchange neighbor information without transferring raw features. Despite its simplicity, this aggregation mechanism enables the design of our communication-efficient FGL algorithm with both **theoretical convergence guarantees** under relaxed assumptions and **strong empirical performance** across diverse benchmarks.
>
> What distinguishes our approach is not just the hierarchical structure of aggregation, but how it is adapted to the FGL setting. In standard hierarchical aggregation (e.g., in federated or distributed learning), intermediate models or gradients are aggregated across devices. In contrast, our method performs **two-stage feature/embedding aggregation across graph neighbors**, where neighbors may reside on different clients, introducing unique structural constraints.
>
> That said, we acknowledge that the "aggregation-then-transfer" design *complements*, rather than solely defines, the core novelty of Swift-FedGNN. The main contributions lie in the integration of aggregation, periodic scheduling, and convergence analysis, each judiciously designed to support scalable, communication-efficient federated graph learning.
>
> > **Weakness 2:** Evaluation scope. Although the method is shown on standard benchmarks, all datasets are homophilic. Performance on heterophilic or highly dynamic graphs (e.g., temporal, relational) is unknown.
>
> **Response:** Thanks for your comment. We acknowledge that all datasets in our experiments are homophilic and would like to clarify why we focuses on federated graph learning for static and homophilic graphs, where the graph structure remains fixed over time and connected nodes tend to share similar features. This setting is particularly relevant for our goal of reducing communication cost, which remains a non-trivial challenge in federated graph learning. We view our work as laying the foundation for future extensions to more complex graph types, such as heterophilic or dynamic graphs.
>
> We agree with the reviewer that extending to heterophilic or dynamic graphs is an important and valuable direction for future research. However, the current design of Swift-FedGNN remains incompatible with these graph types, since the focus of this work is on addressing communication efficiency in federate graph learning, where even supporting homophilic graph datasets is already challenging and not yet well-understood in the literature. Supporting heterophilic or dynamic graphs would require further **specialized GNN architectures** capable of capturing dissimilar node interactions (in heterophilic graphs) or temporal structural evolution (in dynamic graphs), both of which introduce substantial modeling and system-level challenges [1,2]. Addressing such scenarios is beyond the scope and focus of this work and would merit a separate, dedicated study.
>
> That being said, we sincerely thank the reviewer for pointing out this interesting and important future direction!
>
> [1] Wang, Xiao, et al. "A survey on heterogeneous graph embedding: methods, techniques, applications and sources." IEEE transactions on big data 9.2 (2022): 415-436.
>
> [2] Kazemi, Seyed Mehran, et al. "Representation learning for dynamic graphs: A survey." Journal of Machine Learning Research 21.70 (2020): 1-73.

---

> ### Author Response · Authors · 2025-11-22
>
> > **Weakness 3:** Privacy and security claims. The paper asserts improved privacy due to aggregated communication, but lacks formal guarantees (e.g., differential privacy or leakage analysis).
>
> **Response:** Thanks for your comment. We agree with the reviewer that formal privacy guarantees, such as differential privacy (DP) or rigorous leakage analysis, are important for understanding the privacy implications of any federated learning framework.
>
> In this work, our primary privacy motivation is to avoid the direct transmission of raw node features, which are often privacy-sensitive in real-world graph applications (e.g., user attributes in social networks). Our “aggregate-then-transfer” design ensures that:
> * Only aggregated neighbor embeddings (not raw features) are shared across clients, and
> * No raw node information is directly exposed to other clients or the server.
>
> That said, we do *not* claim formal privacy guarantees (e.g., DP bounds) in this work, since simply using aggregation without Gaussian/Laplacian-type noise injection is unlikely to offer $(\epsilon,\delta)$-type DP guarantee. Instead, our focus is on reducing communication overhead in federated graph learning while improving practical privacy-preserving behavior through communication-efficient design.
>
> Importantly, the Swift-FedGNN framework is **compatible with standard differential privacy techniques** and federated encryption protocols, which can be integrated Gaussian/Laplacian-type noise injection to provide formal privacy guarantees. Such extensions are orthogonal to our current design and are left for future work.

---

> ### Author Response · Authors · 2025-11-22
>
> > **Weakness 4:** Hyperparameter sensitivity. The method depends on correction frequency $I$ and sampling size $K$; although tested, guidance for tuning them across heterogeneous clients is limited.
>
> **Response:** Thanks for your comment. We would like to respond to two distinct aspects in this comment point-by-point as follows:
>
> **1) Hyperparameter Sensitivity:** To better guide the tuning of the correction frequency $I$ and the client sampling size $K$, in this rebuttal period, we have conducted additional experiments to  validate the accuracy of Swift-FedGNN on the ogbn-products dataset under different settings of $I$ and $K$ in Tables 1 and 2, respectively. The results demonstrate that:
> * Increasing the correction interval ($I$) from 5 to 40 (Table 1) leads to only a minor accuracy degradation (from 88.91% to 88.44%), demonstrating that less frequent cross-client training still preserves model quality.
> * Decreasing the number of sampled clients ($K$) from 15 to 1 (Table 2) also results in a minor accuracy drop (from 89.22% to 88.47%), indicating that a small number of sampled clients is sufficient to maintain strong performance.
>
> These findings are consistent with our theoretical conclusion in Remark (3) in Line 406 on Page 8.
>
> Complementary to these findings, Tables 3 and 4 present the total communication cost needed to achieve a target accuracy of **87%** under the same parameter variations. The results show that:
> * Increasing $I$ and decreasing $K$ substantially reduce communication cost. For example, raising $I$ from 5 to 40 saves approximately **80%** of the communication overhead, while reducing $K$ from 15 to 1 saves approximately **94%**.
>
> Collectively, these results validate Swift-FedGNN’s ability to reduce communication without incurring major information loss and demonstrate that Swift-FedGNN provides a tunable balance between communication efficiency and accuracy, and the trade-off can be controlled via $I$ and $K$.
>
> **2) Guidance for tuning $I$ and $K$ in Practice:** We observe that a wide range of $I$ and $K$ settings yield strong accuracy while substantially reducing communication overhead. This flexibility of Swift-FedGNN having two "control knobs" in $I$ and $K$ makes it well-suited for practical deployment under varying resource constraints. Practitioners can choose $I$ and $K$ based on their specific system budgets and application requirements. Specifically, for resource-constrained clients, using a larger $I$-value (less frequent cross-client training) and a smaller $K$-value (fewer sampled clients) can significantly reduce communication without major accuracy loss. Conversely, for accuracy-critical applications with more communication budget, a smaller $I$-value and a larger $K$-value offer the best accuracy.
>
>
> Table 1: Final validation accuracy of Swift-FedGNN with different correction frequencies ($I$) on ogbn-products dataset, with the number of sampled clients fixed at $K = 10$.
> | Correction Frequencies ($I$) | 5| 10 | 20 | 40 |
> | -------- | -------- | -------- |-------- |-------- |
> | Validation Accuracy (%)     | 88.91 | 88.88     | 88.60     | 88.44 |
>
>
> Table 2: Final validation accuracy of Swift-FedGNN with different numbers of sampled clients ($K$) on ogbn-products dataset, with the correction frequency fixed at $I = 10$.
> | # of Sampled Clients ($K$) | 1 | 5 | 10| 15|
> | -------- | -------- | -------- |-------- |-------- |
> | Validation Accuracy (%)     | 88.47     | 88.72     | 88.88 |89.22|
>
>
> Table 3: Total communication cost of Swift-FedGNN with different correction frequencies ($I$) on ogbn-products dataset when achieving a target validation accuracy of 87%.
> | Correction Frequencies ($I$) | 5| 10 | 20 | 40 |
> | -------- | -------- | -------- |-------- |-------- |
> |Total Communication Cost (MB)|1344.0|675.5|324.5|275.0|
>
>
> Table 4: Total communication cost of Swift-FedGNN with different numbers of sampled clients ($K$) on ogbn-products dataset when achieving a target validation accuracy of 87%.
> | # of Sampled Clients ($K$) | 1 | 5 | 10|15|
> | -------- | -------- | -------- |-------- |-------- |
> |Total Communication Cost (MB)|57.8|342.2|675.5|1027.4|

---

> ### Author Response · Authors · 2025-11-22
>
> > **Question 1:** Gradient bias quantification: Can the authors provide empirical evidence (e.g., gradient variance plots) to support the theoretical claim that stochastic-gradient bias grows with GNN depth?
>
> **Response:** Thanks for your question. To empirically support our theoretical claim that the stochastic gradient bias increases with GNN depth, in this rebuttal period, we measured the gradient bias between the full gradient and the stochastic gradient on the ogbn-arxiv dataset under varying GNN depths. The results, summarized in the table below, clearly show that **the gradient bias grows as the GNN becomes deeper**, which is consistent with our theoretical analysis.
>
> Table: Gradient bias under varying GNN depths on the ogbn-arxiv dataset.
> | GNN Depth | 2 | 3 | 4 | 5 |
> | -------- | -------- | -------- | -------- |-------- |
> | Gradient Bias     | 0.46     | 0.78     | 1.12     |1.59|
>
>
>
> > **Question 2:** Generalization to heterophilic graphs: How would Swift-FedGNN behave on datasets where cross-client neighbors convey opposing label information?
>
> **Response:** Thanks for your question. We would like to clarify that the current design in Swift-FedGNN is **incompatible** with heterophilic graphs. To avoid repetition, for a more detailed explanation, please refer to our response to your *Weakness 2*.
>
>
> > **Question 3:** Scalability in extreme settings: What happens when the number of clients $M$ is very large (e.g., hundreds)? Does the sampling of $K$ clients still ensure stable convergence?
>
> **Response:** Thanks for your questions. In this rebuttal period, we have extended our experiments to two large-scale settings with **80 clients** and **100 clients** to more thoroughly evaluate scalability. Tables 1 and 3 below report the total communication cost on ogbn-products dataset when achieving a target validation accuracy, and Tables 2 and 4 present the final validation accuracy. These results show that:
> * With $K=10$, Swift-FedGNN reduces total communication cost by **at least 82%** in the 80-client setting and **at least 83%** in the 100-client setting compared to all baselines.
> * With $50\\%$ client sampling, Swift-FedGNN still achieves **at least 38%** communication savings in the 80-client setting and **at least 23%** communication savings in the 100-client setting over all baselines.
> * In both settings, Swift-FedGNN maintains comparable or better validation accuracy, with larger $K$ yielding slightly improved performance.
>
> These results confirm that even with a small subset of sampled clients, Swift-FedGNN ensures stable convergence, while significantly lowering communication overhead.
>
> Overall, this extended evaluation highlights Swift-FedGNN’s effectiveness and scalability, validating its communication efficiency even in large-scale federated graph learning settings. We also emphasize that Swift-FedGNN is designed to reduce communication overhead while maintaining competitive accuracy, not to surpass baselines in absolute accuracy.
>
>
> Table 1: Total communication cost on ogbn-products dataset with **80 clients** when achieving a target validation accuracy of 83%.
>
> ||Swift-FedGNN (K/M=10/80)|Swift-FedGNN (K/M=40/80)|FedGNN-G|LLCG|FedGNN-PNS|
> |-|-|-|-|-|-|
> |Total Communication Cost (MB)|711.5|2476.8|38154.2|3988.8|4601.8|
>
>
> Table 2: Final validation accuracy (%) on ogbn-products dataset with **80 clients**.
>
> ||Swift-FedGNN (K/M=10/80)|Swift-FedGNN (K/M=40/80)|FedGNN-G|LLCG|FedGNN-PNS|
> |-|-|-|-|-|-|
> |Final Validation Accuracy (%)|85.74|86.12|86.52|83.54|85.67|
>
>
> Table 3: Total communication cost on ogbn-products dataset with **100 clients** when achieving a target validation accuracy of 84.3%.
>
> ||Swift-FedGNN (K/M=10/100)|Swift-FedGNN (K/M=50/100)|FedGNN-G|LLCG|FedGNN-PNS|
> |-|-|-|-|-|-|
> |Total Communication Cost (MB)|1196.0|5367.7|60419.5|6987.4|8880.5|
>
>
> Table 4: Final validation accuracy (%) on ogbn-products dataset with **100 clients**.
>
> ||Swift-FedGNN (K/M=10/100)|Swift-FedGNN (K/M=50/100)|FedGNN-G|LLCG|FedGNN-PNS|
> |-|-|-|-|-|-|
> |Final Validation Accuracy (%)|85.07|85.53|85.63|84.35|85.15|

---

> ### Author Response · Authors · 2025-11-22
>
> > **Question 4:** Privacy leakage analysis: Even though raw features are not shared, could aggregated embeddings leak information about rare node attributes?
>
> **Response:** Thanks for your question. While Swift-FedGNN avoids transmitting raw node features, we acknowledge that aggregated embeddings could still leak sensitive information, particularly in cases involving extremely low-degree nodes or sparse subgraphs (e.g., where each node has only one neighbor).
>
> Our current design improves practical privacy protection by minimizing the exposure of raw features and restricting communication to coarsely aggregated information. However, we do not claim formal privacy guarantees (e.g., DP bounds) in this work, since simply using aggregation without Gaussian/Laplacian-type noise injection is unlikely to offer $(\epsilon,\delta)$-type DP guarantee.
>
> To further strengthen privacy and mitigate potential leakage, noise-injection-based DP techniques and/or federated encryption protocols can be integrated into Swift-FedGNN, since the Swift-FedGNN framework is **compatible** with all these mechanisms.
>
>
> > **Question 5:** Applicability beyond node classification: Can the same framework handle link prediction or graph-level tasks where labels are not node-based?
>
> **Response:** Thanks for you question. Swift-FedGNN **can** also be applied to link prediction tasks, as it supports modeling node interactions both within and across clients. The framework’s ability to aggregate cross-client neighbor information enables it to capture relational patterns necessary for predicting links between nodes distributed across different clients.
>
> Our Swift-FedGNN method also **works well** for graph-level tasks. For graph-level tasks where each graph represents an individual data sample (e.g., molecule classification) and no cross-client neighbor relationships exist, the problem setting aligns more closely with conventional federated learning. In such cases, our Swift-FedGNN framework reduces to a general-purpose federated learning algorithm for graph-structured data and remains applicable without requiring additional mechanisms for cross-client neighbor message passing.

---

### Author Response · Authors · 2025-11-30
**Summary of the Rebuttal Discussions**

Dear Area Chairs,

We sincerely thank you and all reviewers for their time, effort, and the constructive, insightful feedback on our work. We have made every effort to address all concerns within the rebuttal period. Before the discussion period was cut short and the review scores were reverted, several reviewers had acknowledged that our responses were satisfactory, and our paper’s score had increased to **8-6-6-6-6-4** (average: 6; Reviewer uxRE: 2 $\rightarrow$ 6, Reviewer uhnB: 6 $\rightarrow$ 8). We also note that this increase occurred **prior to** the identity-exposure incident. Since further discussion was no longer possible afterward, we summarize below the key points from our rebuttal that directly address the reviewers’ comments.

1) **Hyperparameter sensitivity analysis** *(Reviewers 8oJB, Aug9, uxRE, and qPZs):* We conducted a detailed sensitivity analysis of two key parameters in Swift-FedGNN: the correction frequency ($I$) and the number of sampled clients ($K$). The results show that increasing $I$ or decreasing $K$ consistently reduces validation accuracy, aligning with our theoretical conclusions in Remark (3). These results have been incorporated into the *Numerical Results* section in the revised version of our paper.


2) **Evaluations of large-scale settings** *(Reviewers 8oJB, Aug9, and uhnB):* To demonstrate scalability, we have extended our experiments to two large-scale settings with **80** and **100** clients using the ogbn-products dataset. The results show that Swift-FedGNN substantially reduces communication costs (e.g., by at least **82%** with fixed client sampling size $K=10$, and at least **23%** with a client sampling ratio of $K/M=50\\%$) compared to the baseline algorithms. These results have been added to the *Numerical Results* section in the revised version of our paper.


3) **Empirical validation of theoretical findings** *(Reviewers 8oJB, MwMc, uhnB, and Aug9):* We have added new empirical results showing that gradient bias increases and validation accuracy decreases with GNN depth, aligning with Lemma 5.4 and Theorem 5.6. These results are included in *Appendix G.1.1* in the revised version of our paper.

    In addition, we have provided a more concrete mapping between empirical accuracy degradation and the residual error term in Theorem 5.6, confirming a clear **quantitative** relationship (cf. *Response to Follow-Up on Empirical Alignment with the Residual Error Term in Theorem 5.6* for *Reviewer Aug9*).


4) **Communication and computation complexities** *(Reviewers uxRE, and uhnB):* We have formally and theoretically analyzed the communication and computation complexity of Swift-FedGNN and included the results in the *Theoretical Performance Analysis* section and *Appendix D* in the revised version of our paper. Our comparison with FedGNN-PNS, the most closely related baseline, shows that Swift-FedGNN achieves substantially lower communication costs.


5) **Why GNNs with non-element-wise operations (e.g., GAT) is fundamentally unfit for communication-efficient federated graph learning** *(Reviewers Aug9, MwMc, and uhnB):* We clarified that non-element-wise operations (e.g., GAT) are fundamentally not a good fit in any communication-efficient federated graph learning algorithm design due to the requirement of direct access to raw neighbor features/embeddings, leading to significantly high communication overhead. We have included a detailed discussion in *Appendix E* in the revised version of our paper.


6) **Clarifications on privacy preservation** *(Reviewers 8oJB, and uhnB):* Swift-FedGNN improves practical privacy protection by avoiding raw feature transmission and restricting communication to coarsely aggregated information. We do *not* claim theoretical privacy guarantees (e.g., differential privacy bounds), as this is not the focus of our work. That said, our Swift-FedGNN framework is compatible with standard differential privacy techniques and federated encryption protocols to ensure formal privacy guarantees. We have provided a detailed discussion in *Appendix F* in the revised version of our paper.


7) **Theoretical analysis extensions for GraphSAGE and GIN** *(Reviewer MwMc):* We chose GCN for the theoretical analysis due to its mathematical simplicity and tractability. Our convergence analysis naturally extends to other element-wise operation-based GNNs such as GraphSAGE and GIN. The changes only affect the exact form of the bias term in Theorem 5.6 due to architectural-specific adjustments. We have provided guidance on how to extend the analysis in *Appendix I* in the revised version of our paper.


We hope our responses have addressed all the reviewers' concerns, and we believe the revision has strengthened our paper and filled the gaps identified during the review process.

Thank you very much.

Sincerely,\
The Authors

---

### Meta-Review · Area_Chair_Nizm · 2026-01-04

**Summary:**

Reviewers primarily raised concerns about limited novelty. They stated that the core ideas build incrementally on existing federated GNN frameworks and aggregation strategies, rather than introducing a fundamentally new algorithmic paradigm. Reviewers commend on the sound theoretical analysis, but questioned whether the assumptions and convergence results meaningfully reflect practical federated graph settings and real-world heterogeneity. The empirical evaluation was also criticized for being limited to homophilic and static graphs, leaving generalization to heterophilic, dynamic, or more complex graph scenarios unclear. Finally, reviewers found the privacy claims to be qualitative rather than rigorous.

**Reviewer Concerns:**

Based on my reading, the rebuttal addressed the following empirical concerns. (1) It added large-scale experiments with up to 80 or 100 clients and confirmed strong communication savings. (2) It added additional sensitivity analyses for key hyperparameters and showed stable accuracy trends. (3) It empirically validated the theoretical claim that gradient bias increases with GNN depth. At the same time, core concerns remain outstanding: (1) The novelty of the aggregation mechanism relative to prior hierarchical or federated designs is still limited. (2) The method’s applicability to heterophilic, dynamic, or non-node-level graph tasks remains underdeveloped. (3) Privacy claims are still qualitative.

**Reviewer Scores:**

I see very little changes for all reviewers. There was raised concern about Reviewer Aug9, but I do not find the authors' claim to be substantially reasonable to change validity of AUg9's comments.

---

### Decision · Program_Chairs · 2026-01-26

Reject